# Deep Learning for BioImaging: What Are We Really Learning?

Ivan Svatko [* 1 2]  Maxime Sanchez [* 2 3 4 5 6]  Ihab Bendidi [2 7]  Gilles Cottrell [1]  Auguste Genovesio [2]

## Abstract

Representation learning has driven major advances in natural image analysis by enabling models to acquire high-level semantic features. In microscopy imaging, however, it remains unclear what current representation learning methods really learn. In this work, we conduct a systematic study of representation learning for the two most widely used and broadly available microscopy data types, representing critical scales in biology: cell culture and tissue imaging. We investigate whether, in contrast to natural images, existing models fail to consistently acquire high-level, biologically meaningful features. To this end, we introduce a set of simple yet revealing baselines on curated benchmarks, including untrained models and structural representations of cellular tissue. Our results show that, surprisingly, for a considerable subset of evaluation settings, the baselines are comparable to state-of-the-art methods, demonstrating that many commonly used benchmark metrics are insufficient to assess representation quality and often mask a lack of relevant high-level abstractions. In addition, we investigate how detailed comparisons with these baselines provide ways to interpret the strengths and weaknesses of models for further improvements. Together, our results suggest that progress in representation learning for microscopy requires not only stronger models, but also benchmarks that are more indicative of what is actually learned.

## 1. Introduction

Deep learning has transformed image analysis, supported by a large ecosystem of models, datasets, and benchmarks that make it easier to compare methods and reuse them for downstream applications. A similar shift is now happening in machine learning for biology, especially for microscopy imaging: data volumes are rapidly increasing (Chandrasekaran et al., 2023), and datasets are becoming more diverse in cell types, perturbations, and experimental settings. This scale has enabled the rise of *foundation models* trained on broad microscopy collections (Kenyon-Dean et al., 2025; Bioptimus, 2025), with the promise of transferring to many biological tasks. As the field tests how useful these models really are for downstream biology, benchmarking becomes essential, both to rank models fairly and to understand which capabilities actually transfer. Thus, in parallel, new microscopy benchmarking efforts have started to emerge, including tasks focused on cell culture imaging (Chen et al., 2023; Bourriez et al., 2024; Kraus et al., 2024) and histopathology (Jaume et al., 2024; Gindra et al., 2025).

However, systematic benchmarking in microscopy imaging remains limited, and this gap is not only about missing datasets, it is also about how biological images are produced. Microscopy experiments are sensitive to experimental conditions, batch effects, and spatial or plate layout artifacts, and deep learning models can learn these signals instead of the underlying biology (Sypetkowski et al., 2023; Arevalo et al., 2024; Haslum et al., 2024). While these risks are widely recognized in principle, the field has not yet clearly demonstrated (using controlled, benchmark-focused analyses) how much such biases can distort model training and evaluation in microscopy. Importantly, a related biological modality has already made this issue concrete. In transcriptomics, several recent benchmarking studies have shown that confounding structure in the data can strongly influence performance estimates, sometimes to the point where simple statistical baselines rival or even outperform large foundation models (Luecken et al., 2022; Bendidi et al., 2024; Ahlmann-Eltze et al., 2025; Csendes et al., 2025). These results suggest that benchmark scores can reflect dataset-specific shortcuts rather than true biological generalization. Taken together, this motivates a careful look at whether microscopy benchmarks may face similar hidden failure modes, and whether current evaluations are reliably

---

[*]Equal contribution  [1]Université Paris Cité, IRD, Inserm, MERIT, F-75006, Paris, France  [2]IBENS, École Normale Supérieure, Université PSL, Paris, France  [3]Institut Curie, Université PSL, Paris, France  [4]Iktos, Paris, France  [5]Mines ParisTech, Université PSL, Paris, France  [6]INSERM, U1331, Paris, France  [7]Valence Labs, Recursion, London, United Kingdom. Correspondence to: Auguste Genovesio <auguste.genovesio@ens.psl.eu>.

*Proceedings of the 43rd International Conference on Machine Learning*, Seoul, South Korea. PMLR 306, 2026. Copyright 2026 by the author(s).

measuring what they are intended to measure.

In this work, instead of focusing on the confounding structures, we propose to approach this problem by analyzing the discriminative capabilities of benchmarks with respect to the models of interest and baselines that either do not possess biologically relevant representations by construction or operate on strongly ablated input information.

More specifically, we investigate what microscopy foundation models truly learn by evaluating foundation models on curated cell-culture and tissue benchmarks, using ImageNet-1k as a natural-image reference. To make performance easier to interpret, we introduce two simple but informative baselines. First, *untrained models* use post-processed features from randomly initialized networks to test how much signal comes from architectural inductive biases and weak pixel correlations rather than learned biological content. Second, *disentangled tissue structures* represent histology images through the spatial organization of cells, testing whether models capture biological information beyond tissue morphology. We then benchmark widely used foundation models on these tasks and compare them to both baselines, finding that many models unexpectedly fall short. We summarize our contributions as follows:

- We demonstrate shortcomings of several popular benchmark tasks across organizational scales in bioimaging.

- We provide an analysis of layer-wise representations, raising concerns about the lack of consistent acquisition of high-level biologically relevant abstractions.

- We demonstrate relevance of structure-only views of tissue as a strong but incomplete sub-modality, suggesting future work to develop principled representations of tissues.

**Conflict of Interest Disclosure.** Phenomics models MAE-G/8, MAE-L/8, and OpenPhenom (Kraus et al., 2024; Kenyon-Dean et al., 2025) are developed by Valence Labs, Recursion, which employs one of the authors. These models are evaluated throughout the paper.

## 2. Related Works

**Benchmarking pitfalls in biological machine learning.** As biological machine learning shifts toward large pre-trained models, evaluation has become a central bottleneck: reported gains can reflect confounding structures (batch, lab, protocol, cohort, or platform effects) rather than a transferable biological signal. In microscopy, this issue is especially acute because experimental design and acquisition artifacts can imprint strong variation that models may exploit, motivating benchmark designs that explicitly test robustness

to batch structure and technical covariates (Sypetkowski et al., 2023; Arevalo et al., 2024). Closely related concerns have been demonstrated in transcriptomics, where several benchmarking studies show that confounders and dataset shortcuts can inflate performance estimates, and that simple baselines can match or outperform large models on perturbation prediction (Bendidi et al., 2024; Ahlmann-Eltze et al., 2025; Csendes et al., 2025; Wenteler et al., 2025; Wenkel et al., 2026). Importantly, these results highlight why *strong simple baselines* (e.g., linear models, cell-count, or random-feature embeddings) are not an afterthought: they calibrate what a score means and help detect when progress comes from benchmark-specific shortcuts rather than biological abstractions (Seal et al., 2025).

**Baselines for microscopy representations.** Across computer vision, untrained networks can already impose powerful image priors (Ulyanov et al., 2018; Heckel & Hand, 2019), and can exhibit non-trivial selectivity driven by architecture and initialization (Ramanujan et al., 2020; Baek et al., 2021; Kim et al., 2021). Related observations extend to transformer architectures, where non-trivial behavior can arise even when large parts of attention are fixed or random (Zhong & Andreas, 2024; Dong et al., 2025). In microscopy, simple and interpretable baselines such as CellProfiler-derived morphology features (Stirling et al., 2021), cell-count / confluency summaries (Way et al., 2021; Seal et al., 2025), and spatial-organization representations that emphasize cellular arrangement in addition to cellular morphology (Wang et al., 2023) often provide competitive reference points. Training-light or training-free transfer pipelines for high-content screening further show that strong performance can arise from reusing generic representations with minimal task-specific training (Corbe et al., 2023).

Transfer learning, randomly initialized encoders, and hand-crafted features have been used to illustrate training dynamics for a given architecture or method (Kang et al., 2023; Kenyon-Dean et al., 2025). We argue that including these baselines is essential to *calibrate* the benchmark's difficulty and to relativize reported gains under multiple sources of potential shortcuts. In contrast to these works, we conduct a systematic study of evaluation tasks, leveraging uninformed models as a pathway to better understanding the design principles of learning algorithms and their evaluation.

**Microscopy imaging benchmarks.** Benchmarking in microscopy spans diverse assay families and thus benefits from separating *cellular-level* and *tissue-level* organizational scales. For cell culture, widely used public resources include curated collections (Masud et al., 2023) such as BBBC (Ljosa et al., 2012), perturbation imaging datasets designed to expose experimental batch effects such as RxRx1 and related releases (Sypetkowski et al., 2023; Recursion, 2020),

large-scale Cell Painting efforts such as JUMP-CP (Chandrasekaran et al., 2023), and more task-specific suites targeting heterogeneity across channels and acquisition settings (Chen et al., 2023). Recent work has also moved toward making large industrial-scale screens more usable for benchmarking by publishing compressed subsets and standardized tasks (Kraus et al., 2025; Sanchez et al., 2026). For tissue imaging, canonical benchmarks include challenge-style datasets such as CAMELYON16 and PANDA (Ehteshami Bejnordi et al., 2017; Bulten et al., 2022), alongside newer multimodal benchmarks that pair histology with molecular readouts (e.g., spatial transcriptomics) to evaluate cross-modal transfer (Jaume et al., 2024; Gindra et al., 2025; Bendidi et al., 2025). Complementarily, distribution-shift benchmark frameworks provide standardized splits and protocols for robustness testing across domains, including pathology and microscopy (Koh et al., 2021).

**Microscopy imaging foundation models.** Microscopy foundation models are typically organized by modality. In cell culture imaging, large-scale self-supervised pretraining, using masked autoencoders and ViT-style backbones, has leveraged diverse Cell Painting datasets (Kraus et al., 2024; Watkinson et al., 2024; Kenyon-Dean et al., 2025), with extensions addressing channel and assay variability (Chen et al., 2023; Bourriez et al., 2024). For general-purpose fluorescence microscopy, models like Cytoself and SubCell aim to capture proteome-scale patterns (Kobayashi et al., 2022; Gupta et al., 2024). In histopathology, foundation models use both whole-slide and tile-level pretraining (Xu et al., 2024; Wang et al., 2024), self-supervised patch encoders tested across task suites (Chen et al., 2024), vision-language models (Lu et al., 2024), and multimodal approaches combining slides with reports and gene expression (Xu et al., 2025), with growing emphasis on scale and magnification diversity (Zimmermann et al., 2024; Bioptimus, 2025). Beyond pixel-based encoders, graph-based tissue models capture spatial cell organization and motivate structure-aware models (Wang et al., 2023). Given the dominance of ViT backbones, their architectural traits remain central to interpreting model behavior (Raghu et al., 2021; Darcet et al., 2024; Jiang et al., 2026).

## 3. Proposed Baselines & Benchmarks

Deep learning models can have performance confounders on microscopy benchmarks by exploiting low-level intensity cues or acquisition artifacts rather than biologically meaningful features. We therefore introduce a small set of intentionally simple baselines that (i) calibrate how well models of interest perform and (ii) help diagnose when an evaluation metric rewards shortcuts. The baselines span pixel statistics, randomly initialized encoders, and structure-only representations of tissue.

### 3.1. Baseline Strategies

We use three complementary baseline families to probe what information is sufficient to perform well: (1) *pixel-level statistics* that capture only global intensity distributions, (2) *untrained deep encoders* that isolate architectural inductive biases from learned biology, and (3) *disentangled tissue structure* representations that retain spatial cell organization while removing tissue staining and cellular morphology.

**Pixel-level baselines.** To test whether a task can be solved using only low-level intensity correlations, we construct two feature sets from per-channel pixel statistics: `pixel_mean` (channel-wise means) and `pixel_stats` (channel-wise mean, standard deviation, and skewness). These features ignore spatial layout and morphology, and therefore serve as a simple lower bound that is easy to interpret.

**Untrained models.** Untrained models are deep networks with randomly initialized weights, used as fixed feature extractors with the same embedding pipeline as their trained counterparts. Because they contain no *learned* visual concepts, they separate the contribution of architecture (e.g., convolutional locality or tokenization in transformers) from the contribution of representation learning. When an untrained model performs competitively, it suggests that benchmark performance may be driven by low-level cues or dataset structure that aligns with architectural priors, and that the metric may not be faithfully reflecting biologically meaningful feature learning.

**Disentangled tissue structure.** In histology, the spatial organization of cells is biologically informative (e.g., it can reflect tissue types or pathological conditions like cancer) (Wang et al., 2023). At the same time, spatial structure can also become a shortcut: a model may succeed by recognizing coarse architectural patterns (cell density, layering, gland-like arrangements) without learning cell morphology or texture cues that are required for many clinically and biologically relevant distinctions. To study this explicitly, we introduce a *structure-only* baseline that retains cell positions while discarding pixel appearance, enabling us to measure how much of the downstream signal is explainable by organization alone.

We formalize tissue structure as a set of 2D points defined by centroids of the cell nuclei segmented by a pretrained segmentation model. In our experiments, we use segmentation masks obtained from CellViT (Hörst et al., 2024), as provided by the authors of the benchmarks. We proceed to form two complementary views of this representation:

**(i) Binary images of cell graphs.** We build a cell graph from the centroids of segmented nuclei (e.g., using a simple spatial neighborhood rule) and render the resulting *edges* as

binary images at the base resolution of the image encoders (e.g. 224×224 pixels). This lets us apply both pretrained and untrained *image* models to the structure alone, while disentangling completely the cell morphology and stain/texture. Because node (cell nuclei) locations are in a fixed coordinate system, the rendering is deterministic; we control apparent magnification and visibility by adjusting the drawing scale and edge width. We provide the construction details in Appendix A and include additional controlled studies on synthetic graphs in Appendix C.3. Fig. 1 provides an example of a tissue patch and its corresponding cell-graph.

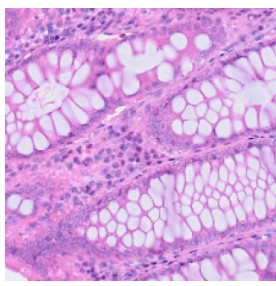 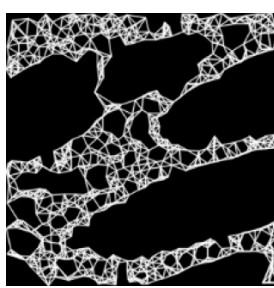

*Figure 1.* **Comparison of aggregated tissue patches (left) and 5-nearest-neighbor cell-graph images (right)**. Both images are rendered at 224×224 pixels. Aggregating tissue patches can expose larger-scale tissue organization in the graph rendering. Cell nuclei are revealed in a dark blue color via H&E staining.

**(ii) Point-cloud view of cell positions.** For completeness, we also treat sets of cell nuclei centroids as point clouds and apply geometric deep learning methods (Bronstein et al., 2021). This view naturally supports permutation invariance of nodes and makes it more straightforward to test hypotheses about which aspects of organization matter (e.g., node count, density, local alignment, or mild position noise). While rotation invariance is desirable, some other common assumptions, such as symmetry with respect to mirror images and robustness to point density do not hold in general and are thus not enforced. This representation also extends naturally to emerging 3D tissue settings (Lin et al., 2025).

### 3.2. Experimental Setup

To keep the analysis focused on representation quality rather than architectural novelty, we evaluated a set of widely used vision backbones and simple geometric encoders under a consistent embedding and evaluation pipeline. A detailed description of model configurations and pretrained weights is provided in Appendix B.

**Selected image architectures.** For image encoders we report both pretrained and randomly initialized variants and probe intermediate layers to study how performance changes across depth. Concretely, we include: (i) a *single-layer CNN* (random local filters followed by global pooling) as a minimal inductive-bias baseline (`SingleConv`, (ii) *ResNet* models (He et al., 2016) pretrained on ImageNet-1k (Russakovsky et al., 2015) (and their random counterparts), and (iii) *ViT* models (Dosovitskiy et al., 2021) pretrained on ImageNet-21k (Ridnik et al., 2021) (and their random counterparts). Unless stated otherwise, we select four representative configurations per model family and evaluate four intermediate stages per network to compare shallow vs. deep representations.

**Selected geometric architectures.** For cell-centroid point sets, we use standard message-passing GNNs in the MPNN framework (Gilmer et al., 2017), instantiating the neighborhood aggregation with *GCN* (Kipf & Welling, 2017) and *EGNN* (Satorras et al., 2021). Because there is no single widely adopted "structure-only foundation model" for these inputs, we complement the main benchmarks with controlled experiments on synthetic graphs to validate and interpret the behavior of these encoders (Appendix C.3).

### 3.3. Benchmarking Setup

We evaluate representations in two organizational scales for microscopy imaging: the cellular and tissue levels, chosen to probe complementary biological scales. These scales are represented by images of cell culture and WSIs of extracted tissue samples. We include a natural-image reference to highlight where common representation learning intuitions do and do not transfer. Across all benchmarks, we extract frozen embeddings and follow the dataset-specific aggregation and evaluation protocols described below. Imaging modalities are described in Appendix A.

**Cell culture.** To evaluate representations on Cell Painting images, we use three complementary retrieval benchmarks.

We evaluate on *RxRx3-core* (Kraus et al., 2025) using the gene-gene retrieval task and following the original aggregation protocol. For each model, we extract image features and aggregate them into gene-level profiles. The original benchmark reports recall from a single fold and includes both the top and bottom 5% similarity tails. We find that the lower tail contains almost no informative pairs of interest (Fig. 23). We therefore (i) restrict the metric to the top 5% tail for stability, and (ii) use three folds to enable variance estimation (additional details are provided in Appendix A.2). We then compute pairwise cosine similarities between all 736 gene profiles and report recall@5%, defined as the fraction of literature-supported functional gene pairs that fall within the top 5% most similar pairs. This tests whether the representation organizes genes by known biology beyond chance. We analyze the effect of the similarity cut-off (Fig. 21) as well as a comparison to original benchmark scores (Fig. 18).

We also evaluate on *JUMP-CP* using a chemical perturbation retrieval task, following the feature extraction and aggregation pipeline of (Sanchez et al., 2026). We focus on eight positive-control compounds that are consistently present across all experiments and laboratories and exhibit diverse yet reproducible phenotypic effects (Fig. 9). We use a five-fold protocol: each fold contains approximately 800 five-channel images per compound, producing roughly 110 aggregated compound profiles per compound and per fold. We report mean average precision (mAP) computed by treating each compound profile as a query and ranking all other profiles by cosine similarity. High mAP indicates that replicate profiles of the same compound are consistently retrieved ahead of profiles from other compounds. Additional details can be found in A.2.

Finally, we perform mechanism of action (MoA) retrieval on JUMP-MoA, following SPACe evaluation framework (Stossi et al., 2024), designed to evaluate whether morphological profiles can retrieve both replicates of the same compound and compounds sharing an annotated mechanism of action. The experimental setup is detailed in A.2.

**Cellular tissue.** To probe tissue-scale representations and analyze structure-aware baselines, we focus on gene expression prediction in the context of spatial transcriptomics (ST) and image classification for cancer subtyping.

We use ST samples from *HEST-1k* (Jaume et al., 2024), which pairs H&E patches with gene expressions. This setting is particularly diagnostic: accurate prediction across many genes requires embeddings that integrate information spanning both single-cell morphology and the surrounding tissue context. We follow the original benchmark task: predicting expression of 50 highly variable genes from patch embeddings within grouped tissue types and pathologies. Performance is measured by gene-wise Pearson correlation coefficient (PCC) across test patches under patient-stratified cross-validation folds (Appendix A.3). HEST-1k patches correspond to individual ST spots and often contain relatively few cells, limiting expressiveness of structure-only inputs. We therefore, in addition to the original version, add a coarser, *binned* version by grouping adjacent spots into larger regions and averaging the corresponding gene expression readouts. We run the experiments on both the original and binned variants, report non-binned results in tables 8 and 9, and analyze the impact on in-domain encoders and structure-only models in Appendix D.

The breast cancer subtyping experiments follow the experimental setup of GraphHist (Öğüt et al., 2026), a recent work on (morphology-attributed) cell graphs. We focus on BRACS (Brancati et al., 2022) and BACH (Aresta et al., 2019), which are two region-of-interest (RoI)-level datasets.

**Natural images.** To contextualize microscopy results, we include a standard natural-image reference using *k*NN probing on ImageNet-1k. This comparison highlights how representation quality evolves across intermediate layers in a setting where high-level semantic features are known to emerge reliably, and clarifies which observations in microscopy are genuinely atypical rather than artifacts of our evaluation pipeline.

## 4. Results

### 4.1. Cellular Level

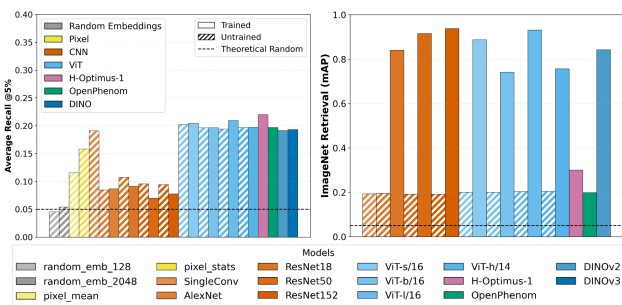

*Figure 2.* **Pretrained versus untrained representations on natural and microscopy images.** *Left*: gene–gene retrieval performance on RxRx3-core (Recall@5%). Mean recall across all 5 literature datasets is reported. *Right*: kNN top-1 accuracy on ImageNet-1k across models.

Fig. 2 shows gene-gene retrieval performance on RxRx3-core (recall@5% of cosine similarities). Unexpectedly, untrained ViTs perform comparably to pretrained ViTs and foundation models like OpenPhenom. Moreover, a minimal untrained `SingleConv` baseline is competitive with the best-performing methods, while pixel-statistics baselines recover a substantial fraction of the signal. In contrast, ResNet representations, whether pretrained or untrained, perform poorly on this task.

Results are stable across three random seeds and three folds, indicating limited impact of weight initialization (table 11 for RxRx3-Core and table 12 for JUMP-CP).

Following (Zhong & Andreas, 2024), who attribute the success of untrained models on certain tasks to their ability to identify suitable low-dimensional subspaces, we analyze the relationship between representation dimensionality and recall performance. For each model, we compute PCA on the final gene-level embeddings and report the fraction of variance explained by the first two components compared to recall performance (Fig. 20). We observe that ResNet-based representations collapse into low-dimensional subspaces where most of the variance is captured by the first two components, coinciding with weaker recall. This trend general-

izes across biological interaction datasets (Fig. 25). In contrast to collapsed ResNet-based representations, pixel-based baselines and ViT representations span substantially higher-dimensional subspaces and consistently achieve stronger retrieval performance.

We observe a similar phenomenon on natural images when relating ImageNet-1k $k$NN top-1 accuracy to the PCA explained variance (Fig. 20). The embeddings of untrained models collapse into low-dimensional subspaces, failing to capture the structure of ImageNet-1k. Notably, some intermediate layers of pretrained models also exhibit this collapse, and their respective performance also remains low.

Notably, both untrained and pretrained ViTs occupy similar high-dimensional spaces and achieve comparable performance on RxRx3-core, forming two close clusters (Appendix Fig. 20).

To validate whether models retrieve the *same* gene relationships, we compute gene-gene similarity rankings across all genes (270k pairs) for each model and measure pairwise Spearman correlations between rankings (Fig. 3 for a selected subset of models and Fig. 24 for all models). Pretrained and untrained ViTs yield highly correlated rankings, indicating shared relational structure; their rankings also correlate with DINOv3 and the SingleConv baseline.

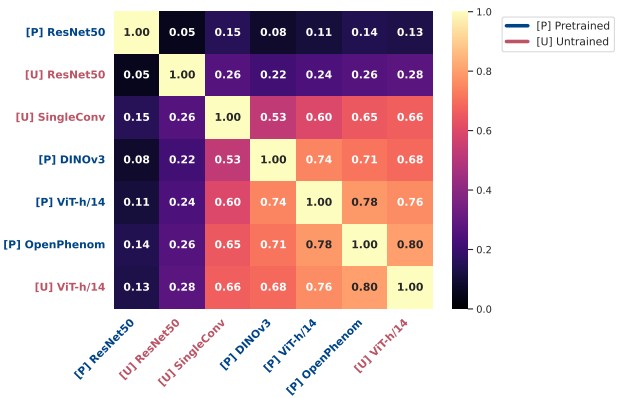

*Figure 3.* **Spearman ranking correlation of gene–gene rankings across model architectures.** The heatmap displays pairwise Spearman correlations between gene–gene similarity rankings induced by different representations on RxRx3-core genes across 269,745 pairs. Models are hierarchically clustered to reveal functional groupings based on their representational alignment. Labels prefixed with **[P]** (blue) denote pretrained models, while **[U]** (red) indicates untrained/random initializations.

Finally, for each model, we select four intermediate layers to evaluate. By averaging performance across architectures and layers (Fig. 4), we observe that performance generally decreases or remains stable in deeper layers for biological recall, in contrast to the monotonic improvements typically

observed on natural image benchmarks (Fig. 4). Here, all models except for the untrained ones demonstrate an increase in accuracy when evaluating hidden representations from deeper layers.

A hyperparameter sweep over untrained ViTs across layers, including ViT size, patch size, and aggregation methods was also performed. Overall, these factors had limited impact on performance. However, we observed a slight tendency favoring earlier layers, smaller patch sizes, smaller models, and CLS-token aggregation (see table. 10).

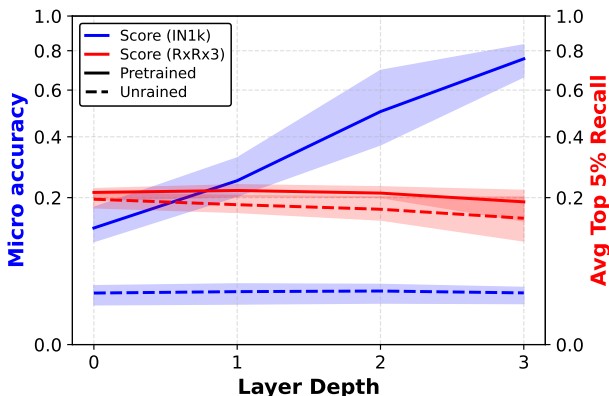

*Figure 4.* **Layer-wise evolution of representation quality across domains.** Average performance across architectures as a function of network depth for natural images (ImageNet-1k kNN accuracy) and cell-culture microscopy (RxRx3-core recall@5%). Average, minimal, and maximal scores are reported for each configuration.

For the JUMP-CP benchmark, per-compound mean average precision (mAP) results are shown in Fig. 5. Several compounds (e.g., JCP2022_050797 and JCP2022_085227) exhibit near-random retrieval performance across all models, suggesting little to no detectable phenotypic effect. Other compounds (e.g., JCP2022_012818) are retrieved almost equally well by untrained baselines and pretrained models, indicating easily detectable phenotypic changes. In contrast, some compounds (e.g., JCP2022_025848) are only reliably retrieved by the pretrained models, suggesting that they capture more subtle discriminative features. Overall, pretrained and foundation models consistently outperform untrained and pixel-based baselines. However, untrained baselines provide an important reference point for quantifying the amount of signal gained through learning. For example, although JCP2022_037716 achieves a higher absolute score than JCP2022_046054, the relative improvement over untrained baselines is larger for JCP2022_046054.

Another layer-wise effect emerges on JUMP-CP (Fig. 5). For untrained baselines and out-of-domain (OOD; e.g., ImageNet-pretrained backbones) models, performance typically saturates or slightly decreases in the final layers. In

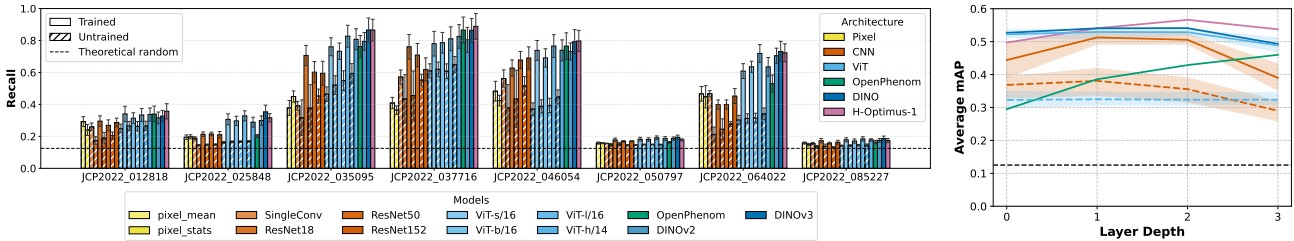

*Figure 5.* **Mean average precision (mAP) per compound on the JUMP-CP benchmark.** Bar plots show mAP scores for eight positive-control compounds. Each bar corresponds to a model configuration, grouped by architecture family and colored accordingly. Solid bars denote pretrained models, hatched bars denote untrained models, and pixel-based baselines are included for reference. Error bars indicate variability across evaluation folds. The dashed horizontal line indicates the theoretical random baseline.

contrast, in-domain (IID) models trained on similar cell-painting data (e.g., OpenPhenom) tend to improve with depth, with the largest gains observed for a subset of compounds (Fig. 26).

To complete this part of our analysis with a cautiously optimistic perspective we evaluate the recently released RxRx3-core embeddings from MAE-L/8 (Kraus et al., 2024) and MAE-G/8 (Kenyon-Dean et al., 2025) models. These backbones were pretrained on large proprietary in-distribution datasets as described in Kraus et al., (2024) and Kenyon-Dean et al., (2025) respectively. Benchmarking on a balanced split of RxRx3-core shows a considerable improvement over all pretrained and untrained baselines, reaching 1.5 times higher averaged top 5% recall (Appendix G). Since the pretrained backbones remain private, we cannot proceed with our layer-wise comparative setup or study the generalization to other datasets. While this limits our conclusions about the acquisition of relevant high-level abstractions, the obtained results are an encouraging sign for further work.

We also evaluated a subcellular cell culture task following SPACe (Stossi et al., 2024): the evaluation of the compound Mechanism of Action (MoA) retrieval. On this task, untrained models are on par with pretrained ones. Additional details are provided in Appendix H.1.

*Table 1.* Performance of the best-in-class model on a selection of HEST-1k-1NN datasets. Mean and standard deviation across cross-validation folds are reported. The best structure-based models were pretrained on natural images.

| MODALITY | TRAINING | CCRCC | PRAD | SKCM | COAD |
|---|---|---|---|---|---|
| FULL IMAGE | NO | 0.12 ± 0.14 | 0.25 ± 0.06 | 0.34 ±0.06 | 0.27 ± 0.08 |
| FULL IMAGE | YES, OOD | 0.26 ±0.07 | 0.41 ± 0.01 | 0.62 ± 0.07 | 0.30 ± 0.07 |
| FULL IMAGE | YES, IID | **0.37** ± 0.06 | **0.49** ± 0.02 | **0.68** ± 0.03 | **0.38** ± 0.06 |
| STRUCTURE | YES, OOD | 0.16 ± 0.04 | 0.38 ± 0.04 | 0.45 ± 0.07 | 0.31 ± 0.01 |

## 4.2. Tissue Level

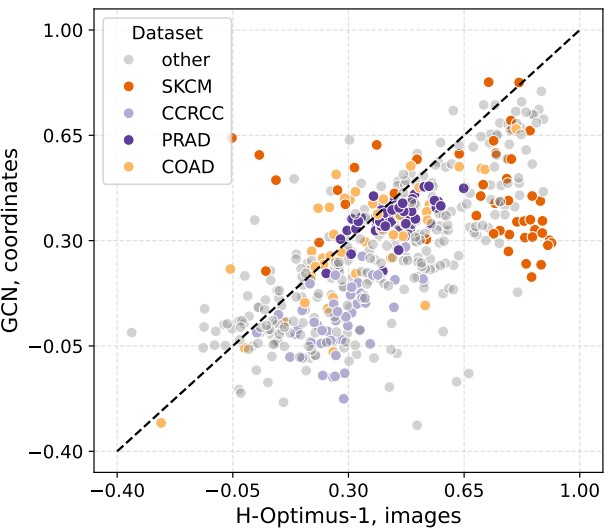

*Figure 6.* **Gene-wise average test PCC across cross-validation folds.** The best structure-based model is compared against H-Optimus-1 on selected datasets from HEST-1k-1NN. Each point represents one of the 50 dataset-specific target genes.

In table 1 we present averaged metrics for the *best performing model* for each selected dataset-modality-training setting (scores of individual models can be found in Appendix E). As expected, in-domain foundation models offer a substantial increase in performance across most datasets. Surprisingly, however, for COAD and PRAD the performance of structure-based models is competitive with OOD vision encoders.

To further investigate this phenomenon we compare performance of the best image model against the best structure model for each of the 50 target genes in Fig. 6. On PRAD and COAD, we observe a surprisingly competitive scores between a structure-only encoder and H-Optimus-1 for a large

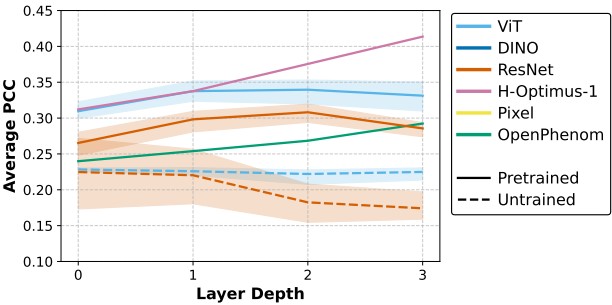

*Figure 7.* **Layer-wise performance on HEST-1k-1NN.** Comparison of performance of out-of-domain pretrained and untrained models and foundation models trained on biological modalities. For each architecture the global average across the benchmark and minimum/ maximum score across model configurations are reported.

*Table 2.* Impact of pretraining on structure-only models on HEST-1k-1NN. We report average gene-wise PCC per dataset and the absolute improvement over the untrained version in parenthesis. Standard deviations for cell count are computed over the cross-validation folds.

| MODEL | TRAINING | CCRCC | PRAD | SKCM | COAD |
|---|---|---|---|---|---|
| CELL COUNT | No | 0.11 ± 0.07 | 0.37 ± 0.05 | 0.40 ±0.06 | 0.30 ± 0.07 |
| GCN | YES, SYNTH. DATA | 0.04 (-0.02) | 0.38 (+0.04) | 0.45 (+0.08) | 0.31 (+0.04) |
| EGNN | YES, SYNTH. DATA | 0.04 (-0.01) | 0.36 (+0.005) | 0.22 (-0.05) | 0.28 (-0.02) |
| RESNET152 | YES, IN1K | 0.11 (+0.04) | 0.34 (+0.09) | 0.37 (+0.23) | 0.24 (+0.14) |

subset of genes. The foundation model achieves an impressive PCC of >0.8 on several genes, while struggling to reach positive correlation with several genes well predicted from the cell coordinates. Our structure-based baseline reaches its best scores on SKCM, suggesting further questions regarding relevant biological functions of the associated genes.

We then compare predictions of pretrained and untrained models. Table 1 shows that untrained models offer better-than-random performance and help provide additional context about the difficulty of the target. Notably, the performance of random encoders on SKCM is more than 2.5 times higher than on CCRCC, despite the task remaining the same.

Nonetheless, in contrast to the experimental results on RxRx3-core, subsets of HEST-1k like CCRCC and COAD demonstrate that in-domain pretraining offers a clear advantage. We proceed to investigate whether pretrained models form relevant deep-layer features (Fig. 7). Indeed, the performance of intermediate layers improves with depth for both IID and OOD models. Moreover, for the in-domain H-Optimus-1 the outputs of the very last stage provide the best results, suggesting that the model has learned relevant high-level features. Interestingly, OpenPhenom, pretrained

on images of cell culture demonstrates a similar trend. Additionally, H-Optimus-1 shows strong performance in experiments on OOD JUMP-CP and RxRx3-core (Fig. 5 and Fig. 19b respectively).

Finally, in the absence of established foundation models we investigate the contribution of different pretraining strategies on the structure-based modalities. Table 2 shows that both natural images and synthetic data can provide a meaningful training signal for structure encoders. It is generally generally true for image-based encoders (table 9), making this baseline easy to implement. Synthetic pretraining of GNNs yields varying results, but can be successful as with our GCN model. Yet, it is possible that the model implicitly leverages even simpler patterns like local cell count, a strong hand-crafted baseline, as discussed in Appendix D.

We conclude the study of spatial organization in tissues with an application to two breast cancer subtyping datasets. Consistently with the observations above, structure-only baselines provide non-trivial performance across BRACS and BACH RoI for simple linear probing and multiple instance learning (MIL) settings. Appendix H.2 details the setup and the obtained results.

## 5. Discussion

### 5.1. What Current Microscopy Benchmarks Certify

Across cell culture (RxRx3-core, JUMP-CP) and tissue (HEST-1k, HEST-1k-1NN), the results show that benchmark performance does not consistently track acquisition of high-level biological abstractions. On RxRx3-core, several pretrained and foundation models perform comparably to intentionally simple baselines, implying that part of the necessary signal is accessible without learning biologically meaningful representations. In contrast, on selected subsets of JUMP-CP and HEST-1k, in-domain pretraining yields clearer gains, indicating that learning can clearly matter when the task exposes subtler or modality-aligned signals. Overall, the experiments support a conservative reading of absolute scores: they can reflect a mixture of biology, architectural priors, and low-level correlates rather than biological abstractions alone.

### 5.2. Baselines and Diagnostics that Change Interpretation

Therefore, strong simple baselines are necessary to interpret a score. On RxRx3-core, untrained ViTs and SingleConv are competitive with pretrained models, while pixel-statistics features achieve non-trivial recall despite discarding spatial structure. Moreover, pretrained and untrained ViTs exhibit highly correlated gene–gene similarity rankings (Spearman), implying that they exploit essentially the same relational signal for the benchmark. These

observations are direct evidence that architectural inductive bias and dataset-accessible cues can dominate measured performance on this task. Separately, representational collapse (high variance explained by the top PCA components) coincides with weak biological recall for ResNet-based representations, while higher-dimensional embeddings (ViTs, pixel baselines) align with stronger recall; the same dimensionality diagnostic separates successful vs. unsuccessful behavior in ImageNet-1k $k$NN probing. Effective dimensionality is therefore a useful sanity check for collapse, but it does not establish biological semantics.

### 5.3. Design Principles for Effective Benchmarks

Our study leads to several actionable insights for improving benchmarking practices. First, a fundamental starting point is addressing quality control (QC) on images (Fig. 10). Notably, in some cases it might be more feasible than ensuring consistency of outcomes in the underlying biological experiments. Second, relevant baselines should be selected beyond theoretical random to highlight the relative complexity of the task. Furthermore, benchmarks should exhibit a clear separation between informed and uninformed baselines and, ideally, IID and OOD-trained models, allowing better detection of learned high-level abstractions. Additionally, greater emphasis should be put on benchmark granularity, enabling *calibration subtasks*. Finally, where applicable, defining realistic upper bounds for the relevant metrics is valuable to further improve our understanding of the achieved progress. For instance, biological replicates can be leveraged as the "best biological predictors", assuming reasonable QC.

### 5.4. Spatial Organization in Tissue Is a Strong Modality with Open Causal Questions

On HEST-1k-1NN, structure-only models (cell-centroid graphs) approach and sometimes exceed OOD image baselines on selected tissue categories (notably PRAD and COAD), and gene-wise analyses show subsets of targets that are comparatively predictable from cell coordinates. At the same time, in-domain histology foundation models consistently perform best on several subsets (e.g., CCRCC, SKCM), demonstrating that morphological cues learned from histology provide additional information beyond organization alone. This suggests further exploration of methods that integrate these signals in a disentangled interpretable way. A plausible alternative explanation for the structure-only performance is correlations with local cell count: the cell-counting baseline is strong, and while structure-based models can demonstrably capture additional organizational features (as studied on synthetic data) it does not yet prove that arrangement (rather than count) drives the gains on real tissue. Stronger evidence would require matching analysis (conditioned on cell count), or controlled interventions beyond synthetic data.

## 6. Future Work

The scope of the presented work focuses on re-contextualizing benchmark results by introducing baselines that are sensitive to simple shortcuts. However, to see if the models leverage these shortcuts, we rely primarily on prediction alignment. This limitation invites further interpretability work and an analysis of the formation of high-level abstractions through the lens of representation alignment across in-domain, out-of-domain, and uninformed models at the global and local scales (Huh et al., 2024; Gröger et al., 2026).

Successful applications of single cell morphology, as also supported by evaluation of CellProfiler embeddings (Appendix G) for cell culture and hand crafted morphological features of cell nuclei in tissue (Appendix H.2), suggest further investigations into how global spatial organization interacts with local morphology as two complementary sub-modalities. Their efficient fusion might go beyond morphology-attributed cell graphs and could support further development of inherently interpretable architectures for cellular imaging with a principled focus on organizational scales in biology.

## 7. Implications

The results support three concrete practices for microscopy representation benchmarks: (i) report strong simple baselines (pixel statistics, untrained encoders, and low-dimensional proxies such as cell count) to contextualize scores; (ii) report layer-wise curves and layer selection, since depth does not behave uniformly across microscopy tasks; and (iii) include diagnostic breakdowns (e.g., ranking correlations, per-compound and per-gene analyses) to detect when performance is driven by shortcuts rather than robust biological signal.

### Software and Data

All real data and pretrained models used in this study are public and are either open source or have publicly available embeddings. We release the code[1] to ensure reproducibility.

### Acknowledgements

This work was performed using HPC resources from GENCI–IDRIS (Grant 2025-AD010316962).

The authors are grateful to the reviewers for their valuable feedback, which helped strengthen the evaluation and further develop the perspectives outlined in this work.

---

[1]`https://github.com/ivsvat/baselines-allyouneed`

## Impact Statement

This paper presents work whose goal is to advance the evaluation of Machine Learning in its application to life sciences. There are many potential societal consequences of our work, especially relating to how improvement in the benchmarking space for machine learning in biology would positively impact the discovery of new biological relationships and potential drug treatments. Utmost care should be taken to validate safety and efficacy of model predictions in preclinical trials.

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

# A. Data Modalities and Datasets

## A.1. Natural Images

**ImageNet-1k.**  As a well-studied reference dataset of natural images we use ImageNet-1k (ILSVRC 2012) as described in (Russakovsky et al., 2015). The dataset consists of 1000 classes and is split into train, validation and test subsets (of 1,281,167 / 50,000 / 100,000 images respectively). In our experiments we use the train / validation split. Samples from the 8-class subset used for mAP computation (Fig. 22b) can be seen in Fig. 8. The classes were chosen based on KNN performance of DINOv2-g/14, more specifically we take the two best and the two worst classes with respect to their validation F1 scores. The remaining four classes were sampled randomly.

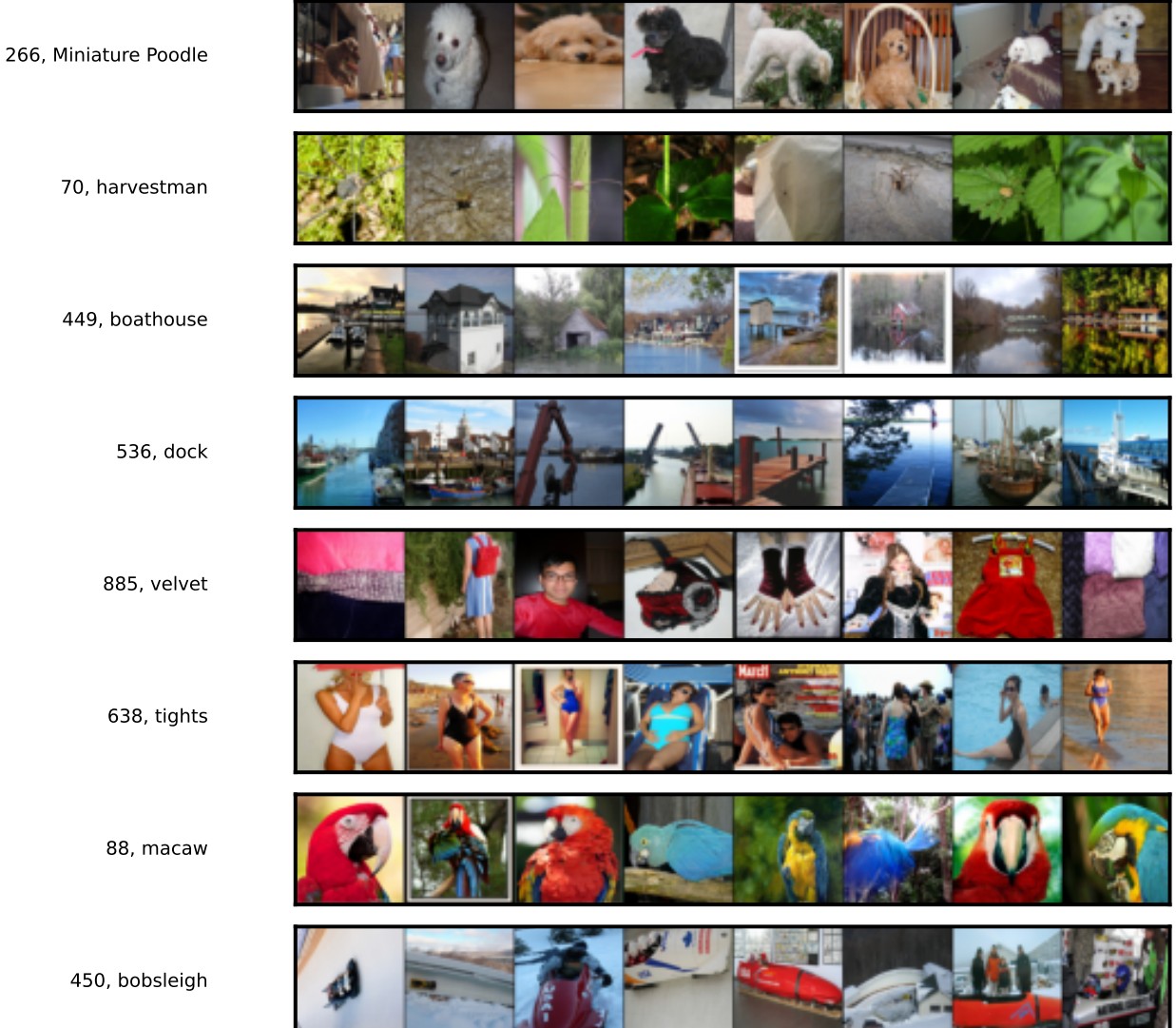

*Figure 8.* Samples from 8 classes sampled for the mAP experiment.

## A.2. Cell Painting

Cell Painting (Bray et al., 2016) is a standardized fluorescence microscopy assay that captures diverse visual characteristics of cells at single-cell resolution. Cells are labeled using a fixed set of fluorescent markers, each highlighting a different cellular component (compartments and organelles, such as the nucleus, cytoskeleton, mitochondria, endoplasmic reticulum, and plasma membrane), and imaged across multiple channels to produce rich, multi-channel microscopy images. Images are typically acquired across five to six fluorescence channels, producing multi-channel microscopy images that encode diverse aspects of cell morphology, organization, and subcellular structure.

Cell Painting is widely used in large-scale biological and pharmaceutical studies, including drug discovery, genetic perturbation screening, and functional genomics. Its central premise is that perturbations that affect similar biological pathways induce similar cellular phenotypes, which can be detected through changes in cell morphology and organization. As a result, Cell Painting has become a key modality for representation learning, where the goal is to learn feature embeddings that capture biologically meaningful variation across perturbations with or without task-specific supervision.

From a machine learning perspective, Cell Painting datasets pose several challenges. They are high-dimensional, multi-channel, and exhibit strong sources of variation unrelated to biological signal, such as technical, called batch effect, imaging artifacts, and cell-cycle heterogeneity. Moreover, biological semantics are indirect: labels often correspond to treatments or genes rather than explicit visual concepts. Consequently, evaluating representation quality is non-trivial and typically relies on downstream proxy tasks, such as perturbation matching or gene–compound association retrieval.

These properties make Cell Painting pertinent for understanding whether representation learning methods can move beyond appearance-driven features and capture higher-level, biologically meaningful abstractions.

**JUMP-CP.** JUMP-Cell Painting is a large-scale microscopy dataset generated by the Joint Undertaking for Morphological Profiling (JUMP) Consortium, a collaboration between ten pharmaceutical companies, six technology partners, and two non-profit organizations (Chandrasekaran et al., 2023). The dataset comprises Cell Painting images of human osteosarcoma (U2OS) cells subjected to diverse perturbations, including chemical treatments, gene overexpression, and CRISPR-Cas9 knockouts. JUMP-CP includes over 116,750 compounds, 12,602 gene overexpression perturbations, and 7,975 gene knockouts, totaling approximately 115 TB of data and capturing single-cell profiles for more than 1.6 billion cells. Each experimental compound plate, across all batches and laboratories, contains the same eight positive-control compounds (Fig. 9) and negative controls (DMSO only). We used these shared controls are used to define a standardized benchmark for evaluating representation robustness across experimental conditions.

**Our JUMP-CP subset benchmark.** We construct a balanced evaluation benchmark by selecting five folds, each composed of four experimental compound plates from each the seven laboratories providing 384-well plates. Across these $28 \times 5$ plates, we use all control wells, including positive controls for retrieval metric computation and negative controls (DMSO) for batch-effect mitigation. This results in approximately $6,400 \times 5$ five-channels images, forming a diverse subset in terms of both laboratory origin and phenotypic variation, which we use to evaluate produced representation of diverse encoders.

**RxRx3-core.** RxRx3 is a large-scale microscopy benchmark designed to evaluate representation learning for cellular imaging under biologically relevant perturbation tasks. It comprises fluorescence six-channels Cell Painting images of cells subjected to both gene knockdown and compound perturbations across multiple experimental batches. The RxRx3-core subset (Kraus et al., 2025) is a publicly available benchmark that defines gene-gene and gene-compound retrieval tasks, in which representations are evaluated based on their ability to retrieve matching perturbations in an unsupervised setting, despite substantial biological and experimental variability. The initial subset from (Kraus et al., 2025) contains 222,601 microscopy images spanning 736 CRISPR knockouts and 1,674 compounds at 8 concentrations. Yet, the dataset exhibits substantial variability in the number of image replicates per gene (ranging from $\sim$50 to $\sim$9,450), along with opportunities for improved quality control (Fig. 10). Therefore, we adopt a subsampled three-fold version of the dataset, using 10 images per gene.

**JUMP-MoA reference plates.** JUMP-MoA reference dataset, used in the SPACe evaluation framework (Stossi et al., 2024), is a compact Cell Painting set designed to evaluate whether morphological profiles can retrieve both replicates of the same compound and compounds sharing an annotated mechanism of action. The original JUMP-MoA plate map contains 90 compounds, each plated in quadruplicate on a 384-well plate, spanning 47 annotated MoA classes, with compounds profiled at a recommended concentration of 3 $\mu$M. MoA annotations were derived from the Broad Drug Repurposing Hub.

JCP2022_085227  JCP2022_037716  JCP2022_025848  JCP2022_046054  JCP2022_035095  JCP2022_064022  JCP2022_050797  JCP2022_012818

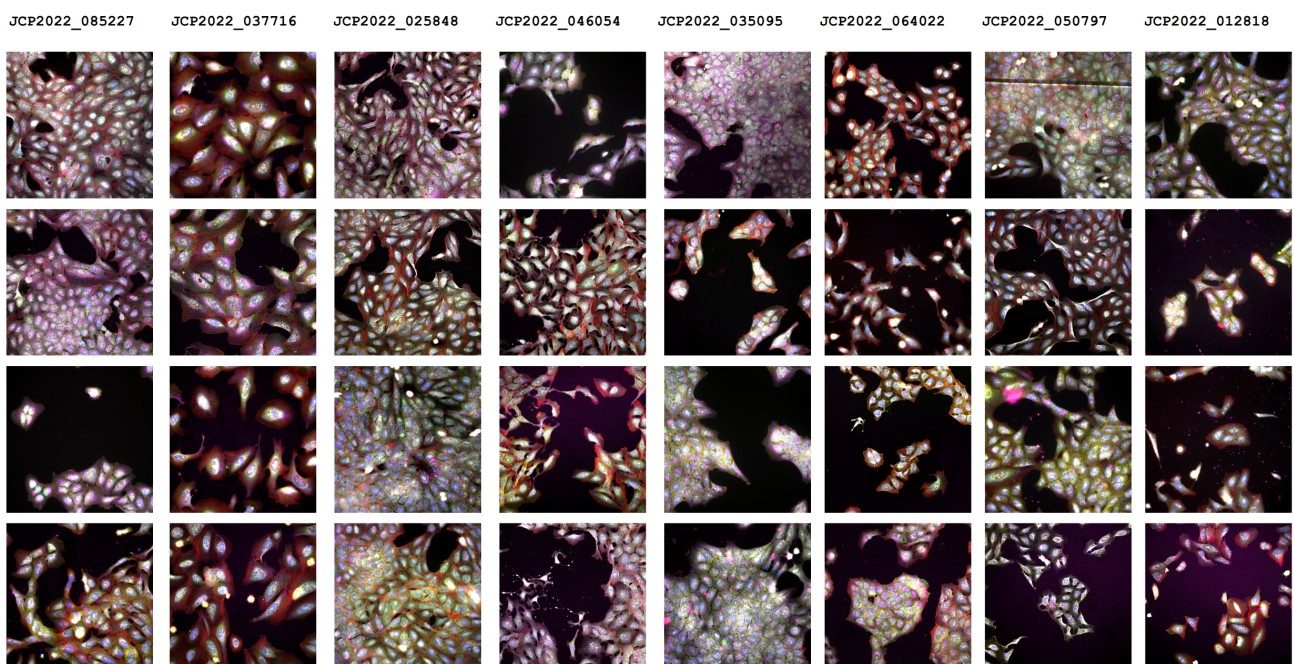

*Figure 9.* **Example of images from the 8 positive controls of JUMP-CP.**

In our analysis, we retained the subset of MoAs represented by exactly two compounds, resulting in 43 MoA classes used for the percent matching evaluation.

The image data correspond to seven JUMP-MoA reference plates, BR00115125–BR00115131, downloaded from the Cell Painting Gallery. These plates were generated using the standard JUMP Cell Painting assay in U-2 OS cells. Each well contains multiple fields of view.

### A.3. Tissue Imaging and Spatial Transcriptomics

**Whole Slide Images (WSI)** are microscopy images of thin slices of tissue. To facilitate analysis of tissue morphology the slides are often stained with hematoxylin and eosin (H&E staining). Hematoxylin reveals cell nuclei in a darker purple-blue color against the cytoplasm colored pink by eosin. Typically WSIs are large in size and resolution making them prohibitively expensive to process as single images. Thus, dividing them into patches possibly at different scales becomes necessary (Fig. 12a). Different scales exhibit varying levels of feature hierarchy. For instance in context of cancer detection and subtyping high resolution local patches can reveal morphological differences from the normal cells, while more global views allow for instance to assess the extent of tumor proliferation in affected tissues.

**HEST-1k** is a dataset of 1,229 spatial transcriptomic samples as introduced in (Jaume et al., 2024) covering 26 organs from two different species. The full dataset consists of 2.1 million image patches matched to gene expression profiles from ST spots. Full experimental details can be found in (Jaume et al., 2024). We focus only on a subset of data. Namely, a subset of H&E-stained slides is used for the *HEST-1k benchmark*. All image samples consist of WSI crops at 112×112 $\mu$m which corresponds to 20× magnification at 224×224 pixel resolution. The estimated pixel size for standardizing resolution between slides is provided with the dataset, allowing us to scale the relative coordinates of the detected cells. For each image as set of 50 genes with the highest normalized variance is provided. The benchmark implements an evaluation pipeline consisting of PCA-reduction of pre-extracted patch embeddings resulting into 256-dimensional vectors which are then used by a trainable linear ridge regression head on the 50 gene expression targets. The gene-wise Pearson correlation coefficients (PCC) are computed for each cross validation fold and are averaged across each subset of the benchmark before being averaged into a global score. We follow the evaluation protocol provided by the authors via https://github.com/mahmoodlab/HEST.

We detail the subsets and the number of corresponding samples in table 3.

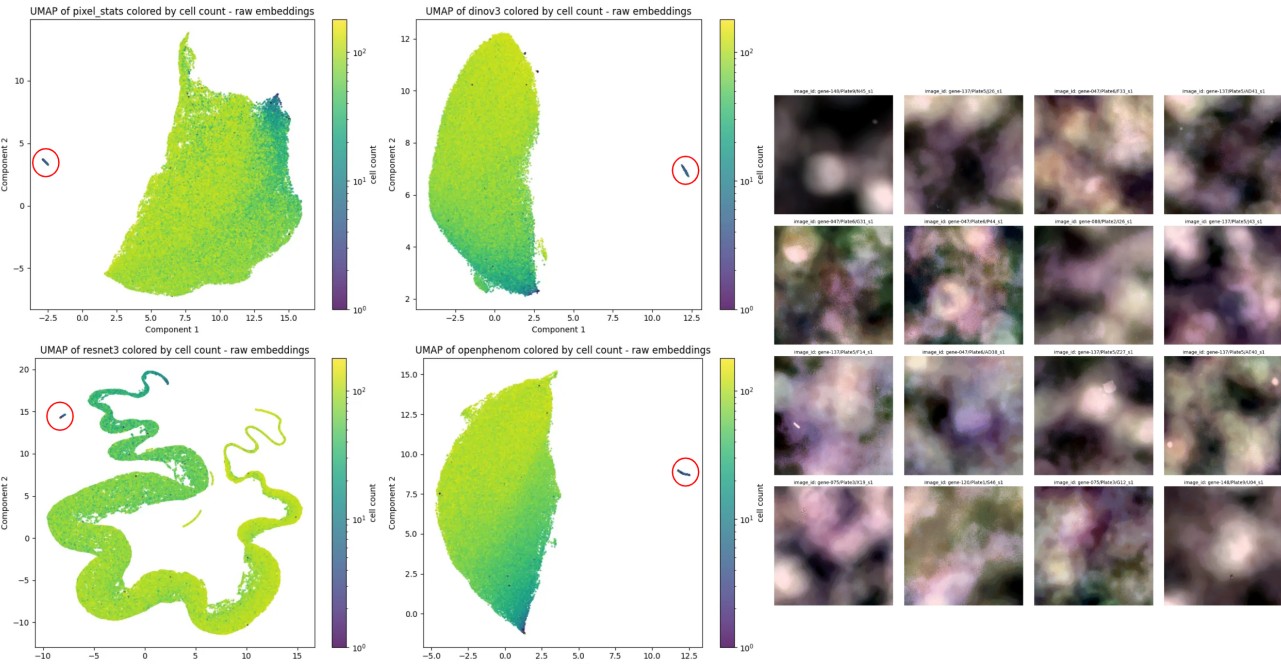

*Figure 10.* **Outlier image detection on RxRx3-Core using raw image representations. Left:** UMAP projections of embeddings produced by different encoders colored by cell count, highlighting an outlier cluster in red. **Right:** Representative images sampled from the identified outlier cluster.

**HEST-1k-1NN** is a coarsened version of HEST-1k benchmark dataset obtained by binning of adjacent spatial transcriptomics spot. More precisely, for each patch from the original dataset we extract the absolute coordinates of the corresponding ST spot which can be given by the coordinates of the corner of the patch or by its centroid. The coordinates of the spots form a square or hexagonal grid depending on the benchmark and for each patch we aggregate up to 9 and 7 total spots respectively. Incomplete neighborhoods at the edges of a slide are not discarded, instead we drop empty patches with no detected cells. The neighbor count distribution is given in Fig. 11. The number of samples in the coarsened dataset has the same order of magnitude as the original one.

Following the coarsening of the ST grid, for each aggregated neighborhood of a spot a bounding box of all constituent image patches is computed. Then, the original WSI is cropped at the location given by the bounding box accounting for the adjustments in resolution as provided in the benchmark metadata.

This strategy yields a dataset of aggregated overlapping image patches. Both the source and the resulting aggregated patches overlap, with the former being caused by the patches being larger than the physical ST spots. Importantly, we do not introduce any patch-based splits. The absence of data leakage is ensured by following the patient-stratified train-test splits as described by the authors of the benchmark (Jaume et al., 2024).

Finally, the target counts are averaged across patches additionally providing a type of smoothing over noisy ST readouts.

**BRACS-RoI** is a dataset of regions of interest (RoI) of H&E-stained human breast tissue (Brancati et al., 2022). Each RoI is labeled according to one of 7 classes, which include normal tissue and 6 pathological conditions of varying severity. RoI exhibit various spatial dimensions and are scanned at 0,25 $\mu$m/px. The complexity of data and the variation of RoI dimensions pose additional challenge for learning algorithms. First, the pathological conditions often exhibit subtle differences and accurate subtyping is challenging even for expert pathologists (Pati et al., 2022). Second the learning algorithm has to be adapted to images with pixel counts ranging varying across several orders of magnitude. Finally, average RoI pixel count differs greatly not only between samples but also between classes, leading to a potential source of shortcuts (Pati et al., 2022). In our experiments we refer to (Öğüt et al., 2026) for the train / test splits and the evaluation setup. Sample counts per class are provided in 15.

*Table 3.* Benchmark subsets of HEST-1k as defined in (Jaume et al., 2024). We keep the original structure of the benchmark unchanged.

| Task | Number of slides | Number of patients (splits) | Condition |
|------|------------------|------------------------------|-----------|
| IDC | 4 | 4 | Invasive ductal carcinoma (breast) |
| PRAD | 23 | 2 | Prostate adenocarcinoma |
| PAAD | 2 | 2 | Pancreatic adenocarcinoma |
| SCKM | 2 | 2 | skin cutaneous melanoma |
| COAD | 4 | 2 | colon adenocarcinoma |
| READ | 4 | 2 | rectal adenocarcinoma |
| ccRCC | 24 | 24 (6) | clear cell renal carcinoma |
| LUAD/LUNG | 2 | 2 | lung adenocarcinoma |
| LYMPH IDC | 4 | 2 | see IDC |

**BACH** is dataset of H&E-stained human brest tissue RoI (Aresta et al., 2019). Each RoI has a fixed size of 2048 × 1536 pixels at 0,42$\mu$m/px. Each RoI is labeled according to one of 4 classes representing normal and pathological conditions. Sample counts per class are provided in 15.

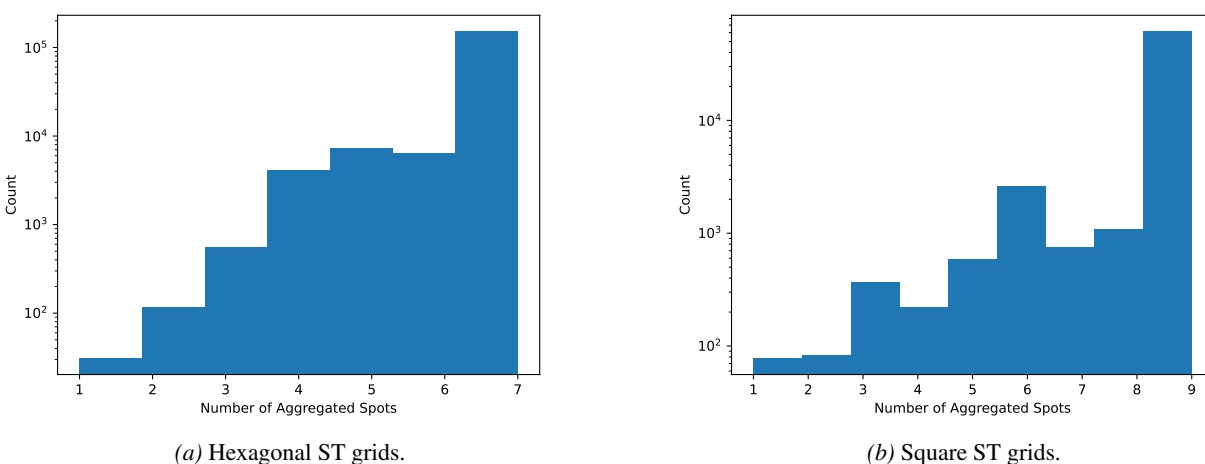

*(a)* Hexagonal ST grids.                     *(b)* Square ST grids.

*Figure 11.* We display neighbor counts for slides with different ST grids. The counts are given in the logarithmic scale.

## B. Baseline Models

### B.1. Expert Hand-Crafted Features

**CellProfiler** (Stirling et al., 2021) is an open-source image analysis software designed for reproducible, high-throughput quantification of biological microscopy images. It enables users to build modular pipelines for common image-processing tasks, including illumination correction, segmentation, object detection, feature extraction, and per-cell or per-image measurement. In Cell Painting experiments, CellProfiler is commonly used to extract handcrafted morphological features describing cell shape, texture, intensity, and spatial organization across multiple fluorescence channels. We used CellProfiler version 4.2.8. and followed, for cell painting based feature extractions, the pipeline from Arevalo et al., (2024).

### B.2. Out-of-Domain Models

**AlexNet** is a convolutional neural network architecture originally developed for large-scale image classification. We evaluate the standard AlexNet architecture using both ImageNet-pretrained weights and randomly initialized (untrained) weights, as provided by the `torchvision` library (Wightman, 2019).

*Table 4.* Configurations of ResNet (He et al., 2016) and ViT (Dosovitskiy et al., 2021) used in our experiments.

| Architecture | `timm` Model Name | `timm` Weights | Model Stages | Aggregation |
|---|---|---|---|---|
| *Convolutional Neural Networks (CNNs)* | | | | |
| ResNet18 | `resnet18` | `tv_in1k` | layer $\in \{1, 2, 3, 4\}$ | Avg-pool |
| ResNet34 | `resnet34` | `tv_in1k` | layer $\in \{1, 2, 3, 4\}$ | Avg-pool |
| ResNet50 | `resnet50` | `tv2_in1k` | layer $\in \{1, 2, 3, 4\}$ | Avg-pool |
| ResNet152 | `resnet152` | `tv2_in1k` | layer $\in \{1, 2, 3, 4\}$ | Avg-pool |
| *Vision Transformers (ViTs)* | | | | |
| ViT-s/16 | `vit_small_patch16_224` | `augreg_in21k_ft_in1k` | block $\in \{3, 6, 9, 12\}$ | `<cls>`-token |
| ViT-b/16 | `vit_base_patch16_224` | `augreg2_in21k_ft_in1k` | block $\in \{3, 6, 9, 12\}$ | `<cls>`-token |
| ViT-l/16 | `vit_large_patch16_224` | `augreg_in21k_ft_in1k` | block $\in \{6, 12, 18, 24\}$ | `<cls>`-token |
| ViT-h/14 | `vit_huge_patch14_224` | `orig_in21k` | block $\in \{8, 16, 24, 32\}$ | `<cls>`-token |

**ResNets and Vision Transformers** are listed in table 4. All models and intermediate stages are is integrated into the HEST-1k benchmarking suite and evaluated using the provided protocol.

**DINOv2 & DINOv3** are self-supervised vision transformers trained using teacher–student distillation, with DINOv3 further incorporates larger-scale training. We evaluated the `dinov2_vitg14` and `dinov3_vit7b16` architectures using pretrained weights released through the official `facebookresearch` DINOv2 and DINOv3 repositories

## B.3. In-Domain Models

**OpenPhenom** is a masked auto-encoder foundation model based on a ViT-s/16 architecture, pretrained directly on large-scale cell culture microscopy data. We evaluate OpenPhenom using the publicly released pretrained weights available from the Hugging Face repository `recursionpharma/OpenPhenom`. We discard the `<cls>`-tokens and use the average of patch tokens following the implementation from `https://huggingface.co/recursionpharma/OpenPhenom`. For stage-wise experiments we use blocks $\{3, 6, 9, 12\}$. The model is integrated into the HEST-1k benchmarking suite and evaluated using the provided protocol.

**UNI / UNI 2** as introduced in (Chen et al., 2024) are histology (designated for tissue analysis) foundation models based on ViT-l/14/ ViT-h/14-reg8 architectures respectively. The models were trained on patches from 100,000 / 350,000 histology slides using DINOv2 (Oquab et al., 2024). The models are evaluated using the HEST-1k benchmarking suite.

**CONCH (v1, v1.5)** (Lu et al., 2024) are ViT-b and ViT-l based histology foundation models, finetuned from UNI checkpoints with iBOT (Zhou et al., 2022) on 1.17 million of histology image/ caption pairs. The models are evaluated using the HEST-1k benchmarking suite.

**Kaiko Base 8** as introduced in (kaiko. ai et al., 2024) is a histology foundation model based on the ViT-b/8 architecture, trained with DINO (Caron et al., 2021). The model is evaluated using the HEST-1k benchmarking suite.

**GigaPath** as introduced in (Xu et al., 2024) is a histology foundation model based on the ViT-g architecture, trained with DINOv2 on patches from 171,189 WSIs. The model is evaluated using the HEST-1k benchmarking suite.

**Hibou Large** as introduced in (Nechaev et al., 2024) is a histology foundation model based on the ViT-l architecture, trained with DINOv2 on patches from 1,138,905 WSIs. The model is evaluated using the HEST-1k benchmarking suite.

**Phikon (v1 / v2)** as introduced in (Filiot et al., 2023) / (Filiot et al., 2024) are histology foundation models based on ViT-b / ViT-l trained with iBOT / DINOv2 on patches from 6,093 / 60,000 WSIs. The model is evaluated using the HEST-1k benchmarking suite.

**CTransPath** is a histology foundation model based on a Swin Transformer (Swin-T/14) (Liu et al., 2021) architecture and trained using MoCoV3 on patches from 32,220 WSIs. The model is evaluated using the HEST-1k benchmarking suite.

**Virchow / Virchow2**  as introduced in (Vorontsov et al., 2024) and (Zimmermann et al., 2024) respectively are histology foundation models based on the ViT-h architecture trained on patches from 1.5 million / 3.1 million WSIs using DINOv2. The model is evaluated using the HEST-1k benchmarking suite.

**H-Optimus-0**  as introduced in (Saillard et al., 2024) is a histology foundation model trained with DINOv2 on patches from 500,000 WSIs. The architecture is based on a 40-block ViT-g/14. The model is evaluated using the HEST-1k benchmark. In our stage-wise experiments we use `<cls>`-tokens of blocks $\{10, 20, 30, 40\}$.

**H-Optimus-1**  is a histology foundation model trained with DINOv2 on patches from more than 1 million slides (Bioptimus, 2025). The architecture is based on a 40-block ViT-g/14. The model is evaluated using the HEST-1k benchmarking suite. In our stage-wise experiments we use `<cls>`-tokens of blocks $\{10, 20, 30, 40\}$.

### B.4. Graph Neural Networks

In absence of commonly used foundation models for *structure-only* cell graphs and point sets we restrict ourselves to simple GNN baselines, demonstrating their potential in controlled settings, and leaving architectural exploration for future work.

**Graph Convolutional Network (GCN)**  is an architecture introduced in (Kipf & Welling, 2017) and defined by a specific aggregation of graph neighborhoods, which was inspired by a first-order approximation of graph spectral convolutions. It can be viewed as an instance of message passing and extension of CNNs to irregular grids (Gilmer et al., 2017; Bronstein et al., 2021).

We define our GCN-based model as a 4-layer convolutional network operating on node embeddings given by a learned linear projection of normalized coordinates of points. Notably, this approach is not invariant with respect to E($n$), Euclidean group in $\mathbb{R}^n$ (i.e. it is not symmetric to transformations that preserve euclidean distance such as rotations, translations, and reflections).

For experiments on synthetic data node embeddings of each convolutional layer are concatenated and followed by a global max-pooling of node embeddings with and a 2-layer MLP classifier. To perform feature extraction we remove the last linear layer of the MLP.

**E($n$)-Equivariant Neural Network (EGNN).**  To overcome the limitations of GCN we also explore E(2)-equivariant networks (and thus, allowing us to get E(2)-invariant embeddings for inputs defined by sets of Cartesian coordinates in $\mathbb{R}^2$). Introduced in (Satorras et al., 2021), they offer a simple recipe to enforce symmetry with respect to the E($n$) groups. The key idea rests on defining messages as functions of pairwise distances between node positions (which are preserved under E($n$) transformations) and E($n$)-invariant node features.

We define node features using local degree profiles as described in (Cai & Wang, 2022) which aggregate statistics of the neighbor degrees. Since the adjacency matrix of our graph is given by the Euclidean distances between the centroids of cell nuclei, these feature vectors remain invariant under rotations, translations, and reflections. The network is constructed of 3-layers of EGNN convolutional layers.

For experiments on synthetic data node embeddings of the last convolutional layer are mean-pooled and passed through a 2-layer MLP classifier. To perform feature extraction we remove the last linear layer of the MLP.

### B.5. Adaptation to Multichannel Images

Cell Painting images comprise five channels in JUMP-CP and six channels in RxRx3-core. As most vision backbones are pretrained on RGB images, their input projection layers must be adapted to accept $n \in \{5, 6\}$ input channels.

For ViT-based encoders, including DINO and H-Optimus, we expand the pretrained patch embedding weights to the required number of input channels using a custom channel-expansion scheme that preserves and copies the original filters and the bias when present, while for ResNet backbones we adapt the first convolutional layer using the `timm.create_model` `in_chans` argument, which additionally rescales the expanded weights to preserve activation magnitudes.

# C. Structure-Based Tissue Representations

## C.1. Cell Count

Benchmark scores on HEST-1k are evaluated on predictions of a single linear ridge regression layer. Since embeddings of structure-based models already contain some non-linear transformation of cell count, we manually create vectors of 16 hand-crafted features based on the initial cell count value using polynomial, trigonometric, and logarithmic functions of the cell count per patch. To avoid issues during PCA computation we tile the embeddings to the target size 256 and add a small amount ($\sigma = 0.01$) of Gaussian noise to the final embeddings. We evaluate standardized and raw cell count embeddings and select the best performing configuration. Another option provided by authors is using XGBoost (Chen & Guestrin, 2016) for regression. We provide additional results for XGBoost in table 7, which yields a slight improvement overall. The results reported in the main text we follow the PCA+ridge evaluation for consistency with other models. However, for each dataset we report the results for the best combination of hand-crafted cell-counting features, giving this baseline a slight advantage.

## C.2. Construction of Cell Graphs and Cell Point Sets

Cell point sets are defined by coordinates of nuclei centroids. We reuse segmentation results obtained with CellViT (Hörst et al., 2024) as provided with HEST-1k benchmark. The absolute coordinates of cell nuclei are linked to absolute positions of the corresponding spatial transcriptomics spots.

For the non-binned version we estimate the size of each image patch in the coordinates of the source WSI using the magnification factors and estimated pixel sizes as provided with HEST-1k. The cells are binned according to the estimated patches.

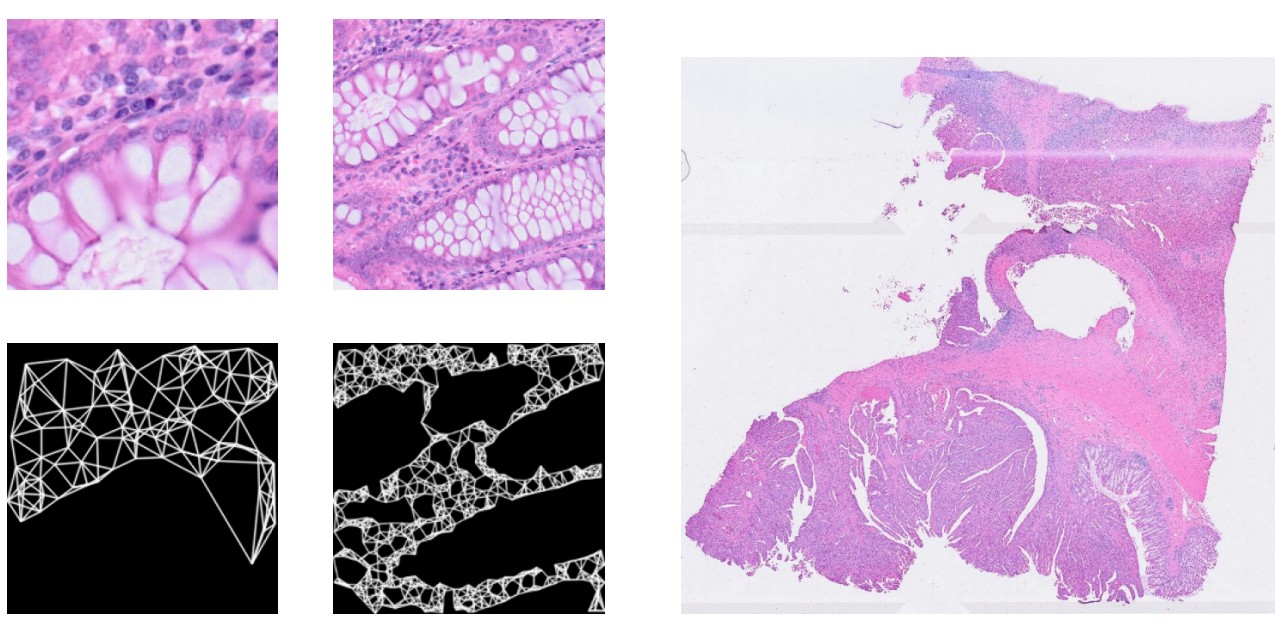

*(a)* 1x1 patches (left), 1NN patches (right).        *(b)* Slide: `TENX147`, dataset: COAD.

*Figure 12.* **Multiscale views in WSIs.** (a) Comparison of the original patches from HEST-1k and HEST-1k-1NN and the corresponding cell graphs. (b) The source WSI from which the samples are taken. The slide overlays a square grid of ST spots and the aggregated patch corresponds to a 9-spot neighborhood.

For the 1NN version, the bounding boxes of aggregated *image patches* are computed as described in A. The bounding boxes are then used to bin cell coordinates into aggregated patches. Coordinates of cell patches are rescaled to [0, 1] using the largest dimension of bounding boxes. That is for each group of cells the most distant cells from the centroid does not necessarily reach the boundary of the resulting [0, 1] × [0, 1] square. The final coordinates are centered and rescaled to approximately [-1, 1].

The binarized images of cell graphs are plots of 5-nearest-neighbor graphs. The number of neighbors is chosen based on a trade-off between reflecting variations in cellular density and following outlines of large cellular formations as shown in Fig. 12a. The graphs are rendered at the pixel count given by the dimensions of bounding boxes of patches at the resolution of the corresponding source WSIs scaled by the estimated pixel size per $\mu$m (provided with HEST-1k). This approach standardizes patch dimensions, approximately linking it to a physical ST spot across datasets imaged at different resolutions. The rendered graphs are then resized to 224×224 pixels using linear interpolation.

### C.3. Synthetic Data

To explore effectiveness of geometric modalities for capturing relevant biological signal we design a synthetic dataset of points on the unit square. Notably we want to explore the following questions while controlling for the cell count.

First, we are interested in detections of local variations of density. Such variations are intended to represent cell clumping or large structural elements of cellular tissue e.g., ducts in gland samples. We model cell graphs using an inhomogeneous Poisson process with spatial intensity $\lambda(x, y)$ for $(x, y) \in [0, 1] \times [0, 1]$.

Second, we want to verify separability of samples defined on different grids. Drawing inspiration from works in cancer subtyping (Wang et al., 2023), which demonstrate how regularity of cell organization can help discriminate between specific pathological conditions, we additionally simulate cell graphs as adaptive square grids with the resolution of the grid given by local cell density. Cell grids are then perturbed with gaussian noise to create the final cell positions. The variance of added gaussian noise is proportional to some $\sigma(x, y)$ for $(x, y) \in [0, 1] \times [0, 1]$. While real cellular tissue is not organized in regular square grids, we are interested in the model's ability to detect localized disruptions of regular patterns.

We select density and noise patterns of interest and unify them into the following landscapes:

$$\text{SlopeLandscape}(x, y; k, b) = 1 + \max(b - kx, 0)$$
$$\text{StepLandscape}(x, y; a_x, \Delta) = 1 + \Delta \cdot \mathbf{1}_{x < a_x}.$$

We define $\text{Discs}(x, y; n, r, \text{emboss}, \Delta)$ as

$$1 + \Delta \cdot \min\left(\sum_i^n \mathbf{1}_{(x,y) \in \text{disc}_i}, 1\right).$$

Then $\text{Discs}(x, y; n, r, \text{deboss})$ is given by

$$1 - \min\left(\sum_i^n \mathbf{1}_{(x,y) \in \text{disc}_i}, 1\right).$$

The discs $\text{disc}_i$, $i = 1, \ldots, n$ have a shared radius $r$ and are uniformly spaced at a distance of $0.25$ from the center of unit square if $n > 1$ and randomly positioned within $[r, 1 - r] \times [r, 1 - r]$ square.

To address the first question we introduce varying numbers of discs as well as a single disc setup with the location of the disc being randomly sampled. The area of the discs is adjusted to either preserve the area of a larger reference disc or not. For the second question we use the slope and step functions as scalers for density and the amount of added noise. This leads us to the classification problem as summarized in table 5. For each class we generate a train/validation/test split of size 1000/100/1000 samples respectively. Finally we randomly sample 90-degree rotations and mirror flips for images and unconstrained random rotations for point sets both during training as data augmentations and during evaluation. From the definition of classes it is clear that some may overlap or not manifest obvious differences under uniform and noisy grid-based sampling.

A visualization of a subset of noise/ density landscapes is given in Fig. 13a. We provide binary images of graphs for each class in Fig. 13b.

We conduct three classification experiments. To evaluate the capacity of vision encoders to classify binary graphs we fully train a ResNet18 model from scratch. To evaluate capabilities of pretrained vision encoders on this task we train a linear classifier on top of a frozen ViT-l/16 pretrained on natural images. Finally, we train a GNN architecture on $k$NN graphs constructed from node positions. Table 6 summarizes our results.

While none of the models achieve a perfect score they demonstrate an above random performance and allow us to conclude that cell position sets as a modality opens a promising direction for studying tissue organization beyond cell count.

*Table 5.* A Summary of Synthetic Classes.

| Class | Sampling | Noise and/ or density | Cell count |
|---|---|---|---|
| 0 | Noisy Grid, 10 voxels | $\sigma(x, y) \propto \text{Id}$ | 900 |
| 1 | Noisy Grid, 10 voxels | $\sigma(x, y) \propto \text{SlopeLandscape}(x, y; 3, 3)$ | 900 |
| 2 | Noisy Grid, 10 voxels | $\lambda(x, y) \propto \text{SlopeLandscape}(x, y; 3, 3), \sigma(x, y) \equiv 0.01$ | 890 |
| 3 | Noisy Grid, 4 voxels | $\lambda(x, y) \propto \text{StepLandscape}(x, y; 0.5, 1), \sigma(x, y) \equiv 0.01$ | 848 |
| 4 | Noisy Grid, 10 voxels | $\lambda(x, y) \propto \text{Discs}(x, y; 1, 0.1, \text{emboss}, 2), \sigma(x, y) \equiv 0.01$ | 964 |
| 5 | Noisy Grid, 10 voxels | $\lambda(x, y) \propto \text{Discs}(x, y; 3, 0.1, \text{emboss}, 2), \sigma(x, y) \equiv 0.01$ | 1028 |
| 6 | Noisy Grid, 10 voxels | $\lambda(x, y) \propto \text{Discs}(x, y; 1, 0.1, \text{deboss}), \sigma(x, y) \equiv 0.01$ | 864 |
| 7 | Noisy Grid, 10 voxels | $\lambda(x, y) \propto \text{Discs}(x, y; 3, 0.1, \text{deboss}), \sigma(x, y) \equiv 0.01$ | 819 |
| 8 | Uniform | $\lambda(x, y) \propto \text{Discs}(x, y; 3, 0.1, \text{emboss}), 2$ | Pois(900) |
| 9 | Uniform | $\lambda(x, y) \propto \text{Discs}(x, y; 1, 0.1, \text{deboss})$ | Pois(900) |
| 10 | Uniform | $\lambda(x, y) \propto \text{Id}$ | Pois(900) |
| 11 | Uniform | $\lambda(x, y) \propto \text{Discs}(x, y; 1, 0.1, \text{emboss}, 2)$ | Pois(900) |
| 12 | Uniform | $\lambda(x, y) \propto \text{Discs}(x, y; 1, 0.1, \text{deboss}, 1)$ | Pois(900) |
| 13 | Noisy Grid, 10 voxels | $\sigma(x, y) \propto \text{StepLandscape}(x, y; 0.5, 1)$ | 900 |
| 14 | Uniform | $\lambda(x, y) \propto \text{Discs}(x, y; 3, \sqrt{0.2^2/3}, \text{emboss}, 2)$ | Pois(900) |
| 15 | Uniform | $\lambda(x, y) \propto \text{Discs}(x, y; 3, \sqrt{0.2^2/3}, \text{deboss})$ | Pois(900) |
| 16 | Uniform | $\lambda(x, y) \propto \text{Discs}(x, y; 5, \sqrt{0.2^2/5}, \text{emboss}, 2)$ | Pois(900) |
| 17 | Uniform | $\lambda(x, y) \propto \text{Discs}(x, y; 5, \sqrt{0.2^2/5}, \text{deboss})$ | Pois(900) |
| 18 | Uniform | $\lambda(x, y) \propto \text{SlopeLandscape}(x, y; 3, 3)$ | Pois(900) |
| 19 | Uniform | $\lambda(x, y) \propto \text{SlopeLandscape}(x, y; 2, 2)$ | Pois(900) |
| 20 | Uniform | $\lambda(x, y) \propto \text{SlopeLandscape}(x, y; 1, 1)$ | Pois(900) |
| 21 | Uniform | $\lambda(x, y) \propto \text{Discs}(x, y; 1, 0.2, \text{emboss}, 2)$ | Pois(900) |
| 22 | Uniform | $\lambda(x, y) \propto \text{Discs}(x, y; 1, 0.2, \text{deboss})$ | Pois(900) |
| 23 | Uniform | $\lambda(x, y) \propto \text{StepLandscape}(x, y; 0.5, 1)$ | Pois(900) |

*Table 6.* Test scores on our 24-class synthetic dataset. Classes with F1-scores above the threshold of 0.98 are provided.

| Model | Average F1 Score | Worst Class (F1-score) | Best Class (F1-score) | Invariance |
|---|---|---|---|---|
| ResNet18 (from scratch) | 0.87 | class 18 (0.38) | classes 1,3,5,7,8,9,14,15,17,19,21 ($\geq$**0.99**) | Learned, data augmentations |
| ViT-l/16 (linear probing) | 0.68 | class 22 (0.15) | classes 7,9,12,15,17,20,21 ($\geq$ **0.98**) | Learned, data augmentations |
| GCN (from scratch) | 0.72 | class 4 (0.15) | classes 7,8,9,12,15,17,21 ($\geq$**0.98**) | Learned, data augmentations |
| E2GNN (from scratch) | 0.73 | class 4 (0.04) | classes 1,5,12,17,18,19 (**=1.0**) | Architectural, E(2) |

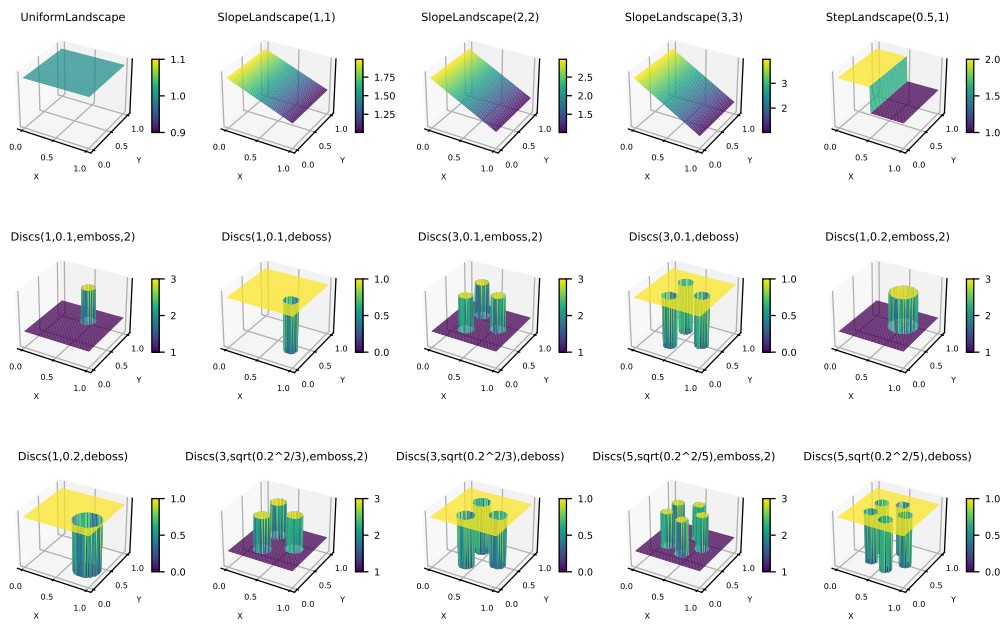

*(a)* Examples of intensity functions.

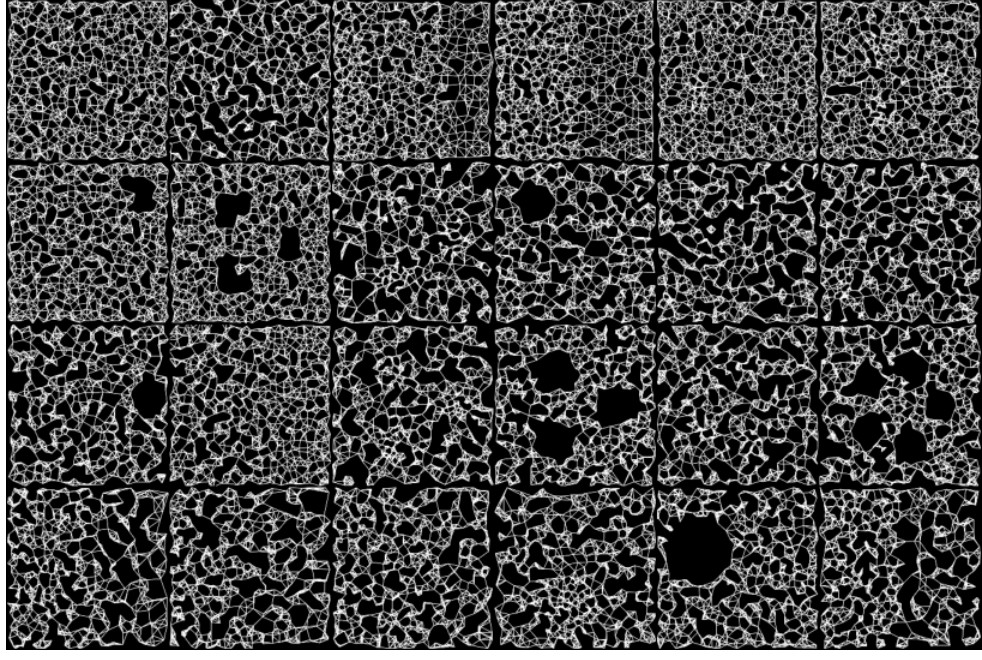

*(b)* Test samples from each of 24 synthetic classes, rendered at 224×224 pixels.

*Figure 13.* **Illustration of synthetic graph data.** Samples are designed to represent variations in local density while controlling for the node count. (a) Selected intensity functions that are used to scale density $\lambda$ and spatial noise $\sigma$ as listed in table 5. (b) Test samples for each of 24 classes.

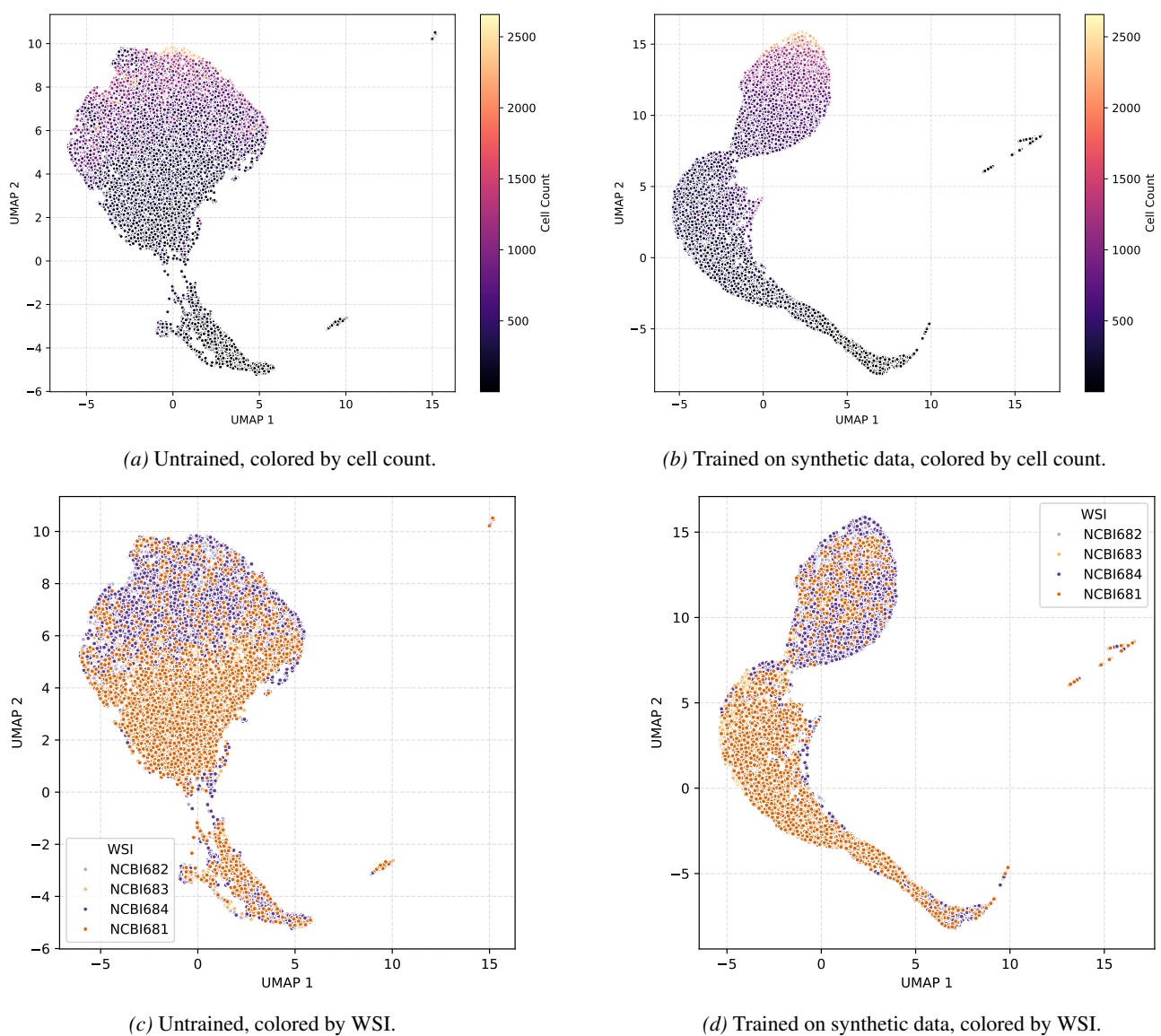

*(a)* Untrained, colored by cell count.

*(b)* Trained on synthetic data, colored by cell count.

*(c)* Untrained, colored by WSI.

*(d)* Trained on synthetic data, colored by WSI.

*Figure 14.* **Cell count and slide separation in GNNs.** UMAP plots of patch-level embeddings extracted with an untrained and trained GCN on LYMPH IDC-1NN.

# D. Comparison and Additional Experimental Results for HEST-1k and HEST-1k-1NN

## D.1. Comparison of HEST-1k and HEST-1k-1NN

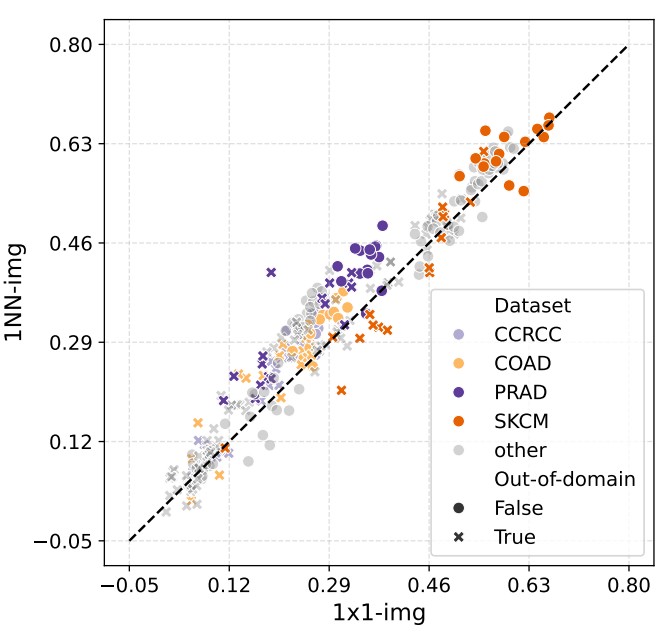

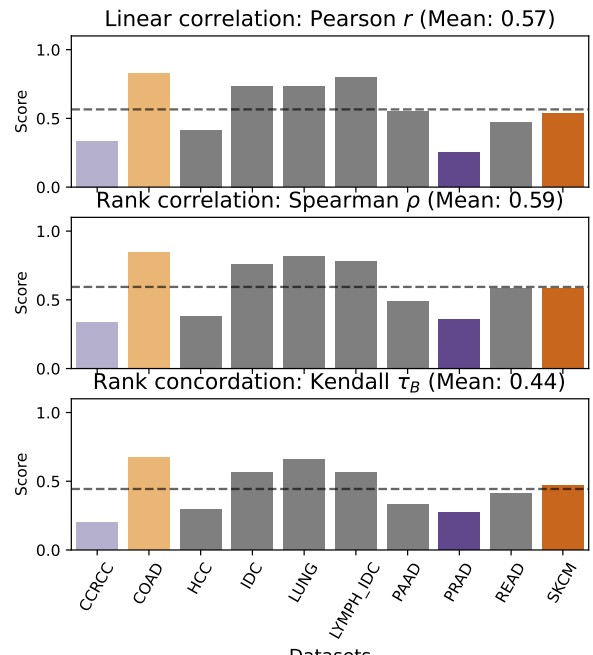

*(a)* Alignment between the performance of models on HEST-1k and HEST-1k-1NN. Each point represents dataset-wise PCC for a given model configuration. The plot includes in domain and out of domain models.

*(b)* Correlations between the performance of models on HEST-1k and HEST-1k-1NN. The correlations are computed for *in domain foundation models* only

*Figure 15.* Comparison between HEST-1k and HEST-1k-1NN. (a) Alignment of average PCC for a given dataset-model configuration pair. (b) Correlations between the performance of histology foundation models.

To better understand the new level of granularity introduced with the construction of HEST-1k-1NN we provide a side-by-side comparison of source and binned patches in Fig. 12. The larger binned patches help better recognize larger structural elements in tissues (e.g., ducts in glands), while sacrificing fine resolution of cellular morphology. While this can be expected to favor strong structure-based models, the impact of the new task on pretrained foundation models is not immediately clear. Especially since both global tumoral organization and single-cell morphology can hint at increased expression of some marker genes.

We re-establish reference performance of the selected models on the two versions of the benchmark. Fig. 15 suggests a strong overall alignment between all (IID, OOD, and untrained) models. Furthermore, the new task appears to be generally easier which can be partially attributed to smoother targets. However, a closer evaluation of the foundation models specifically reveals that the final model rankings do not share the same order, as evidenced by correlation and concordance coefficients in Fig. 15b. This supports our original claim that certain dataset-specific targets are substantially easier to predict with imaging and structural modalities across all models even including naive untrained baselines. A direct inspection of metrics in table 8 points out several datasets that preserve top-1 and top-2 rankings. For instance H-Optimus-1 successfully outperforms its competitors on SCKM under both levels of granularity. Similarly, CONCH v1.5 is a strong performer on PRAD, while H-Optimus-0 reaches the highest scores on the challenging CCRCC dataset.

## D.2. Structural Baselines Applied to the Non-binned HEST-1k

As shown in Fig. 16 the original non-binned dataset already contains a large subset of targets that can be explained by structure-only features. Notably the PCC scores for a large cluster of genes predicted with the structure-only ResNet152 aligns well with scores achieved by evaluating an IID (Fig. 16a) and OOD (Fig, 16b) baselines. This observation not only suggests that evaluation of OOD baselines alone is insufficient to properly contextualize the performance of IID models but

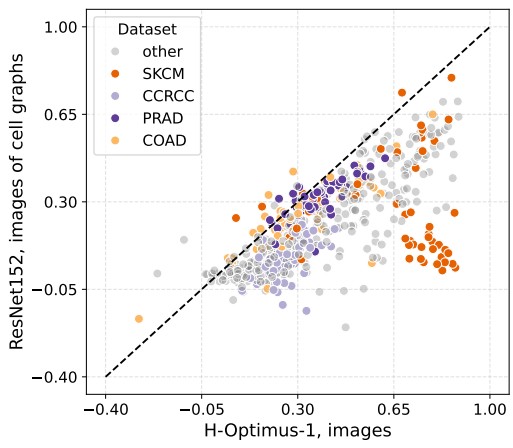

*(a)* Cell graphs and an in-domain foundation model.

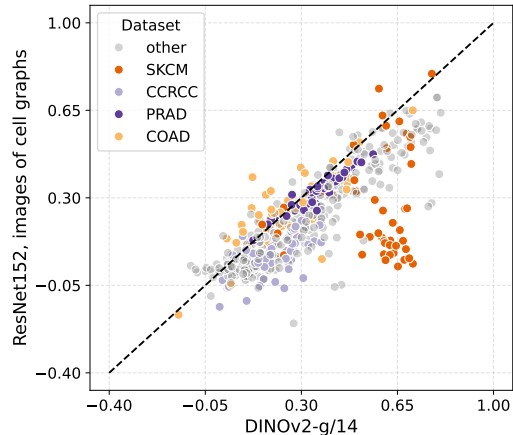

*(b)* Cell graphs and an out-of-domain vision backbone.

*Figure 16.* **Experiments on the non-binned HEST-1k.** Structure-based models remain competitive on a subset of genes across all ranges of PCC scores. (a) Gene-wise PCC for images of cell graphs and an in-domain foundation model H-optimus-1. (b) Gene-wise PCC for images of cell graphs and an out-of-domain DINOv2-g/14.

also helps formulate hypotheses about the nature of the predictive signal captured by different pretrained encoders. For instance, when comparing DINOv2-g/14 and H-Optimus-1 on a subset of SKCM one can choose to focus specifically on the genes that are not well predicted by the structure-based model, hence restricting the analysis to as subset of targets, which appears to be more affected by cellular morphology.

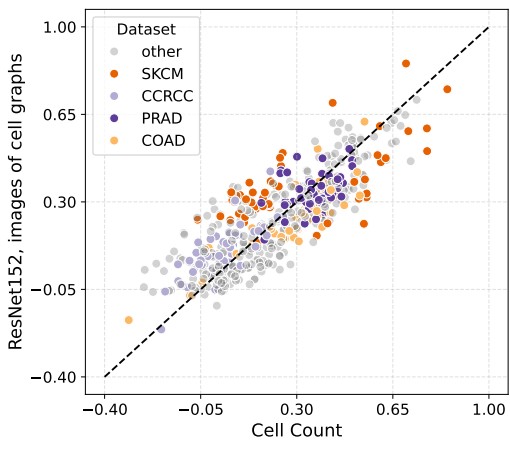

*(a)* Gene-wise PCC, HEST-1k-1NN.

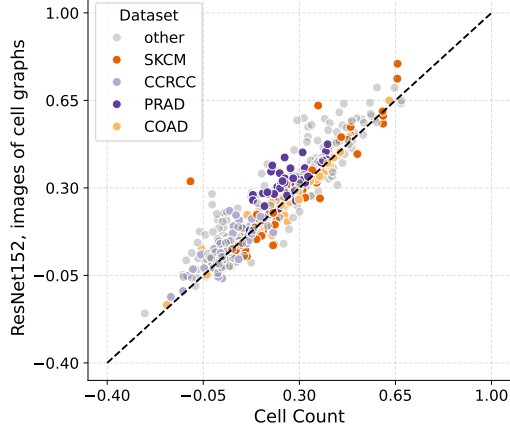

*(b)* Gene-wise PCC, HEST-1k.

*Figure 17.* **Possible contribution of cell count.** Each point represents gene-wise PCC for images of cell graphs and a non-linear cell-counting baseline. (a) Binned HEST-1k-1NN. (b) Non-binnned HEST-1k.

## D.3. The Impact of Cell Count

The performance of structure-only models exceeds random levels and even occasionally reaches pretrained image-based models for both HEST-1k-1NN and HEST-1k. For the latter, larger structural patterns are less prominent, and thus it is of immediate interest to compare the models to a simple cell counting baseline. In Fig. 17 we observe that while the individual PCC values per gene may differ considerably for binary images of graphs and cell count, the overall trend indicates a strong similarity between the two approaches. In particular for the non-binned version of the benchmark, gene-wise PCC scores on COAD are strongly correlated (Fig. 17b). This result is not surprising given that for a given patch the number of cells

may correlate with local density, regularity of the grid formed by a rendered graph, and other visual features captured by a pretrained CNN. Thus, a vision encoder, not trained to do so, might be indirectly estimating the number of cells in a patch. Removal of morphological features allows us to highlight this behavior in encoders pretrained on natural images.

While for a given slide cell count might be perfectly correlated with other spatial features, it is easy to construct counterexamples where discriminating between spatial arrangements requires more expressive approaches. We investigate such cases in C.3 and conclude that both GNNs and common vision encoder can recover features beyond cell count. Explicitly demonstrating their utility however remains a more complicated task, requiring a controlled setup that is difficult to achieve in context of heterogeneous tissue samples.

In table 7 we list scores of cell-counting baselines for all the subsets of the benchmark. Notably our evaluation of GNNs pretrained on synthetic data (controlled for the average cell count) yields inconsistent results: the GCN appears to benefit from such pretraining while, the opposite is true for the EGNN. In Fig. 14 we overlay dimensionality reduction plots with values of patch-wise cell count and indices of the corresponding WSIs. The UMAP plots display a gradient of cell count which appears to be one of the prominent axes of variation for both the untrained and the pretaned GCNs, suggesting that training on estimation of local density patterns still yields representations that correlate strongly with cell count in real tissue samples.

*Table 7.* **Pearson Correlation Coefficient (PCC) across datasets for cell-counting and GNN baselines for HEST-1k and HEST-1k-1NN**. The best and the second best score for HEST-1k-1NN are in **bold** and underlined respectively. Standard deviations across the cross-validation folds are reported.

| Modality | Training | Benchmark Model | IDC | PRAD | PAAD | SKCM | COAD | READ | CCRCC | LUNG | LYMPH IDC | Avg |
|---|---|---|---|---|---|---|---|---|---|---|---|---|
| 1x1 | Hand-crafted | Cell count (PCA+ridge) | 0.32 | 0.26 | 0.32 | 0.26 | 0.26 | 0.02 | 0.04 | 0.42 | 0.11 | 0.22 |
| | | | ± 0.04 | ± 0.02 | ± 0.04 | ± 0.05 | ± 0.004 | ± 0.001 | ± 0.07 | ± 0.003 | ± 0.05 | ± 0.13 |
| | | Cell count (XGBoost) | 0.31 | 0.26 | 0.34 | 0.30 | 0.26 | 0.05 | 0.03 | 0.44 | 0.11 | 0.23 |
| | | | ± 0.05 | ± 0.02 | ± 0.08 | ± 0.07 | ± 0.01 | ± 0.03 | ± 0.06 | ± 0.01 | ± 0.07 | ± 0.12 |
| 1NN | Hand-crafted | Cell count (PCA+ridge) | **0.38** | 0.37 | 0.30 | 0.32 | 0.30 | -0.002 | **0.11** | **0.49** | 0.21 | 0.28 |
| | | | ± 0.06 | ± 0.05 | ± 0.04 | ± 0.02 | ± 0.07 | ± 0.03 | ± 0.07 | ± 0.02 | ± 0.11 | ± 0.14 |
| | | Cell count (XGBoost) | 0.37 | 0.36 | **0.34** | 0.40 | 0.30 | 0.03 | 0.04 | **0.49** | 0.21 | 0.28 |
| | | | ± 0.08 | ± 0.05 | ± 0.07 | ± 0.06 | ± 0.07 | ± 0.02 | ± 0.09 | ± 0.02 | ± 0.16 | ± 0.15 |
| | Synthetic data | EGNN | 0.22 | 0.36 | 0.33 | 0.22 | 0.28 | 0.02 | 0.05 | 0.19 | 0.17 | 0.20 |
| | | | ± 0.07 | ± 0.09 | ± 0.09 | ± 0.06 | ± 0.12 | ± 0.06 | ± 0.068 | ± 0.09 | ± 0.09 | ± 0.12 |
| | | GCN | **0.38** | **0.38** | 0.33 | **0.45** | **0.31** | **0.06** | 0.04 | **0.49** | **0.26** | **0.30** |
| | | | ± 0.04 | ± 0.04 | ± 0.08 | ± 0.07 | ± 0.01 | ± 0.07 | ± 0.07 | ± 0.09 | ± 0.12 | ± 0.16 |
| | Untrained | EGNN | 0.35 | 0.360 | 0.32 | 0.27 | 0.30 | 0.02 | 0.05 | 0.42 | **0.26** | 0.26 |
| | | | ± 0.02 | ± 0.05 | ± 0.03 | ± 0.03 | ± 0.003 | ± 0.06 | ± 0.05 | ± 0.02 | ± 0.10 | ± 0.14 |
| | | GCN | 0.36 | 0.34 | 0.32 | 0.37 | 0.27 | 0.02 | 0.06 | 0.47 | 0.25 | 0.27 |
| | | | ± 0.04 | ± 0.04 | ± 0.03 | ± 0.08 | ± 0.01 | ± 0.04 | ± 0.05 | ± 0.001 | ± 0.10 | ± 0.15 |

# E. Detailed Model Performance on HEST-1k and HEST-1k-1NN

*Table 8.* **Pearson Correlation Coefficient (PCC) across datasets and foundation models for HEST-1k and HEST-1k-1NN**. The best and the second best score per modality are in **bold** and underlined respectively. Standard deviations across the cross-validation folds are reported.

| Modality | Training | Benchmark Model | IDC | PRAD | PAAD | SKCM | COAD | READ | CCRCC | LUNG | LYMPH IDC | Avg |
|---|---|---|---|---|---|---|---|---|---|---|---|---|
| 1x1-img | Foundation Model | CONCH v1 | 0.536 | 0.355 | 0.446 | 0.579 | 0.253 | 0.163 | 0.218 | 0.531 | 0.251 | 0.370 |
| | | | ± 0.084 | ± 0.010 | ± 0.071 | ± 0.050 | ± 0.008 | ± 0.049 | ± 0.035 | ± 0.011 | ± 0.042 | ± 0.157 |
| | | CONCH v1.5 | 0.544 | **0.381** | 0.457 | 0.554 | 0.279 | 0.159 | 0.218 | 0.550 | 0.270 | 0.379 |
| | | | ± 0.085 | ± 0.010 | ± 0.060 | ± 0.036 | ± 0.013 | ± 0.066 | ± 0.039 | ± 0.007 | ± 0.054 | ± 0.154 |
| | | CTransPath | 0.511 | 0.343 | 0.436 | 0.512 | 0.228 | 0.113 | 0.228 | 0.503 | 0.235 | 0.345 |
| | | | ± 0.053 | ± 0.046 | ± 0.067 | ± 0.081 | ± 0.057 | ± 0.077 | ± 0.047 | ± 0.040 | ± 0.048 | ± 0.151 |
| | | GigaPath | 0.553 | 0.370 | 0.474 | 0.556 | 0.291 | 0.193 | 0.241 | 0.544 | 0.252 | 0.386 |
| | | | ± 0.073 | ± 0.021 | ± 0.049 | ± 0.067 | ± 0.025 | ± 0.067 | ± 0.038 | ± 0.035 | ± 0.052 | ± 0.148 |
| | | H-Optimus-0 | 0.598 | 0.379 | 0.492 | 0.655 | 0.302 | 0.222 | **0.274** | 0.561 | 0.260 | 0.416 |
| | | | ± 0.084 | ± 0.002 | ± 0.041 | ± 0.058 | ± 0.002 | ± 0.048 | ± 0.035 | ± 0.028 | ± 0.043 | ± 0.163 |
| | | H-Optimus-1 | **0.604** | 0.375 | 0.490 | **0.665** | 0.321 | 0.239 | 0.256 | **0.577** | **0.277** | **0.423** |
| | | | ± 0.078 | ± 0.006 | ± 0.040 | ± 0.049 | ± 0.018 | ± 0.017 | ± 0.038 | ± 0.012 | ± 0.043 | ± 0.164 |
| | | Hibou Large | 0.569 | 0.304 | 0.467 | 0.588 | 0.298 | 0.196 | 0.271 | 0.575 | 0.238 | 0.390 |
| | | | ± 0.079 | ± 0.024 | ± 0.081 | ± 0.042 | ± 0.035 | ± 0.055 | ± 0.068 | ± 0.004 | ± 0.042 | ± 0.159 |
| | | Kaiko Base 8 | 0.560 | 0.361 | 0.458 | 0.574 | 0.274 | 0.156 | 0.231 | 0.514 | 0.227 | 0.373 |
| | | | ± 0.075 | ± 0.022 | ± 0.069 | ± 0.065 | ± 0.023 | ± 0.084 | ± 0.050 | ± 0.037 | ± 0.031 | ± 0.158 |
| | | Phikon | 0.533 | 0.342 | 0.443 | 0.539 | 0.257 | 0.153 | 0.242 | 0.550 | 0.237 | 0.366 |
| | | | ± 0.091 | ± 0.077 | ± 0.070 | ± 0.055 | ± 0.007 | ± 0.083 | ± 0.026 | ± 0.002 | ± 0.046 | ± 0.153 |
| | | Phikon v2 | 0.538 | 0.353 | 0.444 | 0.553 | 0.247 | 0.177 | 0.267 | 0.538 | 0.246 | 0.374 |
| | | | ± 0.073 | ± 0.003 | ± 0.063 | ± 0.036 | ± 0.020 | ± 0.058 | ± 0.036 | ± 0.013 | ± 0.048 | ± 0.147 |
| | | UNI | 0.572 | 0.311 | 0.479 | 0.624 | 0.262 | 0.180 | 0.245 | 0.552 | 0.258 | 0.387 |
| | | | ± 0.083 | ± 0.078 | ± 0.076 | ± 0.035 | ± 0.031 | ± 0.046 | ± 0.038 | ± 0.016 | ± 0.044 | ± 0.169 |
| | | UNIv2 | 0.591 | 0.359 | 0.502 | 0.663 | 0.314 | 0.216 | 0.266 | 0.562 | 0.274 | 0.416 |
| | | | ± 0.082 | ± 0.042 | ± 0.046 | ± 0.015 | ± 0.010 | ± 0.043 | ± 0.038 | ± 0.010 | ± 0.040 | ± 0.165 |
| | | Virchow | 0.585 | 0.334 | **0.511** | 0.621 | 0.305 | 0.201 | 0.260 | 0.567 | 0.258 | 0.405 |
| | | | ± 0.093 | ± 0.001 | ± 0.059 | ± 0.062 | ± 0.005 | ± 0.046 | ± 0.033 | ± 0.024 | ± 0.042 | ± 0.164 |
| | | Virchow 2 | 0.594 | 0.356 | 0.476 | 0.644 | 0.258 | 0.203 | 0.267 | 0.572 | 0.261 | 0.403 |
| | | | ± 0.086 | ± 0.035 | ± 0.068 | ± 0.036 | ± 0.033 | ± 0.054 | ± 0.049 | ± 0.017 | ± 0.038 | ± 0.170 |
| 1NN-img | Foundation Model | CONCH v1 | 0.590 | 0.414 | 0.463 | 0.612 | 0.320 | 0.201 | 0.310 | 0.550 | 0.361 | 0.425 |
| | | | ± 0.071 | ± 0.003 | ± 0.080 | ± 0.075 | ± 0.020 | ± 0.127 | ± 0.055 | ± 0.026 | ± 0.066 | ± 0.141 |
| | | CONCH v1.5 | 0.598 | **0.490** | 0.488 | 0.596 | 0.337 | 0.204 | 0.280 | 0.589 | **0.383** | 0.441 |
| | | | ± 0.080 | ± 0.019 | ± 0.049 | ± 0.060 | ± 0.038 | ± 0.139 | ± 0.056 | ± 0.008 | ± 0.062 | ± 0.147 |
| | | CTransPath | 0.577 | 0.447 | 0.475 | 0.574 | 0.275 | 0.150 | 0.268 | 0.531 | 0.317 | 0.402 |
| | | | ± 0.070 | ± 0.021 | ± 0.064 | ± 0.075 | ± 0.104 | ± 0.167 | ± 0.090 | ± 0.013 | ± 0.093 | ± 0.154 |
| | | GigaPath | 0.611 | 0.455 | 0.497 | 0.652 | 0.338 | 0.181 | 0.314 | 0.561 | 0.339 | 0.439 |
| | | | ± 0.069 | ± 0.033 | ± 0.069 | ± 0.033 | ± 0.047 | ± 0.120 | ± 0.074 | ± 0.039 | ± 0.080 | ± 0.156 |
| | | H-Optimus-0 | 0.624 | 0.378 | 0.491 | 0.641 | 0.365 | 0.173 | **0.366** | 0.580 | 0.339 | 0.440 |
| | | | ± 0.055 | ± 0.102 | ± 0.018 | ± 0.038 | ± 0.042 | ± 0.164 | ± 0.058 | ± 0.008 | ± 0.053 | ± 0.155 |
| | | H-Optimus-1 | 0.622 | 0.436 | 0.497 | **0.675** | 0.350 | 0.210 | 0.273 | 0.596 | 0.377 | 0.448 |
| | | | ± 0.070 | ± 0.052 | ± 0.063 | ± 0.026 | ± 0.037 | ± 0.129 | ± 0.033 | ± 0.024 | ± 0.028 | ± 0.161 |
| | | Hibou Large | 0.623 | 0.420 | 0.484 | 0.642 | 0.342 | 0.229 | 0.314 | 0.589 | 0.329 | 0.441 |
| | | | ± 0.068 | ± 0.025 | ± 0.077 | ± 0.012 | ± 0.114 | ± 0.166 | ± 0.051 | ± 0.022 | ± 0.083 | ± 0.150 |
| | | Kaiko Base 8 | 0.609 | 0.440 | 0.437 | 0.600 | 0.329 | 0.163 | 0.317 | 0.533 | 0.323 | 0.417 |
| | | | ± 0.053 | ± 0.044 | ± 0.059 | ± 0.004 | ± 0.069 | ± 0.194 | ± 0.067 | ± 0.004 | ± 0.044 | ± 0.148 |
| | | Phikon | 0.555 | 0.408 | 0.406 | 0.605 | 0.280 | 0.086 | 0.299 | 0.504 | 0.316 | 0.384 |
| | | | ± 0.073 | ± 0.042 | ± 0.056 | ± 0.042 | ± 0.025 | ± 0.148 | ± 0.072 | ± 0.054 | ± 0.085 | ± 0.160 |
| | | Phikon v2 | 0.566 | 0.339 | 0.420 | 0.591 | 0.262 | 0.131 | 0.304 | 0.531 | 0.316 | 0.384 |
| | | | ± 0.045 | ± 0.115 | ± 0.035 | ± 0.041 | ± 0.059 | ± 0.143 | ± 0.052 | ± 0.042 | ± 0.069 | ± 0.154 |
| | | UNI | 0.617 | 0.394 | 0.478 | 0.633 | 0.249 | 0.205 | 0.290 | 0.586 | 0.350 | 0.422 |
| | | | ± 0.078 | ± 0.090 | ± 0.033 | ± 0.041 | ± 0.019 | ± 0.113 | ± 0.051 | ± 0.017 | ± 0.062 | ± 0.163 |
| | | UNIv2 | 0.645 | 0.449 | 0.488 | 0.661 | **0.377** | **0.310** | 0.343 | 0.586 | 0.356 | **0.468** |
| | | | ± 0.063 | ± 0.044 | ± 0.047 | ± 0.018 | ± 0.062 | ± 0.105 | ± 0.053 | ± 0.027 | ± 0.044 | ± 0.135 |
| | | Virchow | 0.597 | 0.451 | 0.481 | 0.549 | 0.331 | 0.174 | 0.268 | 0.592 | 0.327 | 0.419 |
| | | | ± 0.081 | ± 0.002 | ± 0.062 | ± 0.049 | ± 0.055 | ± 0.156 | ± 0.081 | ± 0.032 | ± 0.083 | ± 0.151 |
| | | Virchow 2 | **0.651** | 0.408 | **0.507** | 0.655 | 0.305 | 0.254 | 0.323 | **0.596** | 0.374 | 0.453 |
| | | | ± 0.077 | ± 0.086 | ± 0.037 | ± 0.016 | ± 0.114 | ± 0.132 | ± 0.082 | ± 0.031 | ± 0.051 | ± 0.154 |

*Table 9.* **Pearson Correlation Coefficient (PCC) across datasets and `timm` encoders for HEST-1k and HEST-1k-1NN**. The best and the second best score per dataset are in **bold** and underlined respectively. Standard deviations across the cross-validation folds are reported.

| Modality | Training | Benchmark Model | IDC | PRAD | PAAD | SKCM | COAD | READ | CCRCC | LUNG | LYMPH IDC | Avg |
|---|---|---|---|---|---|---|---|---|---|---|---|---|
| 1NN-graph | Pretrained | ResNet152 | 0.340 ± 0.036 | 0.338 ± 0.017 | 0.312 ± 0.072 | 0.366 ± 0.091 | 0.243 ± 0.040 | 0.090 ± 0.058 | 0.111 ± 0.030 | 0.382 ± 0.000 | 0.162 ± 0.042 | 0.261 ± 0.113 |
| | | ResNet18 | 0.338 ± 0.041 | 0.324 ± 0.014 | 0.295 ± 0.060 | 0.333 ± 0.066 | 0.231 ± 0.040 | 0.093 ± 0.062 | 0.109 ± 0.035 | 0.380 ± 0.002 | 0.153 ± 0.055 | 0.251 ± 0.108 |
| | | ResNet34 | 0.347 ± 0.044 | 0.327 ± 0.017 | 0.307 ± 0.072 | 0.315 ± 0.059 | 0.219 ± 0.031 | 0.068 ± 0.046 | 0.115 ± 0.033 | 0.371 ± 0.023 | 0.154 ± 0.047 | 0.247 ± 0.111 |
| | | ResNet50 | 0.341 ± 0.032 | 0.331 ± 0.015 | 0.319 ± 0.067 | 0.361 ± 0.117 | 0.212 ± 0.013 | 0.097 ± 0.051 | 0.116 ± 0.032 | 0.359 ± 0.030 | 0.157 ± 0.044 | 0.255 ± 0.109 |
| | | ViT-b/16 | 0.326 ± 0.031 | 0.339 ± 0.017 | 0.278 ± 0.043 | 0.302 ± 0.063 | 0.198 ± 0.002 | 0.110 ± 0.031 | 0.107 ± 0.026 | 0.363 ± 0.007 | 0.149 ± 0.048 | 0.241 ± 0.101 |
| | | ViT-h/14 | 0.331 ± 0.029 | 0.336 ± 0.015 | 0.296 ± 0.043 | 0.338 ± 0.068 | 0.236 ± 0.036 | 0.080 ± 0.059 | 0.100 ± 0.027 | 0.384 ± 0.007 | 0.147 ± 0.050 | 0.250 ± 0.114 |
| | | ViT-l/16 | 0.335 ± 0.033 | 0.337 ± 0.013 | 0.307 ± 0.044 | 0.306 ± 0.052 | 0.192 ± 0.003 | 0.118 ± 0.042 | 0.113 ± 0.028 | 0.373 ± 0.008 | 0.157 ± 0.046 | 0.249 ± 0.103 |
| | | ViT-s/16 | 0.331 ± 0.032 | 0.336 ± 0.016 | 0.283 ± 0.047 | 0.280 ± 0.078 | 0.200 ± 0.006 | 0.099 ± 0.050 | 0.104 ± 0.030 | 0.355 ± 0.019 | 0.148 ± 0.047 | 0.237 ± 0.102 |
| | Untrained | ResNet152 | 0.316 ± 0.038 | 0.254 ± 0.024 | 0.245 ± 0.025 | 0.139 ± 0.045 | 0.101 ± 0.044 | 0.040 ± 0.020 | 0.067 ± 0.030 | 0.247 ± 0.136 | 0.103 ± 0.079 | 0.168 ± 0.098 |
| | | ResNet18 | 0.320 ± 0.038 | 0.257 ± 0.025 | 0.275 ± 0.067 | 0.157 ± 0.056 | 0.176 ± 0.027 | 0.050 ± 0.031 | 0.082 ± 0.037 | 0.384 ± 0.001 | 0.148 ± 0.069 | 0.205 ± 0.111 |
| | | ResNet34 | 0.319 ± 0.037 | 0.255 ± 0.024 | 0.284 ± 0.061 | 0.169 ± 0.050 | 0.169 ± 0.021 | 0.055 ± 0.034 | 0.082 ± 0.035 | 0.368 ± 0.007 | 0.147 ± 0.068 | 0.205 ± 0.107 |
| | | ResNet50 | 0.316 ± 0.038 | 0.253 ± 0.025 | 0.282 ± 0.060 | 0.057 ± 0.024 | 0.175 ± 0.027 | 0.045 ± 0.028 | 0.066 ± 0.031 | 0.285 ± 0.095 | 0.144 ± 0.069 | 0.180 ± 0.108 |
| | | ViT-b/16 | 0.320 ± 0.052 | 0.277 ± 0.019 | 0.280 ± 0.054 | 0.303 ± 0.074 | 0.185 ± 0.027 | 0.082 ± 0.058 | 0.079 ± 0.019 | 0.390 ± 0.004 | 0.153 ± 0.060 | 0.230 ± 0.110 |
| | | ViT-h/14 | 0.320 ± 0.054 | 0.161 ± 0.140 | 0.271 ± 0.058 | 0.341 ± 0.094 | 0.190 ± 0.026 | 0.079 ± 0.063 | 0.071 ± 0.022 | 0.189 ± 0.145 | 0.147 ± 0.055 | 0.197 ± 0.097 |
| | | ViT-l/16 | 0.318 ± 0.055 | 0.277 ± 0.020 | 0.276 ± 0.058 | 0.337 ± 0.091 | 0.185 ± 0.028 | 0.082 ± 0.061 | 0.079 ± 0.022 | 0.383 ± 0.003 | 0.150 ± 0.060 | 0.232 ± 0.112 |
| | | ViT-s/16 | 0.321 ± 0.051 | 0.278 ± 0.020 | 0.275 ± 0.053 | 0.315 ± 0.090 | 0.188 ± 0.032 | 0.080 ± 0.060 | 0.081 ± 0.023 | 0.388 ± 0.007 | 0.149 ± 0.063 | 0.230 ± 0.111 |
| 1NN-img | Pretrained | ResNet152 | 0.487 ± 0.033 | 0.385 ± 0.040 | 0.393 ± 0.043 | 0.521 ± 0.054 | 0.252 ± 0.096 | 0.110 ± 0.099 | 0.216 ± 0.057 | 0.488 ± 0.012 | **0.324** ± 0.062 | 0.353 ± 0.140 |
| | | ResNet18 | 0.475 ± 0.034 | 0.409 ± 0.009 | 0.428 ± 0.049 | 0.409 ± 0.090 | 0.264 ± 0.096 | 0.108 ± 0.111 | 0.263 ± 0.071 | 0.501 ± 0.016 | 0.264 ± 0.070 | 0.347 ± 0.129 |
| | | ResNet34 | 0.490 ± 0.034 | 0.320 ± 0.082 | 0.392 ± 0.049 | 0.417 ± 0.018 | 0.268 ± 0.037 | 0.095 ± 0.119 | 0.257 ± 0.060 | 0.448 ± 0.081 | 0.303 ± 0.064 | 0.332 ± 0.120 |
| | | ResNet50 | 0.504 ± 0.015 | **0.410** ± 0.014 | 0.387 ± 0.055 | 0.505 ± 0.076 | 0.297 ± 0.104 | 0.122 ± 0.095 | **0.264** ± 0.069 | 0.482 ± 0.026 | 0.294 ± 0.074 | 0.363 ± 0.130 |
| | | ViT-b/16 | 0.509 ± 0.045 | 0.391 ± 0.027 | 0.392 ± 0.064 | 0.510 ± 0.000 | 0.195 ± 0.096 | 0.122 ± 0.062 | 0.216 ± 0.059 | 0.475 ± 0.001 | 0.317 ± 0.055 | 0.347 ± 0.055 |
| | | ViT-h/14 | 0.509 ± 0.035 | 0.391 ± 0.001 | 0.426 ± 0.047 | 0.530 ± 0.071 | 0.275 ± 0.117 | **0.174** ± 0.098 | 0.259 ± 0.062 | 0.503 ± 0.028 | 0.301 ± 0.064 | 0.374 ± 0.128 |
| | | ViT-l/16 | **0.544** ± 0.046 | 0.365 ± 0.040 | **0.428** ± 0.069 | **0.617** ± 0.074 | **0.298** ± 0.069 | 0.107 ± 0.119 | 0.245 ± 0.050 | **0.535** ± 0.008 | 0.315 ± 0.081 | **0.384** ± 0.163 |
| | | ViT-s/16 | 0.499 ± 0.049 | 0.356 ± 0.032 | 0.378 ± 0.047 | 0.469 ± 0.016 | 0.287 ± 0.084 | 0.100 ± 0.073 | 0.190 ± 0.032 | 0.468 ± 0.028 | 0.310 ± 0.064 | 0.340 ± 0.134 |
| | Untrained | ResNet152 | 0.327 ± 0.034 | 0.232 ± 0.053 | 0.298 ± 0.090 | 0.299 ± 0.044 | 0.019 ± 0.009 | 0.039 ± 0.051 | 0.115 ± 0.139 | 0.366 ± 0.068 | 0.182 ± 0.085 | 0.209 ± 0.127 |
| | | ResNet18 | 0.317 ± 0.071 | 0.227 ± 0.055 | 0.284 ± 0.100 | 0.208 ± 0.135 | 0.152 ± 0.013 | 0.074 ± 0.096 | 0.089 ± 0.134 | 0.363 ± 0.036 | 0.199 ± 0.118 | 0.212 ± 0.098 |
| | | ResNet34 | 0.284 ± 0.081 | 0.266 ± 0.086 | 0.311 ± 0.079 | 0.297 ± 0.095 | 0.091 ± 0.094 | 0.045 ± 0.059 | 0.122 ± 0.137 | 0.413 ± 0.107 | 0.173 ± 0.103 | 0.222 ± 0.121 |
| | | ResNet50 | 0.275 ± 0.097 | 0.190 ± 0.072 | 0.276 ± 0.057 | 0.109 ± 0.094 | 0.063 ± 0.076 | 0.056 ± 0.068 | 0.068 ± 0.058 | 0.273 ± 0.058 | 0.142 ± 0.098 | 0.161 ± 0.095 |
| | | ViT-b/16 | 0.282 ± 0.125 | 0.253 ± 0.056 | 0.302 ± 0.048 | 0.316 ± 0.067 | 0.233 ± 0.019 | 0.095 ± 0.035 | 0.104 ± 0.127 | 0.414 ± 0.037 | 0.183 ± 0.119 | 0.242 ± 0.103 |
| | | ViT-h/14 | 0.239 ± 0.153 | 0.194 ± 0.029 | 0.312 ± 0.048 | 0.338 ± 0.055 | 0.229 ± 0.035 | 0.105 ± 0.043 | 0.101 ± 0.124 | 0.407 ± 0.058 | 0.187 ± 0.111 | 0.234 ± 0.103 |
| | | ViT-l/16 | 0.280 ± 0.124 | 0.216 ± 0.030 | 0.307 ± 0.048 | 0.311 ± 0.071 | 0.236 ± 0.038 | 0.098 ± 0.031 | 0.100 ± 0.131 | 0.421 ± 0.041 | 0.183 ± 0.118 | 0.239 ± 0.104 |
| | | ViT-s/16 | 0.286 ± 0.112 | 0.231 ± 0.049 | 0.297 ± 0.047 | 0.319 ± 0.057 | 0.267 ± 0.048 | 0.102 ± 0.035 | 0.093 ± 0.133 | 0.382 ± 0.044 | 0.183 ± 0.111 | 0.240 ± 0.098 |

# F. Additional Experiments on the Untrained Models

## F.1. Impact of Architecture on the Untrained Models

*Table 10.* Hyperparameter sweep for untrained ViTs on the RxRx3-core benchmark. Values report mean Recall@5% $\pm$ standard deviation across three folds.

| Hyperparameter | Value | Mean Recall@5% |
|---|---|---|
| aggregation | avg | 0.19±0.02 |
| | cls | 0.20±0.02 |
| size | s | 0.20±0.02 |
| | b | 0.20±0.02 |
| | l | 0.19±0.02 |
| | h | 0.19±0.02 |
| patch size | 2 | 0.20±0.02 |
| | 4 | 0.20±0.02 |
| | 8 | 0.20±0.02 |
| | 14 (h) / 16 (s,b,l) | 0.19±0.02 |
| | 32 | 0.17±0.01 |
| layer depth | 0 | 0.20±0.02 |
| | 1 | 0.19±0.02 |
| | 2 | 0.19±0.02 |
| | 3 (final layer) | 0.19±0.02 |

## F.2. Impact of Weight Initialization on the Untrained Models

*Table 11.* **Impact of weight initialization on recall at 5% across biological benchmarks.** Mean and standard deviation across 3 random seeds for untrained models. Results are reported for all benchmarks (CORUM, HuMAP, Reactome, SIGNOR, StringDB) across three evaluation folds.

| Model | Architecture | Fold | CORUM | HuMAP | Reactome | SIGNOR | StringDB |
|---|---|---|---|---|---|---|---|
| SingleConv | CNN | 1 | 0.284±0.009 | 0.363±0.007 | 0.080±0.001 | 0.037±0.008 | 0.222±0.003 |
| SingleConv | CNN | 2 | 0.281±0.021 | 0.350±0.030 | 0.064±0.001 | 0.066±0.005 | 0.226±0.011 |
| SingleConv | CNN | 3 | 0.263±0.004 | 0.309±0.014 | 0.068±0.001 | 0.046±0.005 | 0.204±0.005 |
| AlexNet | CNN | 1 | 0.111±0.039 | 0.106±0.029 | 0.059±0.005 | 0.057±0.003 | 0.087±0.016 |
| AlexNet | CNN | 2 | 0.114±0.012 | 0.120±0.006 | 0.059±0.017 | 0.057±0.013 | 0.102±0.015 |
| AlexNet | CNN | 3 | 0.103±0.023 | 0.098±0.023 | 0.049±0.010 | 0.058±0.018 | 0.089±0.021 |
| ResNet18 | CNN | 1 | 0.143±0.024 | 0.166±0.044 | 0.054±0.007 | 0.053±0.015 | 0.124±0.015 |
| ResNet18 | CNN | 2 | 0.138±0.011 | 0.156±0.008 | 0.057±0.005 | 0.055±0.004 | 0.117±0.007 |
| ResNet18 | CNN | 3 | 0.148±0.011 | 0.162±0.013 | 0.060±0.011 | 0.059±0.015 | 0.119±0.004 |
| ResNet50 | CNN | 1 | 0.122±0.018 | 0.131±0.023 | 0.055±0.008 | 0.046±0.013 | 0.100±0.007 |
| ResNet50 | CNN | 2 | 0.139±0.014 | 0.142±0.011 | 0.054±0.012 | 0.047±0.005 | 0.108±0.005 |
| ResNet50 | CNN | 3 | 0.128±0.022 | 0.134±0.029 | 0.069±0.012 | 0.059±0.014 | 0.102±0.014 |
| ResNet152 | CNN | 1 | 0.124±0.007 | 0.135±0.004 | 0.059±0.008 | 0.044±0.001 | 0.107±0.009 |
| ResNet152 | CNN | 2 | 0.133±0.034 | 0.141±0.026 | 0.054±0.010 | 0.048±0.007 | 0.109±0.016 |
| ResNet152 | CNN | 3 | 0.119±0.012 | 0.129±0.016 | 0.057±0.004 | 0.055±0.014 | 0.098±0.007 |
| ViT-s/16 | ViT | 1 | 0.344±0.011 | 0.396±0.014 | 0.056±0.001 | 0.043±0.002 | 0.251±0.011 |
| ViT-s/16 | ViT | 2 | 0.325±0.010 | 0.376±0.012 | 0.050±0.006 | 0.043±0.002 | 0.242±0.005 |
| ViT-s/16 | ViT | 3 | 0.294±0.007 | 0.300±0.002 | 0.051±0.003 | 0.044±0.003 | 0.216±0.005 |
| ViT-b/16 | ViT | 1 | 0.329±0.001 | 0.382±0.009 | 0.058±0.006 | 0.038±0.005 | 0.238±0.003 |
| ViT-b/16 | ViT | 2 | 0.318±0.002 | 0.371±0.006 | 0.052±0.004 | 0.042±0.009 | 0.242±0.004 |
| ViT-b/16 | ViT | 3 | 0.283±0.011 | 0.288±0.006 | 0.050±0.001 | 0.044±0.002 | 0.211±0.005 |
| ViT-l/16 | ViT | 1 | 0.319±0.006 | 0.372±0.013 | 0.065±0.003 | 0.033±0.002 | 0.237±0.007 |
| ViT-l/16 | ViT | 2 | 0.314±0.009 | 0.363±0.006 | 0.053±0.005 | 0.043±0.005 | 0.235±0.007 |
| ViT-l/16 | ViT | 3 | 0.276±0.006 | 0.282±0.009 | 0.061±0.004 | 0.051±0.004 | 0.207±0.005 |
| ViT-h/14 | ViT | 1 | 0.314±0.003 | 0.373±0.010 | 0.062±0.004 | 0.032±0.002 | 0.239±0.003 |
| ViT-h/14 | ViT | 2 | 0.314±0.005 | 0.368±0.009 | 0.065±0.003 | 0.055±0.005 | 0.243±0.008 |
| ViT-h/14 | ViT | 3 | 0.270±0.004 | 0.287±0.003 | 0.059±0.001 | 0.052±0.002 | 0.212±0.008 |

*Table 12.* **Per-fold compound retrieval performance on the JUMP-CP subset benchmark**. Mean mAP is reported as mean $\pm$ standard deviation across 3 random seeds.

| Model | Architecture | Fold | Mean mAP |
|---|---|---|---|
| SingleConv | CNN | 0 | 0.357±0.018 |
| SingleConv | CNN | 1 | 0.315±0.008 |
| SingleConv | CNN | 2 | 0.366±0.017 |
| SingleConv | CNN | 3 | 0.340±0.016 |
| SingleConv | CNN | 4 | 0.341±0.023 |
| ResNet18 | CNN | 0 | 0.146±0.002 |
| ResNet18 | CNN | 1 | 0.252±0.006 |
| ResNet18 | CNN | 2 | 0.293±0.011 |
| ResNet18 | CNN | 3 | 0.272±0.007 |
| ResNet18 | CNN | 4 | 0.251±0.007 |
| ResNet50 | CNN | 0 | 0.152±0.000 |
| ResNet50 | CNN | 1 | 0.275±0.007 |
| ResNet50 | CNN | 2 | 0.332±0.007 |
| ResNet50 | CNN | 3 | 0.303±0.011 |
| ResNet50 | CNN | 4 | 0.274±0.006 |
| ResNet152 | CNN | 0 | 0.313±0.006 |
| ResNet152 | CNN | 1 | 0.274±0.001 |
| ResNet152 | CNN | 2 | 0.329±0.002 |
| ResNet152 | CNN | 3 | 0.302±0.003 |
| ResNet152 | CNN | 4 | 0.300±0.003 |
| ViT-s/16 | ViT | 0 | 0.313±0.007 |
| ViT-s/16 | ViT | 1 | 0.277±0.005 |
| ViT-s/16 | ViT | 2 | 0.336±0.005 |
| ViT-s/16 | ViT | 3 | 0.306±0.002 |
| ViT-s/16 | ViT | 4 | 0.298±0.006 |
| ViT-b/16 | ViT | 0 | 0.328±0.006 |
| ViT-b/16 | ViT | 1 | 0.287±0.004 |
| ViT-b/16 | ViT | 2 | 0.352±0.005 |
| ViT-b/16 | ViT | 3 | 0.323±0.006 |
| ViT-b/16 | ViT | 4 | 0.310±0.001 |
| ViT-l/16 | ViT | 0 | 0.331±0.002 |
| ViT-l/16 | ViT | 1 | 0.289±0.000 |
| ViT-l/16 | ViT | 2 | 0.357±0.002 |
| ViT-l/16 | ViT | 3 | 0.326±0.003 |
| ViT-l/16 | ViT | 4 | 0.316±0.004 |
| ViT-h/14 | ViT | 0 | 0.354±0.003 |
| ViT-h/14 | ViT | 1 | 0.310±0.005 |
| ViT-h/14 | ViT | 2 | 0.381±0.006 |
| ViT-h/14 | ViT | 3 | 0.343±0.003 |
| ViT-h/14 | ViT | 4 | 0.337±0.006 |

# G. Additional Results On RxRx3-Core

We additionally evaluated CellProfiler features on RxRx3-core to compare deep representations with standard hand-crafted morphology descriptors. Results in table 13 indicate that CellProfiler achieves an average Recall@5% of $0.17 \pm 0.01$, slightly outperforming simple pixel statistics ($0.16 \pm 0.02$), but remains slightly below the best untrained and pretrained models, represented here by ViT-l/16 untrained ($0.19 \pm 0.02$) and OpenPhenom ($0.20 \pm 0.02$). In this case, the representations that are aligned to interpretable features behave similarly to the uninformed models, which might indicate their reliance on similar shortcuts and spurious correlations. Importantly, the benchmark does not enable clear discrimination between their behaviors.

*Table 13.* RxRx3-core retrieval performance of CellProfiler features compared with pixel statistics, the best untrained ViT model, and OpenPhenom. Values report mean Recall@5% $\pm$ standard deviation across three folds.

| Feature type | Model | Average Recall@5% |
|---|---|---|
| Pixels statistics | Pixel Stats | $0.16 \pm 0.02$ |
| Human handcrafted features | CellProfiler | $0.17 \pm 0.01$ |
| Untrained network | ViT-l/16 | $0.19 \pm 0.02$ |
| Pretrained Fondation Model | OpenPhenom | $0.20 \pm 0.02$ |

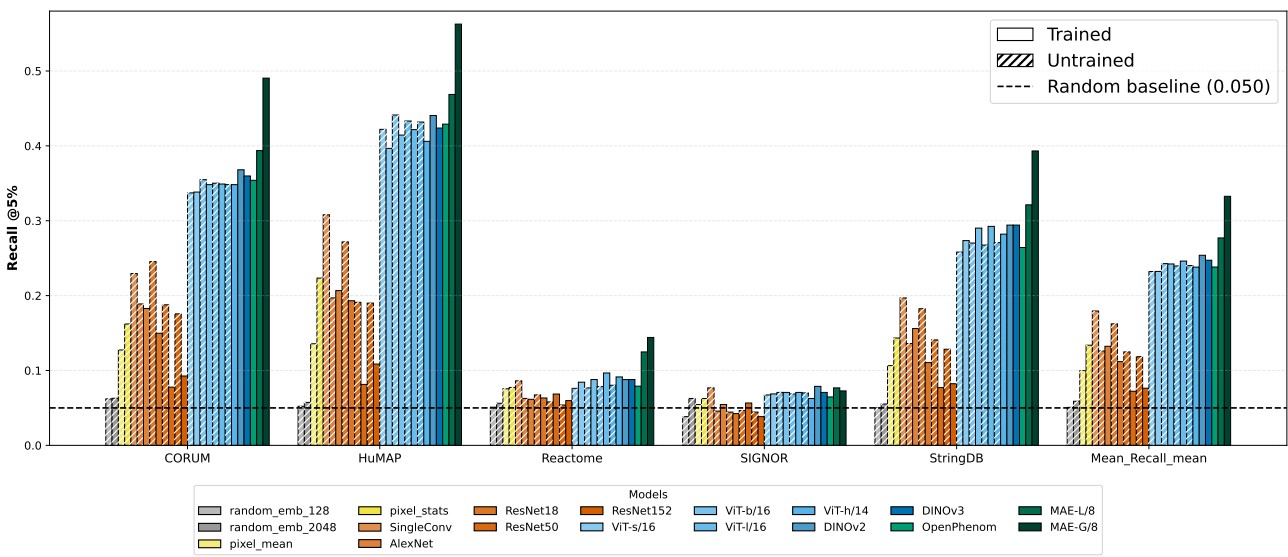

*Figure 18.* **Recall @5% for all models on the original full RxRx3-Core dataset.**

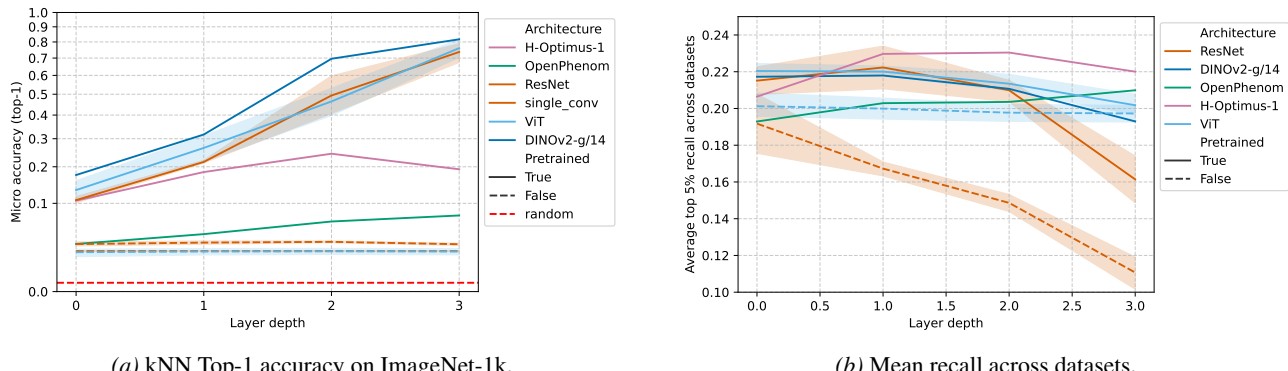

*(a)* kNN Top-1 accuracy on ImageNet-1k.    *(b)* Mean recall across datasets.

*Figure 19.* **Model performance as a function of network stage.** Both panels evaluate embeddings at four intermediate stages including the last layer. (a) Evolution of micro-accuracy on the ImageNet-1k validation set. (b) Average recall across all RxRx3-Core evaluated datasets.

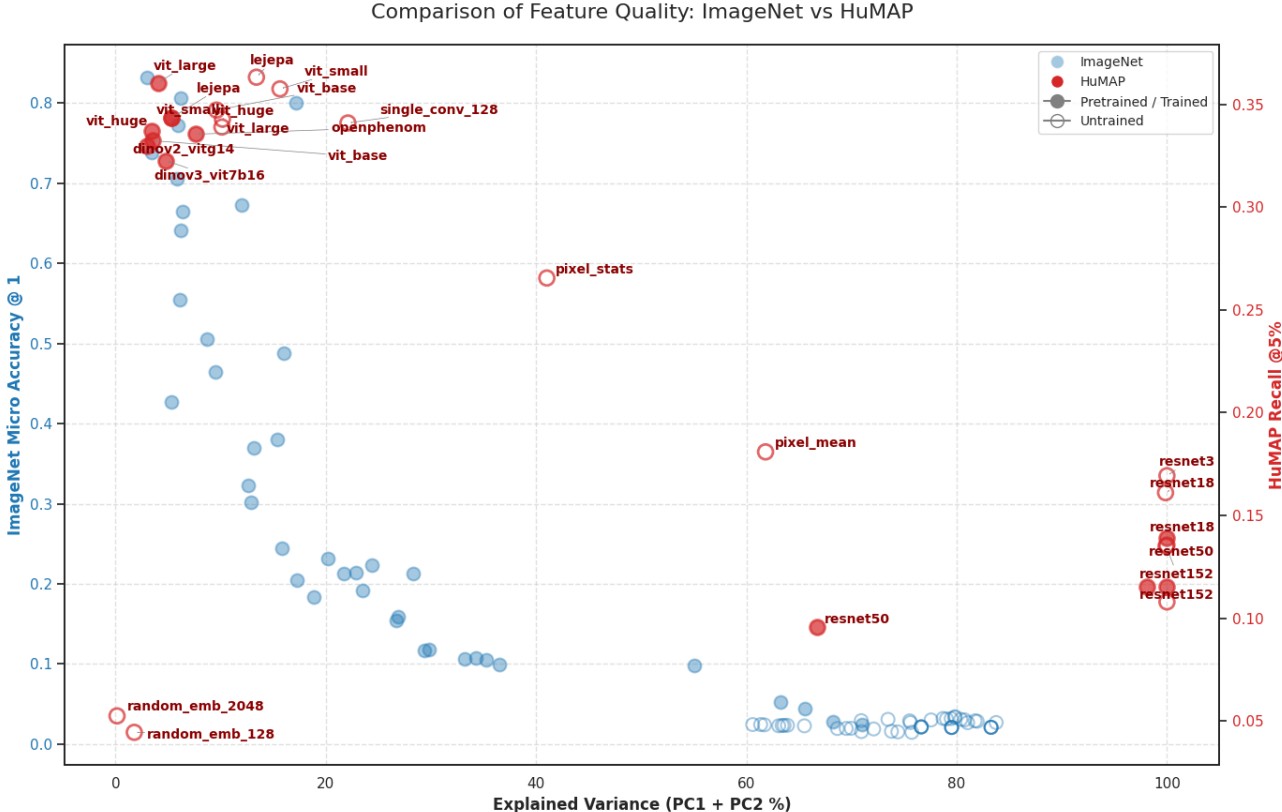

*Figure 20.* **Representational dimensionality versus performance on biological and natural image benchmarks**. Dataset specific scores are plotted against variance explained by the first two PCA components. Right axis: HuMAP mean recall at top 5%. Left axis: ImageNet-1k kNN top-1 accuracy. Points correspond to different model architectures, training states, and extraction layers.

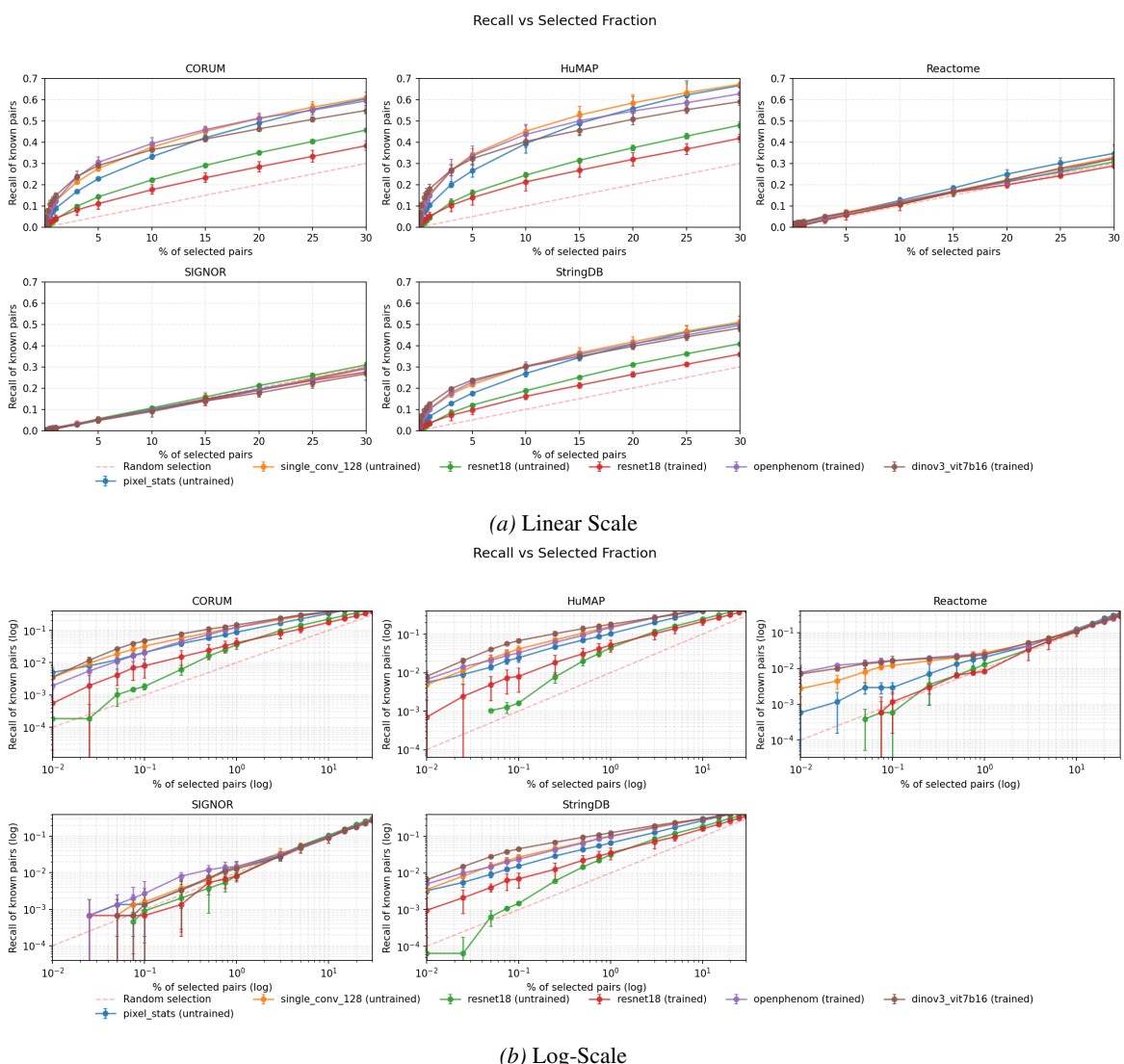

*Figure 21.* **Impact of selected pairs % on the recall per dataset.** A few important and diverse models are displayed. Red dash line indicates the random selection

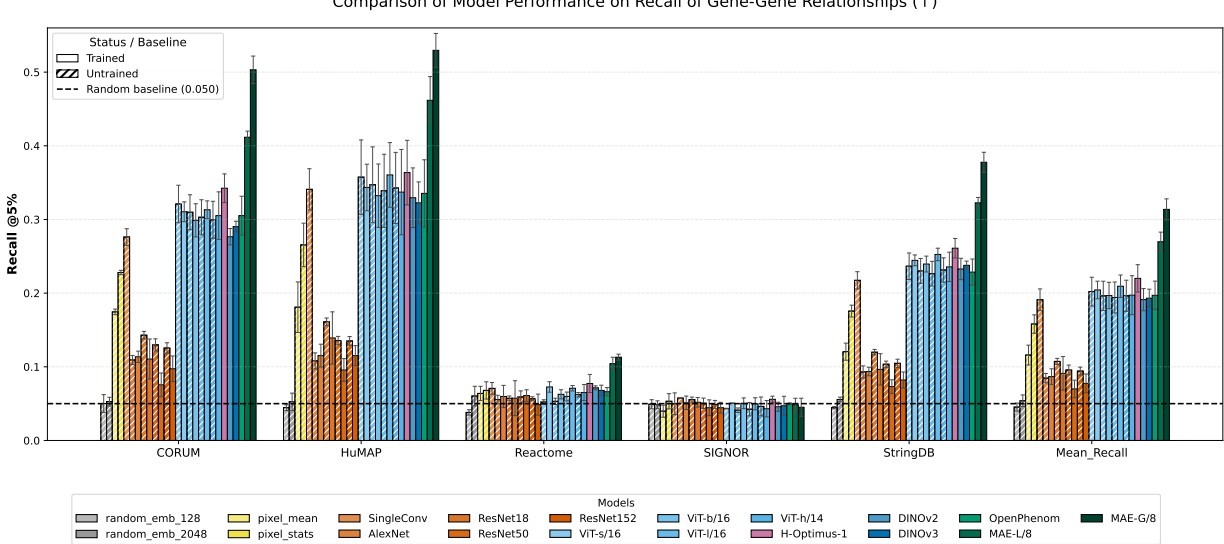

*(a)* Recall at $5\%$ performance across multiple gene interaction benchmarks (CORUM, HuMAP, Reactome, SIGNOR, StringDB).

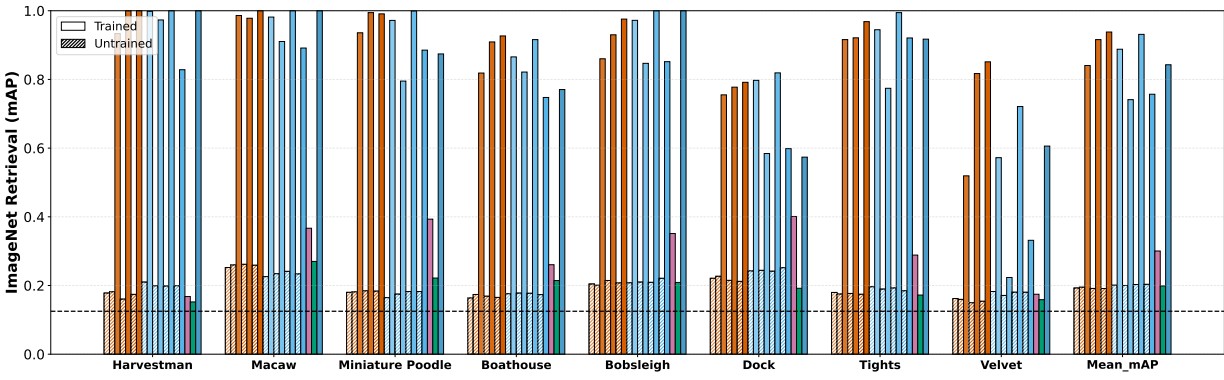

*(b)* Evaluation of mAP on ImageNet-1k validation set.

*Figure 22.* **Benchmarking model performance across cell culture and natural images tasks.** Performance is evaluated across: (a) gene-gene interaction retrieval and (b) ImageNet-1k classification. Solid bars indicate pretrained models; hatched bars indicate untrained models. Error bars represent variability across evaluation folds, and dashed horizontal lines denote random baselines.

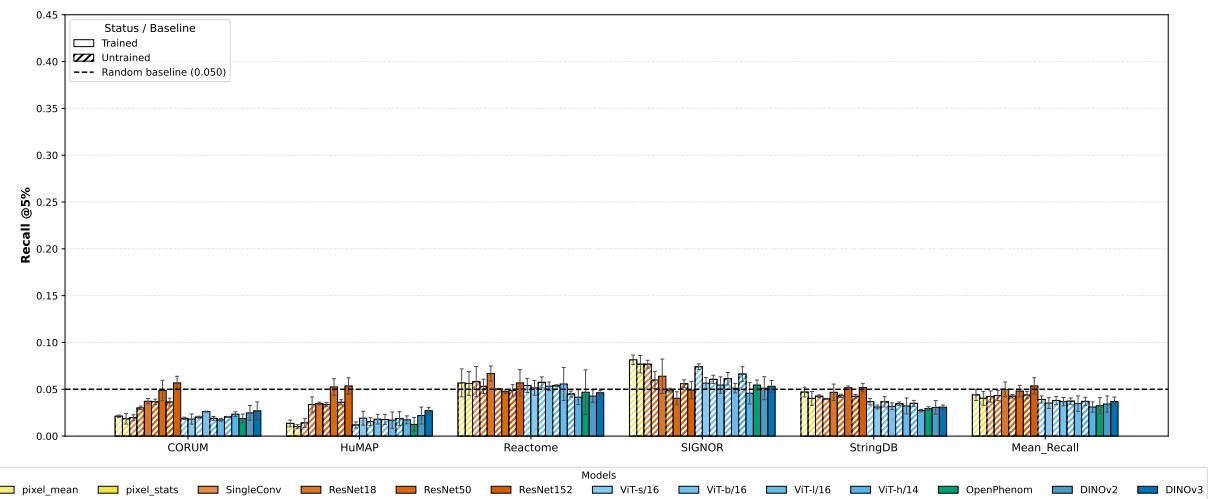

*Figure 23.* Model performance on gene-gene interaction retrieval when looking at bottom $5\%$ of cosine similarities between gene-gene pairs.

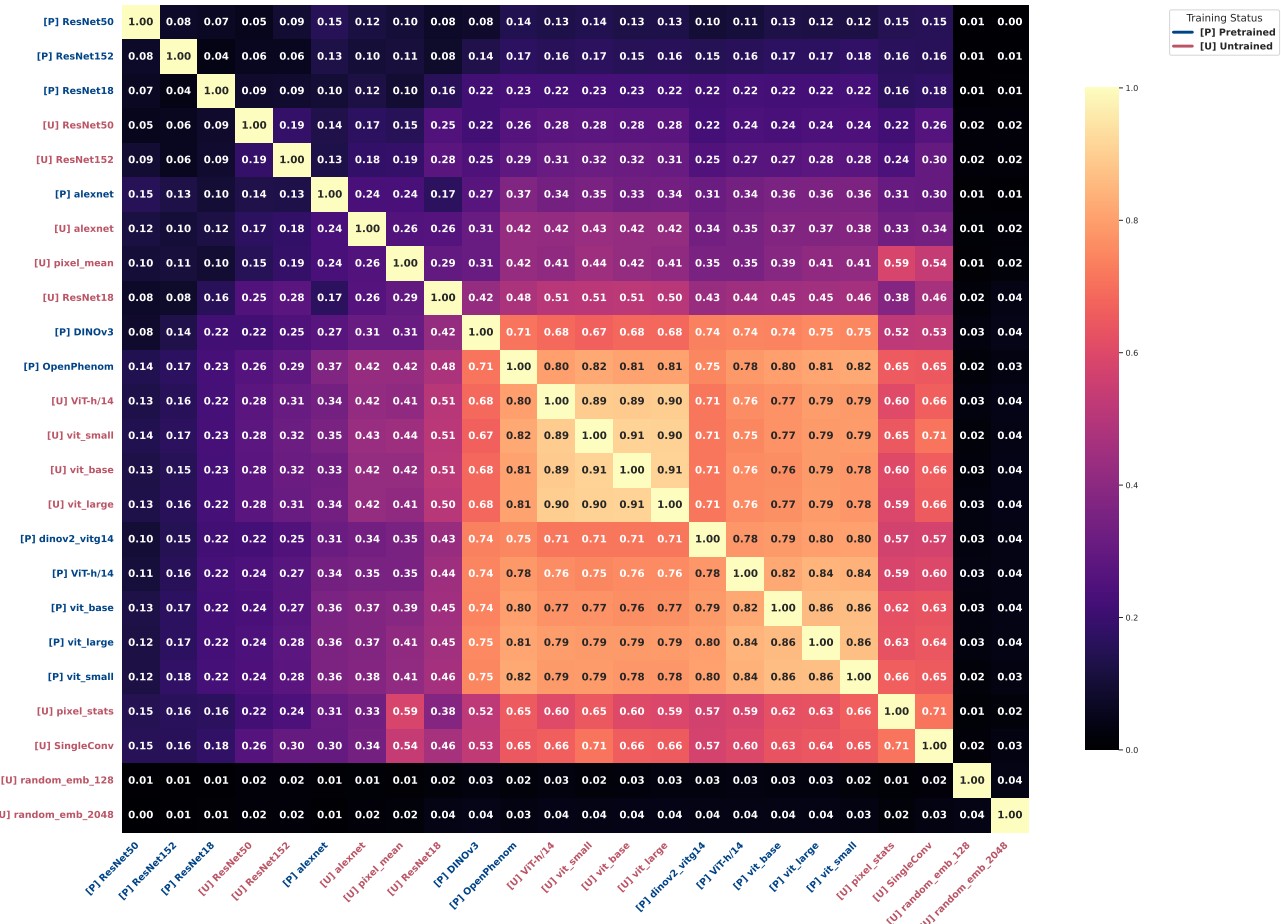

*Figure 24.* **Representational Similarity Analysis (RSA) across model architectures.** The heatmap displays the pairwise similarity between model embeddings calculated using Spearman's rank correlation coefficient. Models are hierarchically clustered to reveal functional groupings. Labels prefixed with **[P]** (blue) denote pretrained models, while **[U]** (red) indicates untrained/random initializations.

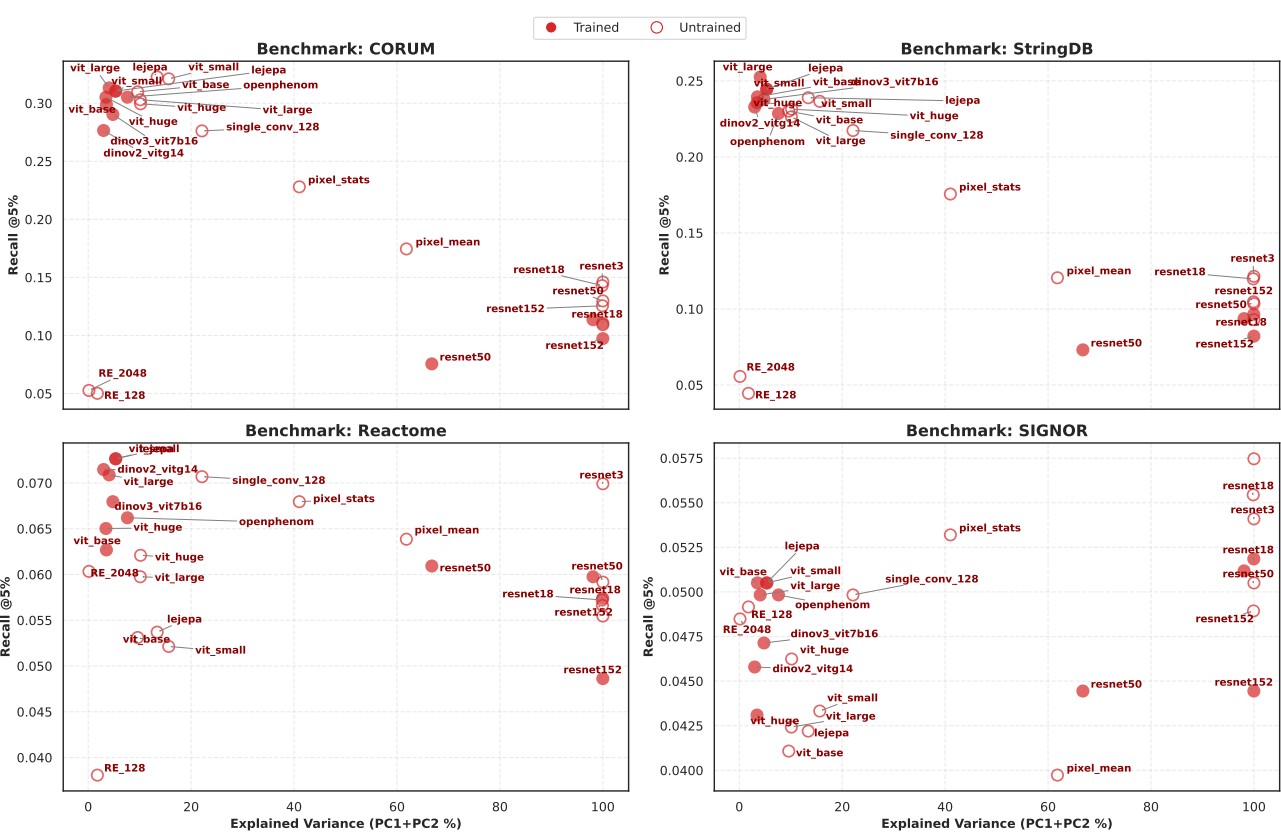

*Figure 25.* PCA Variance Explained of RxRx3-Core vs. Recall @5% across various datasets (CORUM, StringDB, Reactome and SIGNOR).

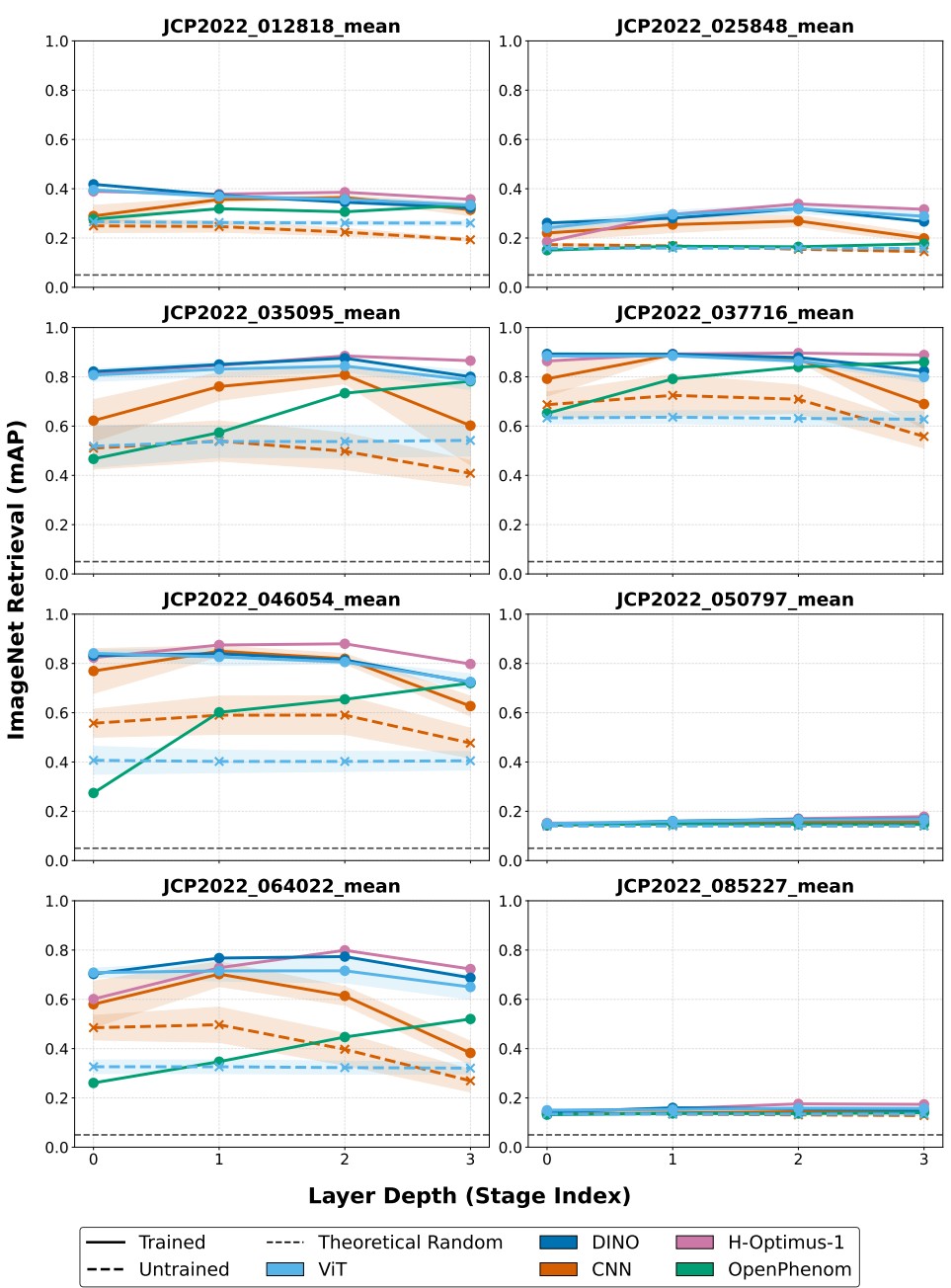

*Figure 26.* **Evolution of representation quality across intermediate layers for JUMP-CP compounds.** Performance is evaluated as Mean Average Precision (mAP) for ImageNet retrieval across eight distinct positive control compound. Each panel displays the mAP score as a function of the network intermediate layers (Layer Depth). Solid lines denote pretrained models, while dashed lines indicate untrained models; shaded areas represent variability across a given architecture. The dashed horizontal line in each plot represents the theoretical random baseline (0.125).

# H. Additional Experiments

### H.1. Retrieval of Mechanisms of Action

**Experimental setup.** We evaluate MoA retrieval using the 43 Mechanisms of Action selected from the JUMP-MoA reference plates described in Appendix A.2, each represented by two compounds. For each model, we extract Cell Painting image representations and aggregate them into well-level phenotypic profiles using the same aggregation and control-based normalization pipeline as in the JUMP-CP benchmark experiment, i.e. following Sanchez et al. (2026).

We then assess whether biologically related profiles are ranked close to each other in representation space using two metrics from the SPACe evaluation framework (Stossi et al., 2024): percent replicating and percent matching.

- Percent replicating measures whether replicate wells of the same compound are retrieved among the most similar profiles; for each compound, replicate wells are treated as positives and all other wells as negatives;

- Percent matching measures whether wells that have different treatments but share the same annotated mechanisms of action are retrieved among the most similar profiles.

To calculate percent replicating and percent matching, we used a threshold based on the 95th percentile of a null distribution

- We generated the null distribution using 20,000 random non-matching pair-wise Spearman correlations for each dataset;

- The final metrics were then determined by calculating the percentage of matching or replicating well pairs that had a correlation above this 95th percentile threshold

Therefore, 5% is the theoretical random for both metrics.

**Results.** Results in table 14 shows that pretrained models achieve the highest percent replicating, indicating stronger recovery of compound-level replicate consistency. However, untrained models substantially outperform the random baseline and reach comparable percent matching scores, suggesting that MoA retrieval partly relies on simple image statistics rather than learned semantic representations alone.

*Table 14.* JUMP-MoA retrieval results. Means across 86 compounds for % of replicating and across 43 MoAs for % of matching.

| Model | Mean % of Replicating | Mean % of Matching |
|---|---|---|
| Random | 5.0 | 5.0 |
| Pixel Stats | 22.3 | 10.5 |
| OpenPhenom | 37.8 | 11.4 |
| ResNet152 (untrained) | 27.3 | 11.6 |
| ViT-l/16 (pretrained) | 36.8 | 12.3 |
| SingleConv (untrained) | 30.5 | 12.8 |

## H.2. Breast Cancer Subtyping

**Experimental setup.** Below we provide additional results on two cancer subtyping datasets (multiclass classification) at region-of-interest (RoI) level, namely BRACS-RoI (Brancati et al., 2022) and BACH (Aresta et al., 2019). We follow the evaluation protocol and the provided implementation from GrapHist (Öğüt et al., 2026). Train/test splits are conducted at the patient level. For BACH some of the patient data is missing, thus only a subset of known patients is used to define the test set. We pre-filter the datasets only allowing samples for which reference graph-based representations are available from https://huggingface.co/ogutsevda/datasets.

The authors default to processing patches at $20\times$ magnification ($0.5\mu$m / pixel) and patchifying them at $224\times224$ pixels. However, the authors also report that the performance of GrapHist is remains reasonably robust across $224\times224$ px, $448\times448$ px, $896\times896$ px, and full-sized RoI patches. When applicable, MIL aggregation is performed on patch embeddings using additive (Javed et al., 2022), conjunctive (Early et al., 2024), and attentive ABMIL (Ilse et al., 2018). In our experiments we additionally evaluate `mean` and `median` pooling as well as a concatenation of feature-wise means and standard deviations per RoI (`mean_std`).

The reference magnification scale might be overly restrictive to perform structure-only evaluation. Following our reasoning for creation of the binned version of HEST-1k, we also evaluate larger patches. In order to keep the scale consistent, we discard the RoI smaller than the requested patch size. For BRACS this approach impacts the total sample count. The summary of sample distribution per class is provided in table 15

For added context, we embed patches with H-Optimus-1 and pool them per RoI with MIL and simple aggregations as described above. Finally, we `mean` / `median` / `mean_std` -pool hand-crafted features of cell nuclei, which leads to representations that ignore the global cell layout by construction. We emphasize, however, that simple morphological features (e.g., nucleus size, eccentricity, etc.) might already leak information about the local density and variability of cells.

*Table 15.* **BRACS-RoI and BACH sample counts for each level of patchification used in the experiments.** The base patch size is chosen to correspond to $112\times112~\mu$m

*(a)* Class distribution for BRACS. The initial resolution of full RoIs varies greatly and the dataset remains imbalanced across all scales.

| Resolution | Full RoI | | $448\times448$ | | $896\times896$ | |
| Class | Test | Train | Test | Train | Test | Train |
|---|---|---|---|---|---|---|
| ADH | 102 | 403 | 100 | 396 | 72 | 275 |
| DCIS | 163 | 605 | 163 | 603 | 143 | 517 |
| FEA | 141 | 611 | 136 | 588 | 56 | 303 |
| IC | 130 | 516 | 129 | 514 | 119 | 483 |
| N | 105 | 369 | 101 | 352 | 90 | 299 |
| PB | 172 | 661 | 171 | 655 | 149 | 559 |
| UDH | 86 | 429 | 83 | 415 | 56 | 287 |
| Total | 899 | 3594 | 883 | 3523 | 685 | 2723 |

*(b)* Class distribution for BACH. The initial resolution of full RoIs is fixed and the dataset remains balanced across all scales.

| Resolution | $267\times267$, $534\times534$, RoI square crop | |
| Class | Test | Train |
|---|---|---|
| Benign | 15 | 85 |
| InSitu | 15 | 85 |
| Invasive | 15 | 85 |
| Normal | 15 | 85 |
| Total | 60 | 340 |

Within this framework our structure-only baselines are represented by images of cell graphs embedded with models of the ResNet family pretrained on ImageNet-1k. To offer a complementary perspective on this problem we also evaluate morphological embeddings of nuclei pooled at the RoI-level without in a structure-agnostic way.

**Results and discussion.** Table 16 reports results of the original experiments on GrapHist (Öğüt et al., 2026). As shown in table 17 the structure-only baselines offer non-trivial macro F1 scores for the multiclass classification problem, challenging the foundation model H-Optimus-1 in some settings. As discussed above in context of HEST-1k, while these results do not allow us to make strong about the importance of structure alone, they provide important context on the behavior of the dataset. For instance, both structure-only and nuclei morphology-only baselines perform similarly suggesting further investigation into their efficient fusion and raising concerns about under-performance of other methods. The latter can be explained by datasets specific biases such as RoI selection, cell count, staining protocols etc. but also hyperparameter selection and the evaluation setup used.

*Table 16.* Reported reference results from (Öğüt et al., 2026). The RoI-level datasets are processed in patches and aggregated using three different MIL approaches. The provided values correspond to the best MIL strategy per model. Held-out test macro F1 scores are reported with standard deviations across models fit on 5 cross-validation folds.

| Modality | Training | Dataset Model | **BACH** | **BRACS** |
|---|---|---|---|---|
| Morphological cell graphs | Pretrained IID | ACM-bio | 38.37 ±2.16 | 21.55 ± 2.62 |
| | | ACM-UNI | 29.03 ± 9.92 | 19.95 ± 3.27 |
| | | GrapHist | **69.16** ± 3.37 | **60.30** ±0.46 |
| Full RGB Images | Pretrained, IID | DINOv2 | 59.25 ± 3.34 | 52.68 ± 1.78 |
| | | MAE | 57.94 ± 3.91 | 56.33 ± 0.76 |

*Table 17.* The best and the second best score per dataset are in **bold** and underlined respectively. Following the reference implementation (Öğüt et al., 2026) we provide the macro F1 scores and standard deviations across five validation folds in parentheses. For all datasets the best performing patch aggregation method is reported. For BACH the best performing patchification strategy is reported.

| Modality | Model | Dataset Training | **BACH** | **BRACS** (448×448) | **BRACS** (896×896) |
|---|---|---|---|---|---|
| Full RGB Images | H-Optimus-1 | Pretrained, IID | **90.88** ± 2.13 | 48.20 ± 1.71 | **75.65** ± 0.99 |
| Morphological Features | `mean_std` | - | 62.06 ± 2.42 | **56.98** ± 1.02 | 58.02 ± 1.17 |
| Structure Only (Images of Binary Graphs) | ResNet152 | Pretrained, OOD | 54.83 ± 2.86 | 47.29 ± 1.98 | 53.32 ± 1.37 |
| | ResNet18 | Pretrained, OOD | 48.21 ± 4.22 | 43.31 ± 1.56 | 46.90 ± 1.71 |
| | ResNet34 | Pretrained, OOD | 53.43 ± 2.22 | 44.95 ± 0.70 | 45.72 ± 1.83 |
| | ResNet50 | Pretrained, OOD | 52.69 ± 3.06 | 49.18 ± 1.32 | 51.90 ± 1.84 |

