# OpenReview forum: "Deep Learning for BioImaging: What Are We Really Learning?"
_ICML.cc/2026/Conference — ICML 2026 regular_

### Official Review · Reviewer_2VWP · 2026-03-11

**Soundness:** 4
**Presentation:** 4
**Significance:** 4
**Originality:** 3
**Overall Recommendation:** 6
**Confidence:** 4

**Summary:**

The paper asks a simple (but very important) question: do foundation models for microscopy actually learn biologically meaningful features? The authors show, surprisingly, that on several widely used benchmarks, pretrained and foundation models perform comparably to untrained random networks and simple pixel statistics baselines. They introduce a set of diagnostic baselines (including a structure-only tissue representation based on cell graph topology, untrained deep encoders, and pixel level statistics) to expose these shortcomings, and conclude with concrete recommendations for more rigorous benchmarking practice in the field.

**Compliance With Llm Reviewing Policy:**

Affirmed.

**Final Justification:**

I maintain a 6: Strong Accept. Concerns were adequately addressed, and has reinforced my initial score.

**Key Questions For Authors:**

The evaluation primarily assesses representation quality through embedding similarity and retrieval-based metrics on raw embeddings. However, in practice, these embeddings are frequently used in downstream tasks such as classification or multiple instance learning (MIL). Do the authors think the conclusions drawn here would hold in those settings? More broadly, is it possible that certain embeddings, while indistinguishable by distance-based metrics, are better suited to specific downstream use cases? It would be valuable to hear the authors' thoughts on whether embedding utility should be assessed in a more task-specific manner, and whether such an analysis is within scope for a future revision.

**Limitations:**

Yes.

**Strengths And Weaknesses:**

S1: The paper is a rigorous and thorough piece of work. The evaluation is comprehensive across models, datasets, parameter settings, and layer-wise analyses. The framing of the problem is compelling and clearly motivated - the authors make a strong case for why this question matters now, as the field increasingly relies on foundation models without scrutinizing what they actually learn. The proposed baselines are well-motivated and elegantly designed, particularly the structure-only tissue representation, which provides a genuinely novel diagnostic lens.

W1: Could the authors detail which batch correction technique was performed for the cell culture results if any at all (4.1)?

---

> ### Author Rebuttal · Authors · 2026-03-30
>
> We thank the reviewer for the thoughtful review and the positive assessment of our contributions.
>
> **[W1]** To perform batch correction of cell culture images, we used plate-wise alignment on negative controls following:
>
> For RxRx3-Core, Kraus et al., [1] with PCA fitted on negative controls, applied to all well representations followed by a standard centering and scaling of features.
>
> For JUMP-CP controls and JUMP-MoA, Sanchez et al., [2] with plate-wised sphering of features fitted on negative controls, applied to all well representations followed by centering and inverse normal transform of features.
>
> **[Questions]** The questions raised by the reviewer are very interesting and we will attempt to provide two complementary perspectives: the one of utility and the one of biological alignment.
>
> * **[Downstream tasks]** We added cancer subtyping (learnt MIL aggregation) and we observe that the initial conclusions about utility of the baselines still hold but the task-specific models do have a clear edge. Additionally, in the case of spatial transcriptomics for tissue imaging we already observe similar discriminative behaviour on subsets of HEST-1k where a linear prediction of target expressions is trained from frozen embeddings. These downstream tasks will reveal the models’ capabilities beyond a coarse study of the organisation of the latent space and may help practitioners to find better solutions for a task at hand.
>
> * **[Beyond extrinsic evaluation]** While deducing representations’ utility from specific downstream tasks one can wonder where this places general-purpose vision encoders, which often perform well without having learnt biologically meaningful alignment. We believe that one of the ways to address it is to split the model by meaningful subtasks at the architectural level. For instance, consider how our results on structure-only and single-cell-morphology -only (response to reviewer **F9Hk, [W1.1]**) naturally suggest fusing two complementary sub-modalities of tissue imaging. This approach shifts from using *extrinsic* evaluation to show the acquisition of biological concepts towards focusing on enforcing domain-informed inductive biases. Such ideas build on a long line of work on models like cell graphs (with morphological features, e.g. [3]) and hopefully, our findings tissue imaging provide an important perspective on it. We intend to clarify it in the “Future work” section of our revisions.
>
> We expect more and more biologically meaningful evaluations to converge to the behaviour expected in reference tasks on natural images, i.e. high scores for good models being indicative of relevant *learnt* semantic concepts. Even though those are still (to an extent) plagued by the presence of shortcuts. Finally, with models’ performance improving significantly, even originally flawed benchmarks might regain relevance.
>
> **References**
>
> [1] Kraus, O. et al. Rxrx3-core: A curated and compressed dataset and benchmarking task for drug-target interaction from high-content screening microscopy, 2025. https://arxiv.org/abs/2503.20158.
>
> [2] Sanchez, M. et al., Large scale compound selection guided by cell painting reveals activity cliffs and functional relationship, 2026. https://doi.org/10.1038/s42003-025-09500-y
>
> [3] Wang, S. et al., Deep learning of cell spatial organizations identifies clinically relevant insights in tissue images. doi: 10.1038/s41467-023-43172-8.

---

> > ### Author Rebuttal · Reviewer_2VWP · 2026-04-03
> >
> > The authors have adequately addressed my concerns. The downstream MIL study is particularly interesting.

---

### Official Review · Reviewer_F9Hk · 2026-03-12

**Soundness:** 3
**Presentation:** 2
**Significance:** 3
**Originality:** 3
**Overall Recommendation:** 4
**Confidence:** 3

**Summary:**

This paper does a number of evaluations on representation learning models for microscopy in the two most popular settings: cellpainting assay for cell cultures, and H&E stains in histology.

For **benchmarks**: In cellpainting, they use two benchmarks for retrieval: RxRx3-core and JUMP-CP; roughly, they test whether small embedding distances corresponds to images with the same perturbation. In histology, they use one benchmark for predicting gene expression from spatial transcriptomic data.

Evaluation method 1 is analysis of performance:
- the authors take recent foundation models in both fields, and propose some simple baselines: simple statistics, an untrained neural network, and (for histology) a graph representation of the cell organization designed to capture *only* relative cell positions.
- One surprising result is that on one CellPainting benchmark, some foundation models show only marginal improvement over ViT's trained on ImageNet, while *untrained* versions of the same models perform similarly. This suggests that most of the performance is due to some prior of transformers; pixel statistics were also a solid baseline.
- The other 2 benchmarks (one cell painting and one histology) did not have this pattern.
- For graph representation of cell organization, were surprisingly competitive, showing that the cell organization. The authors rightly point out that "cell count" is a possible shortcut.

Evaluation method 2 is layer-wise analysis:
- They evaluate performance on the same benchmarks at different layers of a given model and plot performance. For embedding models, one would expect deeper layers to be the strongest.
- One cellpainting model surprisingly has storngest results in intermediate layers (which was shown by prior work), while other show the expected pattern.
- For OOD models (models trained on natural images), there is a drop in performance with deepest layers, which is to be expected since there is distribution shift.

**Compliance With Llm Reviewing Policy:**

Affirmed.

**Final Justification:**

The major weaknesses listed were mostly addressed. I do remain concerned with the quality of presentation / clarity. The significance is good enough to justify a score of 4 - weak accept.

**Key Questions For Authors:**

In "weaknesses" above, each major weakness has a note on "How it could be addressed", which are the main questions.
There is no need to respond to minor weaknesses.

**Limitations:**

yes

**Strengths And Weaknesses:**

**Strengths**
- The high-level questions are interesting and important. There are now many attempts at building image foundation models in microscopy+pathology. But compared to the general image domain, there is not thorough evaluation.
- The idea of comparing models against simple baselines is excellent, and all three baselines are interesting. Moreover, the idea of the graph representation of cell centers is novel as a probe for a lower bound of task signal extractable from only cell organization.
- The findings in RxRxcount are very interesting.
- The point about "cell count" being a possible feature that dominates prediction (in the graph representation) is interesting.
- The choices of image domains - fluoro microscopy for cell cultures and H&E in histology - is appropriate, since they're the popular.

**Major Weaknesses**
This paper is an evaluation of existing models with existing benchmarks, so there must be a very high standard for quality of the evaluations.

(W1) The first major issue is comprehensiveness of benchmarks. There are only 3 benchmarks included in this paper, all pre-existing. This is despite the Related Work covering many others; one example would be CAMELYON16.
*How it could be addressed:* Add in multiple new benchmarks of different types; e.g. one classification and one subcellular task.

(W2) The second major issue is significance. Although the findings in Fig.2 (about RxRxcount & summarized in this review earlier) are interesting, they are unfortunately quite narrow, applying only to this one dataset. The finding about the strength of untrained networks is interesting, but not novel, having been discussed for non-microscopy domain in many papers (Zhong & Andreas). The findings about layer-wise performance are known from prior work (Kenyon-Dean), or not too surprising. The fact that general-image trained models are surprisingly competitive in microscopy tasks compared to trained models has been discussed, for example in (Doron et al 2003, "Unbiased single-cell morphology with self-supervised vision transformers")
*How it could be addressed:* It's difficult to address, but authors could first be very clear about what aspects of this work is novel, and argue that it is significant, perhaps by arguing how these findings would guide future benchmarking or model work.

(W3) A third issue is an important missing baseline, CellProfiler. A central contribution is proposing that the field consider simple baselines compared to fancy foundation models. However CellProfiler (Carpenter et al)  is already a simple baselines that is extremely popular in microscopy, and it appears in many of the cited papers as a baseline.  BTW, CellProfiler in its standard pipeline, ignores any kind of cell organization and instead focusing on pixel features. It would therefore be a nice complement to the paper's graph representation, which ignores pixel features and instead focuses on cell organization.
*How it could be addressed:* add CP as one of the baselines. Adjust analysis and recommendations accordingly.

(W4) The fourth major issue is the persuasiveness of the arguments
- The abstract makes a strong claim with respect to the baselines that "Our results show that, surprisingly, state-of-the-art methods perform comparably to these baselines." But the results section only claim this is true for the RxRx benchmark, but not others.
- It is difficult to figure out the key findings from the introduction.
- Later in the introduction, certain challenges in microscopy are laid out, which are great motivations: "Microscopy experiments are sensitive to experimental conditions, batch effects, and spatial or wellplate-layout artifacts, and deep learning models can learn these signals instead of the underlying biology". Then it claims that studies on transcriptomics studies "have shown that confounding structure in the data can strongly influence performance estimates", and suggest that this work will do the same. But the paper does not test for confounders or batch effects in a controlled way, instead showing that simple features can do okay on some tasks.
*How it could be addressed:* Rewrite the abstract & intro to be very clear.



**Some minor weaknesses**
- Communication of key ideas could be much stronger
	- The figures do not highlight the key ideas. For example, the paper's first figure shows the graph representation of H&E images, which is only one smaller experiment.
	- For a new reader, it's hard to get the key takeaways from the charts without much effort and reading the text. E.g. in Fig.5, the legend could explain "the untrained models in gene-gene retrieval have similar score to the trained model".
- The Fig.2 legend says that left panel is Imagenet-1k and right panel is gene-gene but that seems inconsistent with the y-axis label.

---

> ### Author Rebuttal · Authors · 2026-03-30
>
> We thank the reviewer for the detailed constructive review and for highlighting where our messaging lacks clarity. We detail our modifications below:
>
> * **[W1a]** We add cancer subtyping datasets: BRACS-RoI [1] (7 classes) and BACH [2] (4 classes). We follow the processing and evaluation pipeline from [3] adding H-Optimus-1, a histology FM for reference. For each model we report the results for the best MIL and patchification strategy on 5-fold CV with a held-out test set. Please, refer to our discussion with **X4y4, [Q2]** for the reported metrics.
>
> * **[W1b]** We add a new subcellular task following [4]: the evaluation of mean (across MoA) percent replicating (well replicates retrieval) and percent matching retrieval of same mechanism of action across 43 MoAs with 2 compounds each, present in the JUMP-MoA datasets. Untrained models outperform the theoretical random and are as good as the pretrained ones to retrieve matching MoAs. Below we provide an excerpt (with the best pretrained and untrained model) from the results:
>
> | Model | Mean % of Replicating | Mean % of Matching |
> |---|---|---|
> | Random | 5 | 5 |
> | Pixel Stats | 22.3 | 10.5 |
> | OpenPhenom | **37.8** | 11.4 |
> | ResNet152 *(untrained)* | 27.3 | 11.6 |
> | ViT-l/16 *(pretrained)* | 36.8 | 12.3 |
> | SingleConv *(untrained)* | 30.5 | **12.8** |
>
> **[W2]** We clarify how we deviate from established ideas and how our analysis helps future work:
>
> 1. We use untrained networks to “evaluate the evaluation” rather than focusing on their capabilities. Notably, we contextualise the results on the subset of JUMP controls and show limitations of RxRx3-core and MoA matching (until the latter possibly becomes saturated). Then, our analysis of representation similarity alignment groups together models from the same architecture family (Fig 6 & 23) rather than from the same pretraining domain. Finally, our comprehensive sweep of ViTs and ResNets provides a useful starting point for further work.
>
> 2. Our layer-wise analysis is not aimed at finding the best performing layer (as in Kenyon-Dean et al., 2025) but at highlighting a systematic lack of learnt *high-level* abstractions. Figs 5 &18b show that in-domain OpenPhenom at its best stages is still comparable to early stages of OOD-pretrained ViTs. We believe that this adds important context and hence advocate for layer-wise experiments to be conducted primarily as a diagnostic tool: do we manage to perform better than low-level features of known models? This perspective allows us to highlight the strengths of H-Optimus-1 in Figure 7.
>
> The strengths of general image backbones in transfer learning is indeed a known observation. We do not claim novelty here and use them as discussed in our response to **X4y4, [Weaknesses]**.
>
> * **[W3a]** We add CellProfiler features (using standard pipelines for CellPainting) as a baseline. They land in between untrained models and DINOv2 on JUMP-CP controls and in between pixel_stats and untrained models on RxRx3-Core, confirming our findings.
>
> | Model | Average Recall@5% (± Std) |
> |:---|:---|
> | Cell Profiler | 0.17±0.01 |
> | pixel_stats | 0.16±0.02 |
> | ViT-l/16 *(untrained)* | **0.19±0.02** |
> | OpenPhenom | **0.20±0.02** |
>
> * **[W3b]** For BRACS and BACH we extract similar textural and geometric features of cell nuclei using `scikit-image` as in [2].
>
> **[W4]** We propose the following edits (shortened):
>
> Abstract, L023-026: *“sota models perform comparably to these baselines, on a subset of evaluation tasks limiting the range of conclusions that can be derived from these results.”*
>
> The last paragraph of the introduction:  Incorporate key contributions as follows:
>
> 1. We use simple baselines to demonstrate shortcomings of several popular benchmark evaluations.
>
> 2. We analyse layer-wise representations to raise questions about lack of consistent acquisition of learnt high-level biologically relevant abstractions.
>
> 3. We demonstrate relevance of structure-only views of tissue as a strong but incomplete submodality, suggesting future work to develop principled representations of tissues.
>
> 4. To clarify L027-031, L037-040: add in the intro *“Instead of focusing on the confounding structures, we propose to study the benchmarks by analysing their discriminative capabilities w.r.t. the models of interest and baselines that do not possess biologically relevant representations by construction or operate on strongly ablated input information.”*
>
> **References**
>
> [1] Brancati, N. et al., Bracs: A dataset for breast carcinoma subtyping in H&E histology images, 2022. https://doi.org/10.1093/database/baac093
>
> [2] Aresta, G. et al., Bach: Grand challenge on breast cancer histology images, 2019. https://doi.org/10.1016/j.media.2019.05.010
>
> [3] Ogut, S. et al., Graphist: Graph self-supervised learning for histopathology, 2026. https://arxiv.org/abs/2603.00143
>
> [4]  Stossi, F. et al., SPACe: an open-source, single-cell analysis of Cell Painting data, 2024. https://doi.org/10.1038/s41467-024-54264-4

---

> > ### Author Rebuttal · Reviewer_F9Hk · 2026-04-02
> >
> > Thanks for the response. I've raised the score from 2 to 4 (weak accept).
> >
> > As I said in the final justification: I do remain concerned with the quality of presentation / clarity and suggest that the authors work on this for the final version. The significance is good enough to justify a score of 4.

---

### Official Review · Reviewer_NTHo · 2026-03-13

**Soundness:** 3
**Presentation:** 2
**Significance:** 3
**Originality:** 4
**Overall Recommendation:** 4
**Confidence:** 3

**Summary:**

The paper shows that microscopy deep learning models may not truly learn biological features, because simple baselines and even untrained networks can achieve similar benchmark performance. It argues that the field needs better benchmarks and evaluation methods to verify real biological representation learning.

**Compliance With Llm Reviewing Policy:**

Affirmed.

**Key Questions For Authors:**

a.	The paper argues that common benchmark metrics may hide important limitations. It would be helpful to clarify which component of the benchmark is primarily problematic: the dataset itself, the task formulation, the aggregation protocol, or the evaluation metric. Understanding this distinction could provide clearer guidance on where improvements are most needed.

b.	It would be helpful to further clarify the motivation for including untrained models in the experiments. Competitive performance from untrained models could indicate that the task is relatively easy, or that architectural inductive biases alone are sufficient for the objective. A more detailed explanation of how the authors interpret this result would strengthen the conclusions.

c.	The paper highlights the need for better benchmarks to properly evaluate biological representation learning. However, it would be useful if the authors could outline specific properties or design principles that such benchmarks should include. In addition, while the diagnostic framework identifies limitations in current evaluations, it does not provide clear guidance on how future models or evaluation setups should be designed to address these issues.

**Limitations:**

yes

**Strengths And Weaknesses:**

a.	**Soundness:** The paper is technically sound and the authors design experiments by comparing pretrained models, untrained models, and simple baselines across multiple microscopy benchmarks. The experiments are logical and support the main claims of the paper. The results are generalised across datasets and are further supported with additional analyses.

b.	**Presentation:** The paper is generally well written, but some sections are harder to understand. Especially, the introduction could be further improved by clearly adding the main takeaways. Currently, these are not strongly emphasized.

c.	**Significance:** The authors address an important problem in representation learning, especially in biological imaging. Although existing works report good performance, the authors experimentally evaluate that these models may not actually learn biological representations. They demonstrate this through simple but meaningful experiments using simple baselines, which perform competitively with complex models.

d.	**Originality:** The paper is original from a diagnostic perspective rather than an algorithmic one. It does not propose a new model, but it introduces a systematic diagnostic evaluation framework combining simple baselines, untrained models, and structure-only representations. The novelty lies in demonstrating that current microscopy benchmarks may overestimate representation learning capabilities, which provides new insight into the behavior of existing models. This type of analysis contributes to a better understanding of representation learning in biological imaging.

---

> ### Author Rebuttal · Authors · 2026-03-30
>
> We thank the reviewer for detailed feedback. We first address the key question and then propose modification to the current presentation focusing on the main takeaways and as well as perspectives on future development of models and evaluation tasks.
>
> **[Qa]** Our experiments on RxRx3-core suggest that the issue is the combination of Quality Controls (QC) with the actual data samples, the random baseline not informative enough, and the biological task (retrieving all literature links between genes from KO of genes using Cell Painting in one cell line) being very difficult.
>
> **[Qb]** The relevance from the untrained models can be argued for from two different perspectives.
>
> 1. *We use untrained models to highlight the limitations of metric-driven conclusions for a given dataset.* Reasonably general architectures (e.g. ViTs/ CNNs, and not biology-informed architectures with in-domain priors) do not possess meaningful biological abstractions when untrained. Thus, any claim of learning biologically meaningful representations with a model of interest that is only supported by downstream tasks where untrained models perform well is not substantiated (we do emphasize that the model of interest might in fact have learnt relevant biological abstractions).
>
> 2. *Untrained models may indeed help us recognise relevant inductive biases.* We notably refer to the work of [1] as discussed in the main text. Our experiments on RxRx3core and specifically layer-wise evaluation shows that the performance of CNN-derived features drops with depths significantly, compared to ViTs (e.g. Fig 15, b). The ViTs we consider are isotropic, they do not feature dimensionality reduction beyond the initial CNN patchification layer. We would prefer to abstain from drawing strong conclusions from limited (and likely) flawed datasets, nonetheless, it appears reasonable that preservation of low-level textural features in the deeper layers of the model is a beneficial inductive bias for cell culture imaging.
>
>
> **[Qc]** We partially address the evaluation practices in sec. 6 Implications. We will complement it with several remarks regarding the design of evaluation tasks below:
>
> 1. A fundamental starting point is addressing QC on images (see Fig 10).
>
> 2. Relevant baselines should be selected other than theoretical random to highlight relative complexity of the task.
>
> 3. Benchmarks should exhibit clear separation between trained / untrained / baselines / and ideally ID and OOD models
>
> 4. More emphasis should be put on benchmarks containing baseline subtasks, i.e. replication of basic biological desiderata, besides addressing a more ambitious task. A “positive” example of this might be enforced alignment to a subset of a knowledge graph of interest. A “negative” example can be verifying insensitivity to known spurious factors.
>
> 5. With that, if possible, realistic upper bounds on the values of metrics should be provided. For instance, one can use biological replicates as “best biological predictors assuming reasonable QC”.
>
> Regarding perspectives on how models should be designed we refer to our discussion with reviewer **2VWP**. We will expand our “Implications” section to feature “Future work” discussing directions for model design.
>
> **Presentation:**
>
> We will modify the last paragraph of the introduction to clarify the main conclusions. The main changes are detailed in our discussion with reviewer **F9Hk**. We will also expand our “Implications” sections with outlines for future work, following ideas expressed in the response to **2VWP**.
>
> **References**
>
> [1] Ziqian Zhong and Jacob Andreas. Algorithmic capabilities of random transformers, 2024. https://dl.acm.org/doi/10.5555/3737916.3741231

---

> > ### Author Rebuttal · Reviewer_NTHo · 2026-04-03
> >
> > Thank you for the responses. The authors have addressed my concerns.

---

> > > ### Author Response · Authors · 2026-04-07
> > >
> > > Dear reviewer, we are happy to hear that we managed to provide satisfactory clarifications. Considering this, could the reviewer kindly consider raising the score to reflect the addressed concerns?

---

### Official Review · Reviewer_X4y4 · 2026-03-13

**Soundness:** 3
**Presentation:** 3
**Significance:** 4
**Originality:** 4
**Overall Recommendation:** 6
**Confidence:** 4

**Summary:**

This paper systematically investigates what microscopy foundation models actually learn by comparing state-of-the-art pretrained models against intentionally simple baselines (pixel statistics, untrained networks, and structure-only tissue representations) on cell culture (RxRx3-core, JUMP-CP) and tissue imaging (HEST-1k) benchmarks. The central finding is that on several benchmarks, simple baselines perform comparably to sophisticated pretrained models, suggesting that current benchmarks may reward low-level cues and architectural priors rather than biologically meaningful representations. The paper advocates for more diagnostic benchmarks and stronger baseline reporting.

**Compliance With Llm Reviewing Policy:**

Affirmed.

**Final Justification:**

concerns are adequately addressed

**Key Questions For Authors:**

Can you provide ablations isolating which specific architectural properties of ViTs (patch embedding size, attention mechanism, CLS token aggregation vs. average pooling) drive the strong untrained performance on RxRx3-core?

Have you looked at any tumor subtyping or mutation prediction histology tasks?

The recent GraphHist paper may be a worthwhile comparison to the cell structure baseline?

**Limitations:**

The limitations are adequately discussed.

**Strengths And Weaknesses:**

The paper studies a very important and timely question. There has been a lot of recent literature on microscopy foundation models, but it is unclear if the models are learning meaningful representations, just like has been observed in transcriptomic foundation model literature where basic baselines beat foundation models. So here, this paper proposes various baselines to test this. (Significance)

The paper includes well-designed baselines that are genuinely informative. The three baseline types isolate a different axis of confounding: pixel statistics test whether spatial structure matters at all, untrained models isolate architectural inductive bias from learned features, and the structure-only baseline disentangles tissue organization from morphology/texture. (Soundness/Originality)

The paper overall is well-structured with a clear narrative. (Presentation)

Weaknesses:
Overall the results are quite mixed and in my opinion a bit inconclusive. While it is clear that baselines reach similar performance to foundation models for RxRx3, this is not the case for JUMP-CP and HEST-1k. This may just indicate RxRx3 is a poor dataset rather than the models aren't learning useful signal.

Even if the cell tissue structure baseline reached foundation model performance (which it really doesn't), it does not invalidate the foundation model representations. Because even if the foundation model is following this structure "shortcut", it had to learn how to segment cells and understand how the structure corresponds to the downstream task.

---

> ### Author Rebuttal · Authors · 2026-03-30
>
> We would first like to thank the reviewer for a thorough reading of our work and the detailed remarks.
> Below we address the key questions followed by a discussion of the weaknesses.
>
> **[Q1]** We ran experiments for additional configurations of untrained ViTs. We provide recall results on RxRx3-core for untrained ViT models. We would like to clarify which versions of the attention mechanism the reviewer has in mind, which we could then add to our experiments below:
>
> | Hyperparameter | Level | Mean Recall@5% |
> |---|---|---|
> | *aggregation* | avg | 0.19±0.02 |
> | | cls | 0.20±0.02 |
> | *size* | s | 0.20±0.02 |
> | | b | 0.19±0.02 |
> | | l | 0.19±0.02 |
> | | h | 0.18±0.02 |
> | *patch size* | 4 | 0.20±0.03 |
> | | 8 | 0.20±0.02 |
> | | 14 (h) / 16 (s,b,l) | 0.20±0.02 |
> | | 32 | 0.17±0.02 |
> | *layer depth* | 0 | 0.20±0.02 |
> | | 1 | 0.19±0.02 |
> | | 2 | 0.19±0.02 |
> | | 3 | 0.19±0.02 |
>
> We see that across aggregation, model size, patch size and layer depth, the patch size is what could explain the beneficial inductive bias (as models are untrained and the shallow layers are consistently the best ones).
>
> **[Q2]** We added two subtyping datasets: BRACS [1] and BACH [2]. We conclude that structure only baselines a) perform above random on a 7-class (adjusted for class imbalances) and a 4-class (balanced) subtyping problems providing an important reference for the *benchmark metrics*, b) are competitive w.r.t. vision encoders commonly used as baselines, e.g. vanilla CNNs in [3], c) the ablated structure-only modality provides a reference for development of *structure-morphology fusion architectures*.
>
> We have computed the metrics for all encoders for images of graphs from the main text. Below we provide an excerpt of the results:
> | Model | Encoder training | BRACS (Test F1) | BACH (Test F1) |
> |---|---|---|---|
> | *Hand-crafted* | | |
> | Morphological feats of nuclei | - | **0.57±0.01** | **0.62±0.02** |
> | *Supervised graph* | | |
> | ACM-bio | from scratch | 0.22±0.03 | 0.38±0.02 |
> | *SSL vision* | | |
> | DINOv2 | pt Histology | 0.51±0.01 | 0.59±0.03 |
> | MAE | pt Histology | 0.56±0.01 | 0.57 ± 0.04 |
> | GraphHist | pt Histology | 0.60±0.01 | **0.69±0.01** |
> | H-optimus-1 | pt Histology | **0.74±0.01** | 0.56±0.06 |
> | *Img of graphs, ours* | | |
> | resnet152 | pt IN1k | **0.53±0.01** | 0.52±0.05 |
> | resnet34 | pt IN1k | 0.46±0.02 | **0.52±0.04** |
>
> **[Q3]** We evaluated our structure-only baseline on datasets where the results for GrapHist are publicly available. Notably, our zero-shot baseline outperforms supervised graph-based training from [4] but does not reach the performance of in-domain pretrained graph-based model. This is expected under reasonable optimisation as the model from GrapHist [4] is both structure *and* cell morphology -based.
>
> **[Weaknesses]** We thank the reviewer for the important remark. Flawed benchmarks are not a demonstration of the models’ inability to learn relevant signal. In tissue imaging in particular, the task-specific relevance of foundation models’ representation is more convincing from the results on HEST-1k.  However, we would like to emphasise several additional points
>
> 1. In the context of representation learning for bioimaging the usefulness of general purpose vision backbones is a much weaker claim than learning relevant biological abstractions (which is what the benchmarks like RxRx3-core are aimed at). For instance, DINOv2 scores highly on a curated subset of JUMP, and has found a plethora of applications across the biomedical domain, while staying a generalist image encoder, not aligned to represent biological mechanisms in its latent space.
>
> 2. The ability of foundation models to leverage low-level shortcuts is not an issue in itself. The main problem arises when benchmarks that admit such shortcuts are used to derive conclusions regarding high-level abstractions in foundation models. The baselines are aimed to address this issue.
>
> 3. The layer-wise analysis helps check the acquisition of relevant biological abstractions. We clarify our claims on this matter in discussion with reviewer **NTHo**.
>
> 4. We have also added new experiments of MoA classification as discussed with **F9Hk**, where the results are again strong for the untrained models, which extends the list of applications where untrained models are currently competitive with FMs.
>
> **References**
>
> [1] Brancati, N. et al., Bracs: A dataset for breast carcinoma subtyping in H&E histology images, 2022. https://doi.org/10.1093/database/baac093
>
> [2] Aresta, G. et al., Bach: Grand challenge on breast cancer histology images, 2019. doi:https://doi.org/10.1016/j.media.2019.05.010
>
> [3] Pati, P. et al., Hact-net: A hierarchical cell-to-tissue graph neural network for histopathological image classification, 2020. https://doi.org/10.1007/978-3-030-60365-6_20
>
> [4] Ogut, S. et al., Graphist: Graph self-supervised learning for histopathology, 2026. https://arxiv.org/abs/2603.00143

---

> > ### Author Rebuttal · Reviewer_X4y4 · 2026-04-04
> >
> > My concerns are fully resolved and I increase my score accordingly. I appreciate the BRACS/BACH results, adding GraphHist, and the discussion about the weaknesses. That discussion should be made clear in the revised version of the paper.

---

### Decision · Program_Chairs · 2026-04-30

**Decision:**

Accept (regular)

**Comment:**

This paper received overall positive evaluations, with two Weak Accept and two Strong Accept recommendations.

All the reviewers acknowledge the importance of the objective of this work, that is what current representation learning methods are learning, in the context of biological imaging. The topic is considered relevant, and the scientific questions are overall seen as timely and original, especially in terms of evaluating state-of-the-art models besides performance, exploring whether they learn meaningful biological features, implemented through experiments with appropriate baselines and models.

On the other hand, reviewers raised some concerns, including the need for more benchmarks, to reinforce the claims of this paper, doubts on significance of the specific findings, as some are considered narrow or not novel, and the request for including other relevant baselines.
Other identified weaknesses concerned the presentation of the paper, which could have been stronger in communicating the key ideas, or to be improved in terms of specific sections, for instance including restructuring the introduction to reveal the main findings of the paper.

In their rebuttal, the authors provided additional clarification and experiments.

The reviewers generally acknowledged and appreciated the details and specifications provided in the rebuttal, with most of the initial concerns considered resolved, as reflected by improved or maintained positive recommendations, with only doubts regarding presentation and clarity persisting in one reviewer.

After careful evaluation of the initial reviews, the rebuttal and discussion with the authors, the AC recognizes the value of this work, and agrees with the identified strengths, especially in terms of the significance of the posed research questions and the relevance of the application topic.

Therefore, the paper is recommended for acceptance, given that the committed changes are included in the camera-ready version of this work.

As a side note, the reference *Vision transformers need registers* in the main paper contains an incorrect author list, including an author who does not appear on the cited paper.
The correct citation is: *Darcet, T., Oquab, M., Mairal, J. and Bojanowski, P., 2023. Vision transformers need registers. arXiv preprint arXiv:2309.16588*.

The authors are requested to fix this reference in the camera-ready.